# Unraveling the Interplay between Carryover Effects and Reward Autocorrelations in Switchback Experiments

Qianglin Wen [*1]   Chengchun Shi [*2]   Ying Yang [3]   Niansheng Tang [1]   Hongtu Zhu [4]

## Abstract

A/B testing has become the gold standard for policy evaluation in modern technological industries. Motivated by the widespread use of switchback experiments in A/B testing, this paper conducts a comprehensive comparative analysis of various switchback designs in Markovian environments. Unlike many existing works which derive the optimal design based on specific and relatively simple estimators, our analysis covers a range of state-of-the-art estimators developed in the reinforcement learning (RL) literature. It reveals that the effectiveness of different switchback designs depends crucially on (i) the size of the carryover effect and (ii) the auto-correlations among reward errors over time. Meanwhile, these findings are estimator-agnostic, i.e., they apply to most RL estimators. Based on these insights, we provide a workflow to offer guidelines for practitioners on designing switchback experiments in A/B testing.

## 1. Introduction

***Motivation.*** Policy evaluation has become increasingly important in applications such as economics (Athey & Imbens, 2017), medicine (Luedtke & Van Der Laan, 2016), environmental science (Reich et al., 2021) and epidemiology (Hudgens & Halloran, 2008). In the technology sector, companies such as Google, Amazon, Netflix, and Microsoft extensively use A/B testing to measure and improve the effectiveness of new products or strategies against established ones (e.g., Johari et al., 2017; Waudby-Smith et al., 2022). For instance, ridesharing platforms like Uber, Lyft and Didi

continuously refine their policies for order dispatching and subsidies to optimize key metrics such as supply-demand balance, driver earnings, response rates, and order completion rates (Qin et al., 2025). A/B testing is a common tool used by these companies for policy evaluation and is vital for identifying the most effective strategies to enhance the efficiency and convenience of the transportation system (Xu et al., 2018; Tang et al., 2019; Zhou et al., 2021).

***Challenges.*** In numerous applications, treatments are assigned sequentially over time (Robins, 1986; Bojinov & Shephard, 2019), posing unique challenges for A/B testing:

1. A primary challenge is the **carryover effect** – the effect of previous treatments on future outcomes. Such effects are ubiquitous in many applications. For instance, in ridesharing, the implementation of a specific order dispatch policy can change the spatial distribution of drivers in a city, impacting subsequent outcomes (see Li et al., 2024, Figure 2). These carryover effects substantially challenge A/B testing. Standard solutions like the two-sample $t$-test often fail to capture these effects, frequently resulting in insignificant $p$-values (Shi et al., 2023b).

2. Another challenge is the **limited sample size**, coupled with generally **weak treatment effects**, making it extremely difficult to determine the most effective policy. This issue is particularly prevalent in online experiments in ridesharing, which seldom last more than two weeks and typically exhibit effect sizes ranging from $0.5\%$ to $2\%$ (Xu et al., 2018; Tang et al., 2019).

***Contributions.*** This paper conducts a quantitative analysis to understand the effects of various switchback designs on the precision of their resulting policy value estimators. Switchback designs alternate between a baseline and a new policy at fixed intervals. Each policy is implemented for a specified duration before transitioning to the other. When the duration of each policy extends to a full day, the design becomes an "alternating-day" (AD) design, involving daily policy switches. These designs are increasingly utilized in large-scale ridesharing platforms (Xiong et al., 2024; Qin et al., 2025). Luo et al. (2024) has empirically demonstrated that more frequent policy alternations can reduce mean squared error (MSE) in estimating the average treat-

[*]Equal contribution  [1]Yunnan Key Laboratory of Statistical Modeling and Data Analysis, Yunnan University  [2]Department of Statistics, London School of Economics and Political Science  [3]Center for Applied Mathematics, Shanghai Key Laboratory for Contemporary Applied Mathematics, Fudan University  [4]Department of Biostatistics, The University of North Carolina at Chapel Hill. Correspondence to: Hongtu Zhu <htzhu@email.unc.edu>.

*Proceedings of the 42ⁿᵈ International Conference on Machine Learning*, Vancouver, Canada. PMLR 267, 2025. Copyright 2025 by the author(s).

ment effect (ATE). However, the mechanisms driving the improvement in estimation accuracy lack thorough exploration. Our study fills this gap by offering a comprehensive comparative analysis of switchback experiments within a reinforcement learning (RL, Sutton & Barto, 2018) framework, where the experimental data follows a Markov decision process (MDP, Puterman, 2014) model.

Importantly, our analysis unravels the interplay between carryover effects and reward auto-correlations in determining the optimal switchback experiment. In particular, when the **carryover effect is weak**, we show that:

(i) In scenarios with **positively correlated** reward errors, the precision of the ATE estimator tends to be improved with more frequent alternations between policies. This leads us to an interesting conclusion: the off-policy ATE estimator under switchback designs outperforms its on-policy counterpart under the AD design in terms of estimation efficiency. This conclusion remains valid even in the presence of some negatively correlated errors. The superior efficiency of the switchback design is attributed to the switchback design's inherent capability to neutralize the influence of autocorrelated errors over time, leading to a more accurate estimator. This insight has not been systematically documented in existing literature, to our knowledge. We also remark that positively autocorrelated errors are commonly observed in practice, as demonstrated in Figure 1.

(ii) When reward errors are **uncorrelated**, all designs become asymptotically equivalent in theory. Our numerical studies indicate that AD generally exhibits superior performance in finite samples.

(iii) When the majority of errors are **negatively correlated**, AD becomes the most efficient.

Additionally, with **a large carryover effect**, AD or switchback designs with less frequently switches work the best.

These findings apply to a range of policy value estimators developed in the RL literature, such as model-based estimators, least squares temporal difference (LSTD) estimators, and double reinforcement learning (DRL) estimators (see Uehara et al., 2022, for a review). While existing works also studied switchback designs (refer to the next section), they primarily focused on the use of simple importance sampling (IS) estimators for policy evaluation. According to our numerical studies, IS estimators suffer from much larger MSEs than ours (see Figure 5 for details).

## 2. Related Works

Our paper intersects with three related lines of research: A/B testing, off-policy evaluation and experimental designs.

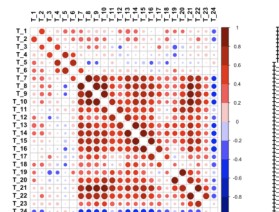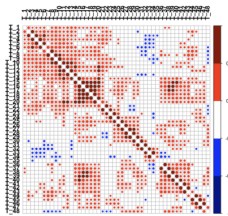

*Figure 1.* The estimated correlation coefficients between pairs of fitted reward residuals, based on two datasets provided by a ridesharing company. Most residual pairs are non-negatively correlated, with a large proportion exhibiting positive correlation. The diagonal components have been omitted to enhance clarity.

*A/B testing.* A/B testing has been widely adopted across tech companies (see Larsen et al., 2024; Quin et al., 2024, for reviews of methodologies). It relies on causal inference to estimate treatment effects, typically assuming "no interference" or the stable unit treatment value assumption (SUTVA, see e.g., Imbens & Rubin, 2015). However, as noted earlier, this assumption can be violated in temporally-dependent experiments, rendering most A/B testing methods unsuitable for switchback designs.

Recently, there is a growing interest in developing A/B testing and/or causal inference solutions in the presence of interference effects. Depending on the type of interference, these papers can be grouped into four categories:

- The first category studies spatial or network interference effects where the policy implemented in one location can affect outcomes at other locations (see Pollmann, 2020; Tchetgen Tchetgen et al., 2021; Leung & Loupos, 2022; Bhattacharya & Sen, 2024; Dai et al., 2024; Jia et al., 2024; Shirani & Bayati, 2024; Zhang et al., 2024, for some recent proposals).

- The second category focuses on temporal carryover effects and is the most relevant to our proposal (see e.g., Robins, 1986; Sobel & Lindquist, 2014; Boruvka et al., 2018; Liang & Recht, 2025; Viviano & Bradic, 2023). Notably, there is a line of papers that proposed to employ the RL framework that models the observed data via MDPs to capture the carryover effects (Farias et al., 2022; 2023; Shi et al., 2023b).

- The third category handles both interference effects over time and space (see e.g., Jia et al., 2023; Shi et al., 2023a).

- Finally, the last category focuses on interference effects that appear in two-sided markets or recommender systems (Munro et al., 2021; Johari et al., 2022; Zhan et al., 2024).

However, the effectiveness of different experimental designs are less explored in these papers, which is our primary focus.

*Off-policy evaluation (OPE).* OPE methods generally fall into two categories: model-based and model-free approaches. Model-based methods estimate an MDP model from offline data and compute policy value based on the estimated model (Gottesman et al., 2019; Yin & Wang, 2020; Wang et al., 2024; Yu et al., 2024). On the other hand, model-free methods can be further classified into three subtypes: (i) Value-based approaches, such as LSTD, focus on estimating the policy value via an estimated Q-function or value function (Bradtke & Barto, 1996; Sutton et al., 2008; Luckett et al., 2020; Hao et al., 2021; Liao et al., 2021; Chen & Qi, 2022; Shi et al., 2022; Li et al., 2023a; Liu et al., 2023; Cao & Zhou, 2024; Bian et al., 2025); (ii) IS-type methods adjust rewards by the density ratio between target and behavior policies (Thomas et al., 2015; Liu et al., 2018; Nachum et al., 2019; Xie et al., 2019; Dai et al., 2020; Wang et al., 2023; Hu & Wager, 2023; Thams et al., 2023; Zhou et al., 2025); (iii) Doubly-robust methods such as DRL combine value-based and IS methods for more robust policy evaluation (Zhang et al., 2013; Jiang & Li, 2016; Thomas & Brunskill, 2016; Bibaut et al., 2019; Uehara et al., 2020; Kallus & Uehara, 2020; 2022; Liao et al., 2022; Xu et al., 2023; Shi et al., 2024).

Despite the popularity of developing advanced OPE estimators, the strategies for generating offline data to maximize their estimation efficiency have not been thoroughly investigated. Existing works either focus on a contextual bandit setting without carryover effects (Wan et al., 2022) or do not study switchback designs (Hanna et al., 2017; Mukherjee et al., 2022; Zhong et al., 2022; Li et al., 2023b). Our paper fills this gap by investigating how switchback designs affect the efficiency of various OPE estimators, including model-based, value-based, and doubly-robust estimators, providing a comprehensive analysis that enriches the OPE literature.

*Experimental design*. There is a rich literature on experimental designs tailored for clinical trials, with a range of proposed optimal designs (Begg & Iglewicz, 1980; Wong & Zhu, 2008; Jones & Goos, 2009; Atkinson & Pedrosa, 2017; Rosenblum et al., 2020) and sequential adaptive designs (Hu et al., 2009; Baldi Antognini & Zagoraiou, 2011; Atkinson & Biswas, 2013; Hu et al., 2015; Kato et al., 2024) to guide treatment allocation strategies. However, these methods are developed under settings where data are identically and independently distributed and are thus not applicable to our settings in the presence of carryover effects.

Recent developments have expanded the scope to accommodate spatial or network spillover effects (Ugander et al., 2013; Li et al., 2019; Kong et al., 2021; Leung, 2022; Ni et al., 2023; Viviano et al., 2023; Yang et al., 2024; Zhang & Wang, 2024; Zhu et al., 2025) and to address the complex interactions inherent in two-sided marketplaces (Bajari et al., 2021; Li et al., 2022; Bajari et al., 2023). Despite these advancements, a gap remains concerning designs that adequately account for temporal carryover effects in sequential decision making.

Glynn et al. (2020), Hu & Wager (2022), Bojinov et al. (2023), Basse et al. (2023), Sun et al. (2024) and Xiong et al. (2024) studied the design of temporally-dependent experiments. In particular, Xiong et al. (2024) made an important step forward for understanding the trade-offs among switchback designs by deriving a rigorous bias-variance decomposition of the ATE estimator and summarizing four key factors that determine the estimation error. However, their analysis is confined to simple IS estimators derived within a bandit framework, which can be severely biased with large carryover effects. Our empirical studies in Appendix C also confirm that IS-type estimators are prone to large MSEs. Moreover, Xiong et al. (2024)'s analysis did not adopt our MDP framework, which is commonly utilized for policy learning and evaluation in the motivating ridesharing application (Xu et al., 2018; Shi et al., 2023b).

Finally, recent works in the machine learning literature have developed deep learning or RL algorithms to numerically compute optimal designs (Foster et al., 2021; Blau et al., 2022; Lim et al., 2022).

## 3. Preliminaries

In this section, we describe the data, detail our model, introduce switchback designs and formulate our objective.

*Data.* Suppose a technology company conducts an online experiment over $n$ days to evaluate the effectiveness of a new policy compared to a baseline policy. Each day is divided into $T$ non-overlapping intervals. For each day $i = 1, \ldots, n$ and for each time interval $t$, let $S_{i,t} \in \mathbb{R}^d$ denote certain market features (e.g., the number of available drivers and pending ride requests in ridesharing) observed at the beginning of the interval. The policy in effect during each interval $t$ is represented by $A_{i,t}$, which, in the context of A/B testing, is a binary variable indicating one of two policies. Finally, $R_{i,t} \in \mathbb{R}$ denotes the immediate outcome or reward observed at the end of each interval $t$ (e.g., the total revenue at time $t$). We assume all trajectories $\{(S_{i,t}, A_{i,t}, R_{i,t}) : 1 \leq t \leq T\}_{i=1}^n$ are i.i.d. instances of a stochastic process $\{(S_t, A_t, R_t) : 1 \leq t \leq T\}$. That is, the data are independent across different days. This assumption is likely to hold in applications such as marketing auctions where each company's budget resets at the end of the day, eliminating any carryover effects across days (Basse et al., 2016; Liu et al., 2020), and in ridesharing where order volume typically wanes between 1 am and 5 am (Luo et al., 2024, Figure 1), making it plausible that each day's observations can be treated as independent realizations. Additionally, in online advertising, impression

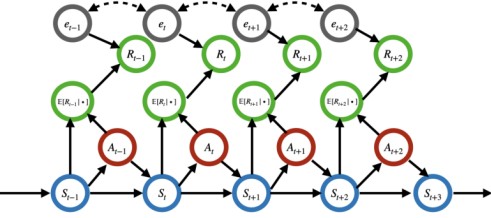

*Figure 2.* Visualization of our MDP with autocorrelated reward errors. The solid lines represent the causal relationships. The dash lines imply that the reward errors are potentially correlated.

allocation often follows a daily schedule, reinforcing the assumption of independent data across days.

***Model.*** We model the experimental data by a finite MDP with autocorrelated errors, based on three assumptions:

(i) First, we assume that the state satisfies a Markov assumption. Specifically, we require

$$\mathbb{P}(S_{t+1} = s'|A_t, S_t, \{S_j, A_j, R_j\}_{j<t}) = p_t(s'|A_t, S_t),$$

for any $s'$ and $t$. This assumption requires the future state to be conditionally independent of the past data history given the current state-action pair, and is consistent with a wide body of work in RL (Sutton & Barto, 2018).

(ii) Second, we assume the reward satisfies a conditional mean independence assumption: there exists a sequence of reward functions $\{r_t\}_t$ such that for any $a$ and $s$,

$$\mathbb{E}(R_t|A_t = a, S_t = s, \{S_j, A_j\}_{j<t}) = r_t(a, s).$$

Such an assumption is commonly imposed in the literature (Chernozhukov et al., 2022; Shi et al., 2022; Wang et al., 2023).

(iii) Third, the residual errors $e_t = R_t - r_t(A_t, S_t)$ can exhibit temporal correlation. If the residuals are uncorrelated, the resulting data-generating process simplifies to a standard MDP.

A graphical visualization of our model is given in Figure 2. Notably, while our methods rely on the MDP assumption, we also consider cases where this assumption may not hold; refer to Appendix B for details.

Finally, we remark that both the reward and transition functions are explicitly indexed by $t$. This is essential to capture the time-dependent dynamics that are often inherent in practical applications (Luo et al., 2024).

***Designs.*** We introduce the *switchback design* frequently utilized in practice. Under this design, the company alternates between the two policies, each for a fixed duration per day. Let $m \geq 1$ represent the time span for each switch. A smaller value of $m$ corresponds to more frequent switching between policies. To illustrate this design, consider the following examples:

- For $m = 1$, the policies alternate at every time step, formally expressed as $A_t = 1 - A_{t+1}$ for any $t \geq 1$;
- For a general $2 \leq m \leq T/2$, the policy remains constant for $m$ time steps and then switches, which can be mathematically represented as $A_{mt-m+1} = A_{mt-m+2} = \ldots = A_{mt} = 1 - A_{mt+1} = \ldots = 1 - A_{mt+m}$;
- For $m = T$, the same policy is applied throughout each day, i.e., $A_1 = A_2 = \ldots = A_T$.

Furthermore, the initial policy alternates across days, which can be mathematically described as $A_{i,1} = 1 - A_{i+1,1}$ for any $i$, and the initial policy on the first day $A_{1,1}$ is uniformly generated. Consequently, when $m = T$, the design essentially becomes an *alternating-day* scheme where the two policies are switched on a daily basis.

Moreover, to improve clarity and facilitate a more intuitive comparison between our SB design and the standard A/B testing, we present a detailed discussion in Appendix B.

***Objective.*** We aim to estimate the global average treatment effect, defined as the difference between the average cumulative rewards when implementing the new policy throughout each day and that when using the baseline policy,

$$\text{ATE} = \frac{1}{T}\sum_{t=1}^{T}\mathbb{E}^1(R_t) - \frac{1}{T}\sum_{t=1}^{T}\mathbb{E}^0(R_t),$$

where $\mathbb{E}^1$ and $\mathbb{E}^0$ represent the expectation when the new policy (coded as 1) and the baseline policy (coded as 0) are applied at all times, respectively. This metric is particularly relevant for A/B testing in RL (Tang et al., 2022).

In standard terms, an AD design operates under an *on-policy* framework where within each day, the behavior policy generating the experimental data aligns with the target policy under evaluation (which assigns a constant action, either 0 or 1, at each time). Conversely, when $m \neq T$, the switchback (SB) design operates under an *off-policy* framework where the behavior policy differs from the target policy.

In standard MDPs, off-policy estimators are considered less efficient than on-policy ones due to the distributional shift between the behavior and target policies (Li et al., 2023b). However, in the next section, we will demonstrate a trade-off between AD and SB. This distributional shift issue becomes predominant when the carryover effect is large, thus favouring AD in such settings. Conversely, when the carryover effect is small, the effectiveness of different designs depends crucially on the autocorrelations among reward errors. In particular, when the majority of reward errors are positively autocorrelated, SB can actually outperform an AD. These findings provide valuable insights for technology companies to optimize their A/B testing strategies.

# 4. Optimality of Switchback Designs

This section investigates the effectiveness of switchback designs when applied to various ATE estimators. We begin with a toy example to offer insight into why off-policy estimators under switchback designs can be more effective than their on-policy counterparts under alternating-day designs.

**Toy example.** To facilitate understanding, we consider the following simple model without carryover effects over time: $R_t = \beta_1 A_t + \beta_0(1 - A_t) + e_t$. By definition, the oracle ATE equals $\beta_1 - \beta_0$. A natural plug-in estimator for the ATE is given by: $\frac{\sum_{i,t} R_{i,t} A_{i,t}}{\sum_{i,t} A_{i,t}} - \frac{\sum_{i,t} R_{i,t}(1 - A_{i,t})}{\sum_{i,t}(1 - A_{i,t})}$. Upon calculation, it can be shown that its MSE under AD and a particular SB design with $m = 1$ is proportional to

$$\text{MSE(AD)} \propto \text{Var}(e_1 + e_2 + e_3 + \cdots + e_T),$$
$$\text{MSE(SB}^{(1)}) \propto \text{Var}(e_1 - e_2 + \cdots - (-1)^T e_T),$$

respectively. Based on these formulas, the ATE estimator's MSE depends solely on the correlation structure of the residuals $\{e_t\}_t$. In the AD design, the MSE is proportional to the variance of the sum of all residuals, which inflates when errors are positively correlated. In contrast, the SB design's MSE is determined by the variance of a weighted sum of residuals with alternating signs, effectively canceling positively correlated errors and improving ATE estimation accuracy. We extend this analysis to incorporate carryover effects and examine various policy value estimators, including model-based estimators, LSTD, and DRL.

## 4.1. Methods

This section is organized as follows. We first study linear-model-based estimators. We next analyze the LSTD estimator, a popular value-based estimator. Finally, we consider the DRL estimator, an advanced model-free estimator known for its double robustness.

**Model-based method.** In model-based approach, we assume a system dynamics model and utilize this model to construct the ATE estimator. In particular, we apply linear models to both the reward function and the expected value of the next state, resulting in the following set of linearity assumptions:

$$\begin{cases} r_t(A_t, S_t) = \alpha_t + S_t^\top \beta_t + \gamma_t A_t, \\ \mathbb{E}(S_{t+1}|A_t, S_t) = \phi_t + \Phi_t S_t + \Gamma_t A_t, \end{cases} \quad (1)$$

where $\alpha_t$ and $\gamma_t$ are real-valued, $\beta_t, \phi_t$, and $\Gamma_t$ are vectors in $\mathbb{R}^d$, and $\Phi_t \in \mathbb{R}^{d \times d}$.

We make two remarks. First, the model presented in (1) resembles the linear dynamic system model commonly found in linear-quadratic-Gaussian control problems (Krishnamurthy, 2016). It is also consistent with the linear MDP assumption – a condition frequently employed in the RL literature (Jin et al., 2020; Li et al., 2021; Xie et al., 2023).

Second, under Model (1), as outlined in Luo et al. (2024), the ATE can be expressed as

$$\frac{1}{T}\sum_{t=1}^T \gamma_t + \frac{1}{T}\sum_{t=2}^T \beta_t^\top \Big[ \sum_{k=1}^{t-1}(\Phi_{t-1}\Phi_{t-2}\dots\Phi_{k+1})\Gamma_k \Big], \quad (2)$$

where the product $\Phi_{t-1}\dots\Phi_{k+1}$ is treated as an identity matrix if $t - 1 < k + 1$. The first term on the right-hand side (RHS) of (2) represents the direct effect of actions on immediate rewards, while the latter term accounts for the delayed or carryover effects of previous actions.

Equation (2) motivates us to employ the ordinary least square (OLS) regression to compute the estimators $\widehat{\alpha}_t, \widehat{\beta}_t, \widehat{\gamma}_t, \widehat{\phi}_t, \widehat{\Gamma}_t$ and $\widehat{\Phi}_t$ and plug them into (2) to compute the final ATE estimator. Refer to Appendix A for details.

**LSTD.** LSTD is a popular model-free, value-based OPE estimator. To illustrate the LSTD estimator, we first introduce the notion of value function. For any time $t \geq 1$, action $a \in \{0, 1\}$, and state $s$, the value function $V_t^a(s)$ represents the expected cumulative return from time $t$ in state $s$, assuming the agent follows a constant action $a$ and can be mathematically expressed as: $V_t^a(s) = \sum_{j=t}^T \mathbb{E}^a(R_j|S_t = s)$. The ATE can then be equivalently represented by $T^{-1}\mathbb{E}[V_1^1(S_1) - V_1^0(S_1)]$. LSTD computes an estimated value function $\widehat{V}_t^{a,m}$ and approximates the expectation using empirical averages under the $m-$switchback design, leading to the following ATE estimator:

$$\frac{1}{nT}\sum_{i=1}^n [\widehat{V}_1^{1,m}(S_{i,1}) - \widehat{V}_1^{0,m}(S_{i,1})]. \quad (3)$$

We next outline the approach for estimating the value function using LSTD, which employs linear sieves (Grenander, 1981) to approximate the value function $V_t^a(s)$ by $\varphi_t^\top(s)\theta_{t,a}^*$, with a given basis function $\varphi_t$ and the associated regression coefficients $\theta_{t,a}^*$. A crucial aspect of this methodology is that these value functions follow the Bellman equation: $\mathbb{E}[R_t + V_{t+1}^a(S_{t+1}) - V_t^a(S_t)|A_t = a, S_t = s] = 0$ for every state-action pair $(s, a)$. This leads to the formulation of the following estimating equation under the $m-$switchback design:

$$\frac{1}{n}\sum_{i=1}^n \varphi_t(S_{i,t})\mathbb{I}(A_{i,t} = a)\Big[R_{i,t} \\ + \varphi_{t+1}^\top(S_{i,t+1})\widehat{\theta}_{t+1,a,m} - \varphi_t^\top(S_{i,t})\widehat{\theta}_{t,a,m}\Big] = 0. \quad (4)$$

The coefficients $\{\widehat{\theta}_{t,a,m}\}_{t,a}$ are computed in a backward manner, as detailed in Algorithm 1. With these estimators in hand, we construct the value function estimator and plug them into (3) to derive the final ATE estimator.

**DRL.** The DRL estimator extends the double machine learning estimator, originally developed for contextual bandit

---

**Algorithm 1** Estimating ATE via LSTD.

**Input:** $\{(S_{it}, R_{it}, A_{it}) : 1 \le i \le n, 1 \le t \le T\}$.
Set $\widehat{\theta}_{T+1,a,m} = 0$, for $a \in \{0, 1\}$.
**for** $t = T$ **to** 1 **do**
    Solve (4) to obtain $\widehat{\theta}_{t,a,m}$.
**end for**
**Output:** The ATE estimator (3) with the estimator $\widehat{\theta}_{1,a,m}$.

---

settings (Chernozhukov et al., 2018) to sequential decision making. This approach combines the value-based estimator with the marginal IS estimator (Liu et al., 2018) for more robust and efficient policy evaluation. A key feature of the DRL estimator is its double robustness: it remains consistent as long as either the estimated value function or the marginal IS ratio is consistent. Additionally, the DRL estimator is semiparametrically efficient, achieving the lowest MSE among the class of regular and asymptotically linear estimators (Bickel et al., 1993; Tsiatis, 2007).

To present the DRL estimator, we define an estimating function $\psi(\{S_t, A_t, R_t\}_t; \{V_t^a\}_{t,a}, \{\omega_t^{a,m}\}_{t,a})$ as follows:

$$V_1^1(S_1) - V_1^0(S_1) + \sum_{t=1}^{T} \sum_{a=0}^{1} (-1)^{a+1} \omega_t^{a,m}(A_t, S_t)$$
$$\times \left[ R_t + V_{t+1}^a(S_{t+1}) - V_t^a(S_t) \right].$$

Here, the first part corresponds to the value-based estimator. The second part serves as an augmentation term, which is mean-zero according to the Bellman equation when the value function is correctly specified. This term enhances the estimator's robustness against potential model misspecification of the value function. In particular, $\omega_t$ corresponds to the marginalized IS ratio, which is crucial for efficient OPE in MDPs (Liu et al., 2018). For any action $a \in \{0, 1\}$ and time $1 \le t \le T$, let $p_t^a$ denote the probability mass function of $(S_t, A_t)$ under consistent application of policy $a$. Additionally, let $p_t^m$ denote the probability mass function of $(S_t, A_t)$ under an AD or SB design. The marginalized IS ratio $\omega_t^{a,m}(s, a')$ is defined as $p_t^a(s, a')/p_t^m(s, a')$. It can be shown that $\psi$ is unbiased to the ATE if either $\{V_t^a\}_{t,a}$ or $\{\omega_t^{a,m}\}_{t,a}$ is correctly specified.

Notice that the value function and marginalized IS ratio can be estimated using any advanced RL algorithms. We plug their estimators into the estimating function $\psi$, and utilize sample-splitting and cross-fitting (Chernozhukov et al., 2018) to construct the final DRL estimator. The detailed estimating procedure is summarized in Algorithm 2.

### 4.2. Theoretical analysis

We begin by introducing two key notations essential for our analysis: (i) First, let $\sigma_e(t_1, t_2)$ denote the covari-

---

**Algorithm 2** Estimating ATE via DRL.

**Input:** $\{(S_{it}, R_{it}, A_{it}) : 1 \le i \le n, 1 \le t \le T\}$.
**Step 1:** Randomly divide the data trajectories into $K$ equally-sized folds $\{\mathcal{D}_k\}_{k=1}^{K}$.
**Step 2:** For $k = 1, \ldots, K$, construct estimators $\{\widehat{\omega}_{t,-k}^{a,m}\}_{t,a}$ and $\{\widehat{V}_{t,-k}^{a,m}\}_{t,a}$ using all trajectories except those in $\mathcal{D}_k$.
**Output:** The ATE estimator

$$\frac{1}{nT} \sum_{k=1}^{K} \sum_{i \in \mathcal{D}_k} \psi(\{S_{i,t}, A_{i,t}, R_{i,t}\}_t; \{\widehat{V}_{t,-k}^{a,m}\}_{t,a}, \{\widehat{\omega}_{t,-k}^{a,m}\}_{t,a}).$$

---

ance between reward residuals $e_{t_1}$ and $e_{t_2}$; (ii) Second, we define $\delta$ as the measure of the impact of the new policy on the state transition functions $\{p_t\}_t$, such that $\delta = \max_{s,t} \sum_{s'} |p_t(s'|1, s) - p_t(s'|0, s)|$.

Notice that $\delta$ inherently quantifies the size of the carryover effect since under the MDP model, the carryover effect is modeled via state transitions; refer to Figure 2. In particular, when $\delta = 0$, past actions have the same effects on state transitions, eliminating any carryover effect.

We next impose a common assumption required by all the three estimators.

**Assumption 1** (Bounded rewards)**.** *The rewards $\{R_t\}_t$ are uniformly bounded, i.e., $\max_t |R_t| \le R_{\max}$ for some $R_{\max} < \infty$ almost surely.*

Assumption 1 is frequently employed in the RL literature (see e.g., Chen & Jiang, 2019; Fan et al., 2020).

In Appendix D.1, we introduce other estimator-specific assumptions. Notably, each type of estimators only requires a subset of these assumptions. These assumptions are mild and can be easily satisfied, as discussed in Appendix D.1.

**Theorem 1.** *Under the given conditions, the difference in the MSE of the ATE estimator between the alternating-day design and an $m$-switchback design (where each switch duration equals $m$) is lower bounded by*

$$\frac{16}{nT^2} \sum_{\substack{k_2 - k_1 = 1, 3, 5, \ldots \\ 0 \le k_1 < k_2 < T/m}} \sum_{l_1, l_2 = 1}^{m} \sigma_e(l_1 + k_1 m, l_2 + k_2 m) \quad (5)$$
$$- \frac{c \delta R_{\max}^2}{n} - o\left(\frac{1}{n}\right),$$

*for some constant $c > 0$ and some reminder term of the order $o(1/n)$.*

Based on Theorem 1, it is immediate to see that the lower bound for the difference in the MSE depends on three terms: (i) an autocorrelation term, quantifying the auto-correlations

among reward errors; (ii) a carryover effect term which is proportional to $\delta$ and quantifies the magnitude of the carryover effect; (iii) a reminder term.

Here, the reminder term is a high-order, estimator-dependent term. Specifically, for the model-based estimator, it is of the order $O(n^{-3/2})$ up to some logarithmic factors, which depends on $n$ through $n^{-3/2}$ as opposed to the first two terms, which depend on $n^{-1}$. Consequently, the reminder term decays to zero at a much faster rate. For LSTD and DRL, the reminder term additionally relies on the estimation errors of value function and/or MIS ratio. Its specific order for these estimators is detailed in Appendix D.2.

As such, the first two terms are the primary drivers. They indicate that the effectiveness of switchback designs depends on two crucial factors: **the autocorrelation structure** and **the size of the carryover effect**. With a large carryover effect, different designs will induce substantially different state distributions, and the off-policy estimator under the switchback design will suffer from substantial distributional shifts when estimating the ATE. Mathematically, this effect is manifested by the second term in (5), making AD the most efficient design. This observation has been empirically verified in our numerical studies.

Conversely, with a weak carryover effect where $\delta$ is sufficiently small, the first term becomes the leading term, and the effectiveness of different designs is primarily determined by the autocorrelation structure of reward errors. By the definition of $\sigma_e$, it is evident that:

- When the majority of reward errors are positively correlated (as demonstrated in our real-data applications depicted in Figure 1), the first term in (5) is strictly positive. This implies that SB is more efficient than AD. Additionally, when the covariance function is stationary and satisfies $\sigma_e(t_1, t_2) = \sigma_e^*(|t_1 - t_2|)$ for some $\sigma_e^*(\bullet)$ being a monotonically decreasing function, the second line becomes a monotonically decreasing function of $m$; see e.g., Corollary 1-3 below. This formally verifies that increasing the frequency of policy switches (reducing the value of $m$) can enhance the efficiency of the switchback design.

- With uncorrelated errors, the first term in (5) becomes zero, and all designs become asymptotically equivalent.

- When the majority of errors are negatively correlated, AD becomes the most efficient.

To the best of our knowledge, the aforementioned findings have not been systematically established in the RL literature. While existing works have studied switchback designs, they often focus on simple and specific policy value estimators.

Next, we investigate three commonly used covariance structures – autoregressive, moving average and exchangeable to further elaborate Theorem 1.

**Corollary 1** (Autoregressive). *Let $\sigma_e(t_1, t_2) = \sigma^2 \rho^{|t_1-t_2|}$ for some $-1 < \rho < 1$ and $\sigma^2 > 0$. For sufficiently large $T$, the first term in* (5) *becomes asymptotically equivalently to*

$$\frac{16\sigma^2\rho(1-\rho^m)}{mT(1-\rho)^2(1+\rho^m)},$$

*which is a strictly decreasing function of $m$ when $\rho > 0$*

**Corollary 2** (Moving average). *Let $e_t = K^{-\frac{1}{2}} \sum_{k=1}^{K} \varepsilon_{t+k}$ for a white noise process $\{\varepsilon_t\}_t$ with $Var(\varepsilon_t) = \sigma^2 > 0$. For any $m \geq K$ that divides $T$, the first term in* (5) *becomes*

$$\frac{8\sigma^2(T/m - 1)(K^2 - 1)}{3T^2},$$

*which is a strictly decreasing function of $m$.*

**Corollary 3** (Exchangeable). *Assume $\sigma_e(t_1, t_2) = \sigma^2[\rho\mathbb{I}(t_1 \neq t_2) + \mathbb{I}(t_1 = t_2)]$ for some $-1 < \rho < 1$ and $\sigma^2 > 0$. Then the first term in* (5) *equals*

$$\begin{cases} 4\sigma^2\rho, & \text{if } T/m \text{ is even,} \\ 4\sigma^2\rho(1 - m^2/T^2), & \text{if } T/m \text{ is odd,} \end{cases}$$

*which is a constant function of $m$ when $T/m$ is even, and varies strictly monotonically (increasing or decreasing) as a function of $m$ when $T/m$ is odd, depending on whether $\rho > 0$ or $\rho < 0$.*

We remark that while these structures may appear simple, they are widely adopted in practice (Williams, 1952; Berenblut & Webb, 1974; Zeger, 1988).

## 5. Numerical Experiments

In this section, we conduct numerical experiments to verify our theory. Our code is available at https://github.com/QianglinSIMON/SwitchMDP.

### 5.1. Synthetic Environments

*DGP*. We design two data generating processes (DGPs) with a common time horizon $T = 48$ and state dimension $d = 3$: one with a linear DGP and the other with a nonlinear DGP (refer to Appendix C for the detailed setup), to evaluate the performance of various switchback designs and different ATE estimators. The reward errors follow an autoregressive covariance function so that $\text{Cov}(e_{t_1}, e_{t_2}) = 1.5\rho^{|t_1-t_2|}$ whenever $t_1 \neq t_2$, with the parameter $\rho$ varied among the set $\{0.3, 0.5, 0.7, 0.9\}$. We also vary the size of carryover effects, characterized by a parameter $\delta$. The number of days $n$ used in our simulations is selected from a range of 16 to 52 in increments of four.

*Results*. We implement various $m$-switchback designs with $m \in \{1, 3, 6, 12, 24, 48\}$ in these environments and report

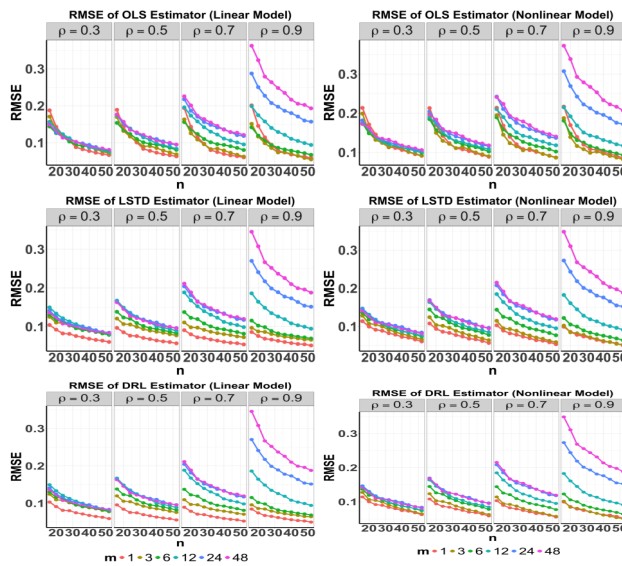

*Figure 3.* Simulation results with different combinations of $(n, m, \rho)$ alongside different estimation procedures. $\delta$ is fixed to zero, resulting in a weak yet nonzero carryover effect.

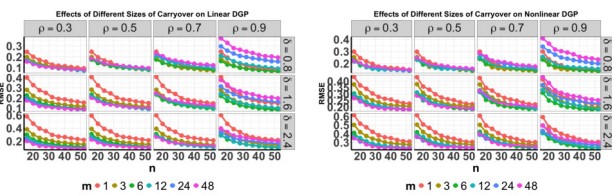

*Figure 4.* RMSEs of OLS estimator with different combinations of $(n, m, \rho, \delta)$. $\delta$ denotes the constant shift in generating $\Gamma_t$, i.e. $\{\Gamma_t^{(j)}\}_{t,j} \overset{i.i.d.}{\sim} N(\delta, 0.3^2)$. A larger $\delta$ in absolute value leads to a more pronounced carryover effect.

the root MSEs of the resulting OLS, LSTD, and DRL estimators (refer to Appendix A for their implementation details) aggregated over 200 simulations in Figures 3 and 4, considering different combinations of $n, \rho, \delta$ and the estimating procedure. Notably, when $m = 48$, the resulting design coincides with AD. Figure 3 report the results with weak carryover effects. It can be seen from Figure 3 that the root MSE (RMSE) decays with $m$ in most cases. Additionally, the difference in RMSE between the SB design and the AD design grows with $\rho$, which corresponds to the autocorrelation coefficient of $\{e_t\}_t$. This aligns with our analysis, suggesting that a higher degree of positive correlation in the residuals favors SB over AD. Meanwhile, it can be seen from Figure 4 that as the carryover effects increase (i.e., as $\delta$ becomes larger), the AD becomes progressively more efficient. These results empirically validate our theories.

***Sensitivity of covariance structure***. We further examine four additional covariance structures: (i) moving average, (ii) exchangeable (with a positive correlation), (iii) uncorrelated and (iv) autoregressive (with a negative autocorrela-

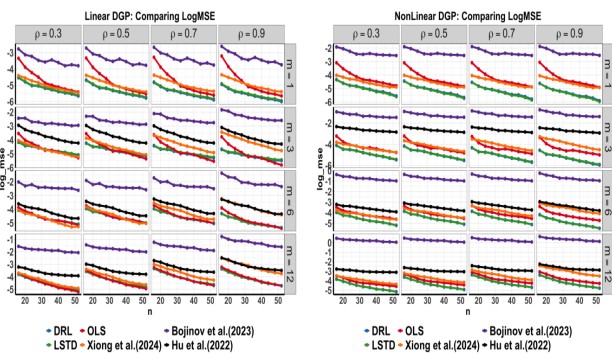

*Figure 5.* Comparing log MSEs of all ATE estimators with different combinations of $(n, m, \rho)$ under the linear DGP (left) and nonlinear DGP (right).

tion). We focus on OLS estimators and report their MSEs under different designs in Figures S1 and S2 of Appendix C. The efficiency of switchback designs varies with $m$, depending on the presence of negative or positive correlation. In cases of uncorrelated errors, most designs attain similar performance and AD works the best in small samples.

***Comparison***. We further conduct simulations to compare the RL-based estimator with three other baseline estimators: (i) the sequential IS estimator (Bojinov et al., 2023) which addresses carryover effects via multi-step importance sampling; (ii) the difference-in-mean estimator of Hu & Wager (2022) which uses burn-in to mitigate carryover effects during policy transitions; (iii) the simple IS estimator of Xiong et al. (2024) which does not account for carryover effects. Results are reported in Figure 5, where RL-based estimators consistently outperform (i) and (ii), and both DRL and LSTD significantly outperform (iii) in most settings.

### 5.2. Real-data-based Simulation

We use two real datasets from a leading ridesharing company, each with 40 days of data ($N = 40$). The state variable includes the number of order requests and the driver's total online time in each interval, measuring the demand and supply dynamics that impact the platform's outcomes (Shi et al., 2023b). The reward is the total income earned by the drivers within each time interval, whose residual shows a noticeable positive correlation, as depicted in Figure 1. Both datasets are collected from A/A experiments, in which a single order dispatch policy was consistently applied over time ($A_t = 0$ for all $t$). To utilize these datasets for evaluating the performance of different designs, we create two simulation environments using the wild bootstrap (Wu et al., 1986), as detailed in Appendix C. To generate data under different policies, we introduce an effect size parameter $\lambda$ and consider four choices, corresponding to 0 (i.e., no treatment effect at all), 2%, 5%, 7.5%, 10%, 12% and 15%. By adjusting $\lambda$, the data are generated so that both the direct effect and carryover effect of the new policy (see Equation

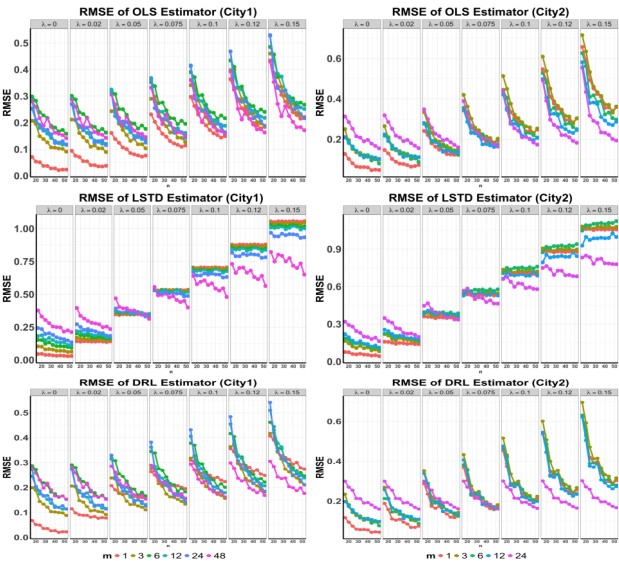

Figure 6. Results from real-data-based simulation.

(2)) are increase by $\frac{\lambda}{2}$, leading to an overall ATE increase of $\lambda$.

Figures 6 summarize the results, which strongly support our theoretical findings. Specifically, with positive correlated reward residuals, the benefits of employing switchback designs with more frequent switches are evident when there is no or only a minor effect enhancement (i.e., 0 or 2% increase). However, as the size of the carryover increases to 15%, the AD becomes more efficient.

To better understand our estimators, we conducted additional simulations based on the City II dataset, employed a non-parametric bootstrap method to construct confidence intervals (CIs), and reported both the coverage probability (CP) and the average CI width in the Figure S3 of Appendix. Most CPs exceed 92%, which is close to the nominal level. For small values of $\lambda$, more frequent policy switching reduces the average CI width. As $\lambda$ increases to 10%, AD yields the narrowest CI on average. These results support our claim that a lower MSE corresponds to shorter CIs.

## 6. Discussion

This paper studies switchback designs under an RL framework. To offer practical guidance, we outline a workflow in this section (see also Figure 7):

(i) The first step is to discretize the data to define appropriate time intervals, ensuring that both the state and reward follow the Markov assumption (see Section 3). This assumption can be tested via existing state-of-the-art methods (see e.g., Chen & Hong, 2012; Shi et al., 2020; Zhou et al., 2023). When the Markov assumption is violated, it is necessary to increase the length of time intervals accordingly to satisfy this condition.

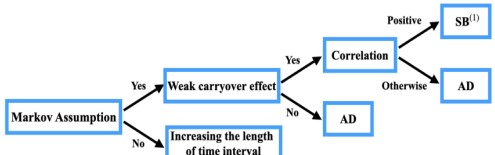

Figure 7. The proposed workflow guideline.

(ii) The second step is to assess the magnitude of the carryover effect. Should the effect be strong, AD is recommended. In our numerical studies, we observe that with a large carryover effect, the ATE estimator under SB suffers from much a larger bias than that under AD.

(iii) Finally, if the carryover effect is weak, we proceed to analyze error correlations. When errors exhibit positive correlations, we recommend to employ the switchback design with $m = 1$. In cases where errors are uncorrelated or negatively correlated, AD would be preferred.

We also remark that in our motivating ridesharing example, there are three different types of experiments: (i) temporal randomization (over time), (ii) spatial randomization (across geographic areas), and (iii) user-level randomization (across drivers/passengers). Our primary focus is (i), which applies to the evaluation of order dispatch policies that must be implemented city-wide, making (ii) and (iii) unsuitable. Spatial randomization (ii) is typically used for testing localized subsidy policies in different regions, while user-level randomization (iii) applies when assigning personalized subsidies to individual users.

When restricting to temporal randomization, the population corresponds to the entire time horizon, with each time interval representing an individual unit. When no carryover effects exist and residuals are uncorrelated (satisfying i.i.d. assumptions), similar to Theorem 1, we can show that standard uniform randomization over time is equivalent to both AD and SB. This is because temporal ordering becomes irrelevant under the uncorrelatedness assumption. Similarly, when randomizing is conducted at the daily level rather than per time unit, standard A/B testing procedures that ignore carryover effects are equivalent to AD designs.

Finally, confidence intervals (CIs) and $p$-values are equally important metrics, as A/B testing is fundamentally a statistical inference problem. Our experimental designs are tailored to minimize the MSE of the resulting ATE estimators. A closer examination of the proof of Theorem 1 reveals that the three RL-based estimators are asymptotically normal, with their MSEs primarily driven by their asymptotic variances. Consequently, optimal designs that minimize the variance of the ATE estimator also reduce CI length and enhance the power of the corresponding hypothesis test. We empirically validate this claim using real-data-based simulation in Figure S3.

## Acknowledgement

We thank the anonymous referees and the meta reviewer for their constructive comments, which have led to a significant improvement of the earlier version of this article. Shi's research is partly supported by an EPSRC grant EP/W014971/1. Yang's research is partly supported by the National Natural Science Foundation of China (No. 72301276). Wen and Tang's research is partly supported by the National Key R&D Program of China (No. 2022YFA1003701).

## Impact Statement

This paper investigates switchback experiments in modern digital platforms, focusing on the key factors that determine their efficiency. The study finds that the effectiveness of switchback designs is primarily influenced by the autocorrelation structure and the magnitude of carryover effects, with an inherent trade-off that applies to most state-of-the-art RL estimators. Additionally, a practical workflow is proposed to improve the methodological approaches for both researchers and practitioners. However, it is important to note that switchback designs are not universally applicable and may be unsuitable in the following cases: (i) lack of temporal dependence, (ii) significant carryover effects, (iii) complex interactions across users or regions, (iv) high switching costs, and (v) rapidly changing systems. Researchers should carefully evaluate these limitations and apply switchback designs with caution.

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

# A. Implementation Details

In this section, we first detail the parametric estimation of the model-based method using the OLS approach, as described in Section 4.1. Next, we introduce a modified LSTD estimator which is designed to improve the efficiency of the original LSTD estimator and is implemented in our numerical study. Finally, we detail our implementation of the DRL estimator, focusing on the estimation of the IS ratio and the value function.

***OLS Estimation.*** Given the observational data $\{(S_{i,t}, A_{i,t}, R_{i,t}) : 1 \leq t \leq T\}_{i=1}^n$, for each $1 \leq t \leq T$, we deploy the OLS regression to the dataset $\{(S_{i,t}, A_{i,t}, R_{i,t}) : 1 \leq i \leq n\}$ with $R_{i,t}$ as the response and $(1, S_{i,t}, A_{i,t})$ as the predictor to compute the estimators $\widehat{\alpha}_t$, $\widehat{\beta}_t$ and $\widehat{\gamma}_t$. Similarly, we apply OLS to $\{(S_{i,t}, A_{i,t}, S_{i,t+1}^{(j)}) : 1 \leq i \leq n\}$ with the $j-$th component of $S_{i,t+1}$, $S_{i,t+1}^{(j)}$ as the responses and $(1, S_{i,t}, A_{i,t})$ as the predictor to estimate the $j$th element of $\phi_t$, the $j$th element of $\Gamma_t$, as well as the $j$th row of $\Phi_t$. Concatenating all the estimators across $j$ produces $\widehat{\phi}_t$, $\widehat{\Gamma}_t$ and $\widehat{\Phi}_t$. With these estimators in hand, we plug them into (2) to compute the final estimators $\text{ATE}_{\text{AD}}$ and $\text{ATE}_{\text{SB}}^{(m)}$.

***Modified LSTD.*** In the original LSTD algorithm described in Section 4.1, the value function at a specific time $t$ is estimated using only the data subset corresponding to that particular time. This approach might be inefficient when the system dynamics remain relatively consistent over time. To enhance estimation efficiency, we incorporate the time index into the state, resulting in an augmented state, denoted as $\widetilde{S}_t$ for each time $t$. We then approximate the value function using a linear combination of sieves. It is important to note that the basis functions $\varphi$ contains not only the bases for the original state but also those for the time component, addressing potential nonstationarity.

To lay down the foundation, for any $t$, $a$ and $\tilde{s}$, we first define a value function $V_{t:t}^a(\tilde{s}) = \mathbb{E}^a(R_t|\widetilde{S}_t = \tilde{s}, A_t = a)$. Next, we recursively define $V_{t-j:t}^a(\tilde{s}) = \mathbb{E}^a[R_{t-j} + V_{t-j+1:t}^a(\widetilde{S}_{t-j+1})|\widetilde{S}_t = \tilde{s}, A_t = a]$ for $j = 1, 2, \ldots, t-1$. Essentially, for any $t_1 \leq t_2$, $V_{t_1:t_2}^a$ represents the expected cumulative reward from time $t_1$ to $t_2$ starting from a given state at time $t_1$. Additionally, the final ATE estimator can be represented by $\mathbb{E}[V_{1:T}^1(\widetilde{S}_1) - V_{1:T}^0(\widetilde{S}_1)]$.

It remains to estimate these doubly indexed value functions. A key observation is that, with the time index included in the state to account for nonstationarity, $V_{t_1:t_2}^a$ shall equal $V_{t_3:t_4}^a$ provided the time gaps $t_2 - t_1$ and $t_4 - t_3$ are equal. This allows us to aggregate all data over time to simultaneously estimate all value functions.

Under the $m-$switchback design, our first step is to estimate $\{V_{t:t}^a\}_t$ by solving the following estimating equation:

$$\sum_{i,t}[R_{i,t} - \varphi^\top(\widetilde{S}_{i,t})\widehat{\theta}_{0,a,m}]\varphi(\widetilde{S}_{i,t})\mathbb{I}(A_{i,t} = a) = 0. \tag{6}$$

From this, we compute $\widehat{V}_{t:t}^{a,m}(\tilde{s})$ as $\varphi^\top(\tilde{s})\widehat{\theta}_{0,a,m}$. Next, we sequentially compute $\{V_{t-1:t}^a\}_t, \{V_{t-2:t}^a\}_t$, and so forth. Specifically, for each $j = 1, \ldots, T-1$, we recursively solve the following estimating equation,

$$\sum_{i,t}[R_{i,t-j} + \varphi^\top(\widetilde{S}_{i,t-j+1})\widehat{\theta}_{j-1,a,m} - \varphi^\top(\widetilde{S}_{i,t-j})\widehat{\theta}_{j,a,m}]\varphi(\widetilde{S}_{i,t})\mathbb{I}(A_{i,t} = a) = 0. \tag{7}$$

This leads to the derivation of $\widehat{\theta}_{j,a,m}$, based on which we set $\widehat{V}_{t-j:t}^{a,m}(\tilde{s})$ to $\varphi^\top(\tilde{s})\widehat{\theta}_{j,a,m}$. Finally, we set the ATE estimator to

$$\frac{1}{n}\sum_{i=1}^n \varphi^\top(\widetilde{S}_{i,1})(\widehat{\theta}_{T-1,1,m} - \widehat{\theta}_{T-1,0,m}). \tag{8}$$

We provide a pseudocode in Algorithm 3 to summarize the aforementioned procedure.

---

**Algorithm 3** Estimating ATE via the modified LSTD.

---

**Input:** $\{(S_{it}, R_{it}, A_{it}) : 1 \leq i \leq n, 1 \leq t \leq T\}$.
**Output:** The ATE estimator. Solve (6) to obtain $\widehat{\theta}_{T,a,m}$.
**for** $t = 1$ **to** $T - 1$ **do**
    Solve (7) to obtain $\widehat{\theta}_{T-t,a,m}$.
**end for**
**Return:** The ATE estimator constructed according to (8).

---

---

**Algorithm 4** Model-based estimation of the IS ratio.

---

**Input:** $\{(S_{it}, R_{it}, A_{it}) : 1 \leq i \leq n, 1 \leq t \leq T\}$.

**Output:** The marginal IS ratio estimator.

**Initialization:** Calculate the least square estimators $\left\{\widehat{\phi}_t\right\}_{t=1}^{T-1}$, $\left\{\widehat{\Phi}_t\right\}_{t=1}^{T-1}$ and $\left\{\widehat{\Gamma}_t\right\}_{t=1}^{T-1}$ in (1). Impose models for $S_1$ and $\{E_k\}_k$, and estimate the associated model parameters.

**for** $t = 1$ **to** $T - 1$ **do**

    **1.** Derive the probability density/mass function of $S_t$ using the aforementioned estimators and (9).

    **2.** Estimate the numerator $p_t^a$ by replacing $\{A_k\}_k$ in Equation (9) with the target policy $a$.

    **3.** Estimating the denominator $p_{a,t}^m$ replacing $\{A_k\}_k$ in Equation (9) with the treatment sequence under the $m$-switchback design given that $A_t = a$.

    **4.** Calculate the ratio.

**end for**

---

***Estimation of the IS ratios***. We have devised a model-based approach to estimate the marginal IS ratio based on the linear model assumption presented in Equation (1). It is worth noting that both the numerator and the denominator of the ratio correspond to the marginal probability density/mass function of the state at a given time, given a sequence of past treatments it has received. As a result, we can focus on estimating these marginal density/mass functions to construct the ratio estimator.

Within the framework of the linear model assumption, we can express the state at time $t$, denoted as $S_t$, as follows:

$$S_t = \sum_{k=1}^{t-1} \left(\Pi_{l=k+1}^{t-1}\Phi_l\right)\phi_k + \left(\Pi_{l=1}^{t-1}\Phi_l\right)S_1 + \sum_{k=1}^{t-1}\left(\Pi_{l=k+1}^{t-1}\Phi_l\right)\Gamma_k A_k + \sum_{k=1}^{t-1}\left(\Pi_{l=k+1}^{t-1}\Phi_l\right)E_k. \tag{9}$$

Where $\Pi_{l=k+1}^{t-1}\Phi_l := \Phi_{k+1}\cdots\Phi_{t-1}$. Consequently, given a sequence of treatments, we can replace $\{A_k\}_k$ with this treatment sequence to derive the distribution function of $S_t$. To estimate this distribution function, we follow these steps:

1. Estimate the model parameters in Equation (1).

2. Impose models for the initial state and the residuals $\{E_k\}_k$. In our implementation, we utilize normal distributions, and estimate the mean and covariance matrix parameters within these distributions using the available data. According to (9), this ensures that each $S_t$ follows a normal distribution as well.

3. Plug the estimated parameters obtained in the first two steps into (9) to construct the mean and covariance matrix estimators for $S_t$.

4. Return the normal distribution function with the estimated mean and covariance matrix estimators obtained in Step 3.

A pseudocode summarizing our procedure is presented in Algorithm 4.

***Estimation of the value function***. We devise a model-based approach to estimate the value function. Given the linear models outlined in (1), we can readily express the value function as follows:

$$\begin{aligned} V_t^a(s) &= \sum_{j=t}^{T}\mathbb{E}^a(R_j|S_t = s) = \sum_{j=t}^{T}[\alpha_j + \gamma_j a + \beta_j^\top\mathbb{E}^a(S_j|S_t = s)] \\ &= \sum_{j=t}^{T}\left\{\alpha_j + \gamma_j a + \sum_{j=t}^{T}\beta_j^\top\left[\sum_{k=t}^{j-1}(\Pi_{l=k+1}^{j-1}\Phi_l)\phi_k + (\Pi_{l=t}^{j-1}\Phi_l)s + \sum_{k=t}^{j-1}(\Pi_{l=k+1}^{j-1}\Phi_l)\Gamma_k a\right]\right\}. \end{aligned} \tag{10}$$

This leads us to the approach of initially applying OLS to estimate the model parameters and subsequently incorporating these estimates into (10) to formulate the value function estimators.

## B. Additional Discussions

Our analysis is built upon the Markov assumption (MA), which is fundamental to most RL-based estimators. In collaboration with our ride-sharing industry partner, we have observed that intervals of 30 minutes or 1 hour typically satisfy MA, showing strong lag-1 correlations with rapidly decaying higher-order correlations. This justifies the use of RL in our application.

When applied to more general applications, we recommend to properly select the interval length to meet MA as an initial step in the design of experiments (see Fig. 7). If that's challenging, we further propose three approaches below, tailored to different degrees of violation of MA. Our current results directly extend to the first two cases, while the third case requires further investigation:

- **Mild violation:** Future observations depend on the current observation-action pair and a few past observations. This mild violation can be easily addressed by redefining the state to include recent past observations. With this modified state, MA is satisfied. Our RL-based estimators and theoretical results remain valid.

- **Moderate violation:** Future observations depend on a few past observation-action pairs. Here, the RL-based estimators remain applicable if the state includes these historical state-action pairs. However, our theoretical results on optimal designs must be adjusted. Preliminary analyses show that, under weak carryover effects and positively correlated residuals, the optimal switching interval extends to 1+k (where k is the number of included past actions) rather than switching at every time step. This is because each observed reward is affected by a k+1 consecutive actions, not just the most recent action. More frequent switching under these conditions causes considerable distributional shift, inflating the variance of the ATE estimator.

- **Severe violation:** Data follows a POMDP. Although the existing literature provides doubly robust estimators and AD-like optimal designs (Li et al., 2023b) to handle such non-Markov MDPs, these estimators suffer from the "curse of horizon" (Liu et al., 2018). Recent advances propose more efficient POMDP-based estimators (Liang & Recht, 2025) and designs (Sun et al., 2024); however, these proposals are limited to linear models. Extending these methodologies to accommodate more general estimation procedures (e.g., Uehara et al. (2023)) represents an important direction for future research.

We also remark that in the first two cases, existing tests are available for testing the Markov assumption and for order selection (Chen & Hong, 2012; Shi et al., 2020; Zhou et al., 2023).

## C. Additional Experimental Results

In this section, we systematically present the details of our numerical experiments and provide all relevant figures and results discussed in the preceding sections.

*DGP (Continued).* The initial state for each day is drawn from a 3-dimensional multivariate normal distribution with zero mean and an identity covariance matrix. The coefficients in Linear DGP are specified as : $\{\gamma_t\}_t \overset{i.i.d.}{\sim} U[0.5, 0.8]$, $\{\Gamma_t^{(j)}\}_{t,j} \overset{i.i.d.}{\sim} N(0, 0.3^2)$, $\{\Phi_t^{(j_1,j_2)}\}_{t,j_1,j_2} \overset{i.i.d.}{\sim} U[-0.3, 0.3]$ and

$$\{\alpha_t\}_t \overset{i.i.d.}{\sim} \begin{cases} U[-1, -0.5] & \text{with probability } 0.5 \\ U[0.5, 1] & \text{with probability } 0.5 \end{cases}, \{\beta_t^{(j)}\}_{t,j} \overset{i.i.d.}{\sim} \begin{cases} U[-0.3, -0.1] & \text{with probability } 0.5 \\ U[0.1, 0.3] & \text{with probability } 0.5 \end{cases},$$

$$\{\phi_t^{(j)}\}_{t,j} \overset{i.i.d.}{\sim} \begin{cases} U[-1, -0.5] & \text{with probability } 0.5 \\ U[0.5, 1] & \text{with probability } 0.5 \end{cases}.$$

Here, the superscript $j$ denotes the $j$th component of each vector, while $(j_1, j_2)$ indicates the element in the $j_1$th row and $j_2$th column of each matrix. Both the reward error $e_t$ and the residual in the state regression model $E_t = S_{t+1} - \mathbb{E}(S_{t+1}|A_t, S_t)$ are set to mean zero Gaussian noises. The sequence $\{E_t\}_t$ is set to an i.i.d. multivariate Gaussian error process, with a covariance matrix 1.5 times the identity matrix, and it is independent of $\{e_t\}_t$.

In Nonlinear DGP, we consider the nonlinear reward function: $r_t(a, s) = \alpha_t + 2\beta_t^\top [\sin(sa) + \cos(s)]^2 + 3(\beta_t^\top s)\gamma_t a + [a\gamma_t + \cos(a\gamma_t)]^2$, where the sine, cosine, and square functions are applied element-wise to each component of the vector. The state regression function remains linear and identical to the one presented in (1). All model parameters, including $\{\alpha_t\}_t$, $\{\beta_t\}_t$, $\{\gamma_t\}_t$, $\{\Gamma_t\}_t$, $\{\phi_t\}_t$, $n$ and $T$, are the same as those in the above Linear DGP, with the exception of $\Phi_t^{(j_1,j_2)} \overset{i.i.d.}{\sim} U[-0.6, 0.6]$ for $j_1, j_2 = 1, 2, 3$.

*Comparison (Continued).* We compare our ATE estimators with three non-RL alternatives under the regular Bernoulli switchback design (see Definition 1). The first is proposed by Bojinov et al. (2023). While it similarly relies on a hyperparameter $m$ that determines the duration of each policy application, their design differs in that, after applying a

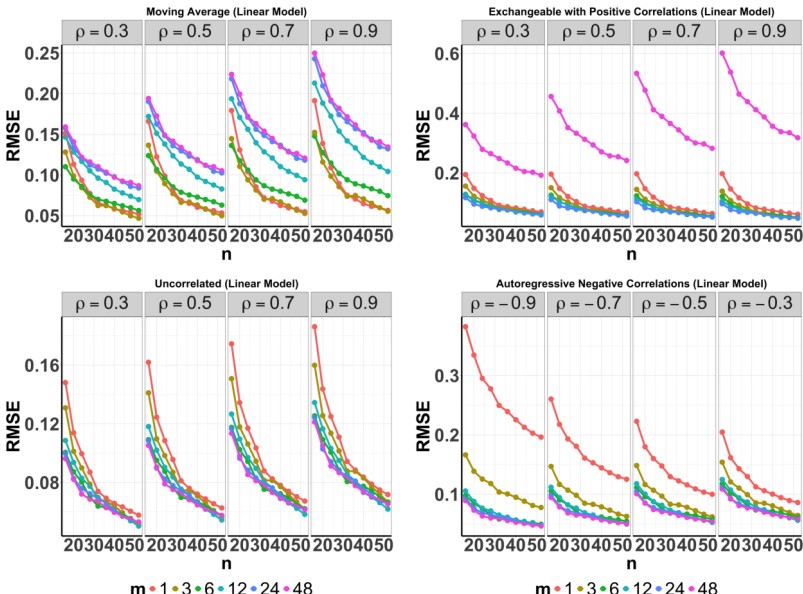

*Figure S1.* Numerical results for the linear DGP: RMSEs of OLS estimator with different combinations of $(n, m, \rho)$ and four covariance structures: moving average (top left), exchangeable with positive correlation (top right), uncorrelated (bottom left), and autoregressive with negative autocorrelation (bottom right).

treatment for $m$ time steps, there is a 50% chance of continuing with the same treatment or switching to the alternative. They employ multi-step importance sampling to construct their ATE estimator and show that the optimal choice of $m$ depends on the duration of the carryover effect, which is infinite in the MDP setting.

The second is the "burn-in" difference-in-mean estimator of Hu & Wager (2022), and the third is the simple importance sampling (IS) estimator of Xiong et al. (2024).

We begin by introducing the regular Bernoulli switchback design (Hu & Wager, 2022).

**Definition 1.** *Given a time horizon $T$ and a block length $m \geq 1$ such that $K := T/m$ is a positive integer, the regular switchback design assigns treatment sequentially as follows:*

$$A_t \mid A_{t-1} = \begin{cases} Bernoulli(0.5) & \textit{if } t = km + 1 \textit{ for some } k = 0, 1, \ldots, K - 1, \\ A_{t-1} & \textit{otherwise.} \end{cases}$$

We next present the three non-RL estimators under our notation. Under the $m$-carryover assumption, the following multi-step IS estimator is consistent for the ATE:

$$
\begin{aligned}
\widehat{\text{ATE}}^{(m)} = \frac{1}{nT} \sum_{i=1}^{n} \Bigg\{ & \sum_{t=m+1}^{T} \left[ \frac{R_{i,t}\mathbb{I}(A_{i,t-m:t} = \mathbf{1}_{m+1})}{\mathbb{P}(A_{i,t-m:t} = \mathbf{1}_{m+1})} - \frac{R_{i,t}\mathbb{I}(A_{i,t-m:t} = \mathbf{0}_{m+1})}{\mathbb{P}(A_{i,t-m:t} = \mathbf{0}_{m+1})} \right] \\
& + \sum_{t=1}^{m} \left[ \frac{R_{i,t}\mathbb{I}(A_{i,1:t} = \mathbf{1}_t)}{\mathbb{P}(A_{i,1:t} = \mathbf{1}_t)} - \frac{R_{i,t}\mathbb{I}(A_{i,1:t} = \mathbf{0}_t)}{\mathbb{P}(A_{i,1:t} = \mathbf{0}_t)} \right] \Bigg\},
\end{aligned}
\tag{11}
$$

where $A_{t_1:t_2} = (A_{t_1}, A_{t_1+1}, \ldots, A_{t_2})^{\top}$ for $1 \leq t_1 < t_2 \leq T$. As shown in Bojinov et al. (2023), the optimal block length equals $m$ under this design.

We also include the following two estimators: the "burn-in" difference-in-mean estimator (with burn-in length $0 \leq b < m$)

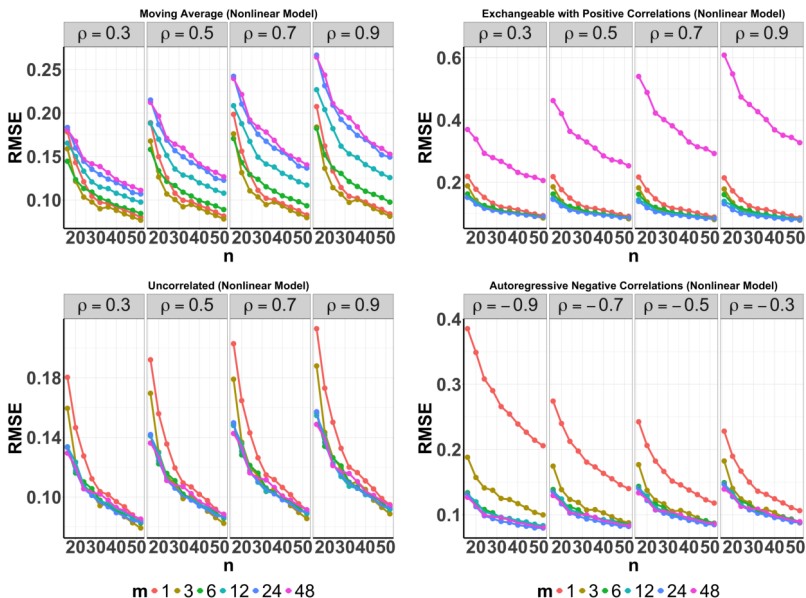

*Figure S2.* Numerical results for the nonlinear DGP: RMSEs of OLS estimator with different combinations of $(n, m, \rho)$ and four covariance structures: moving average (top left), exchangeable with positive correlation (top right), uncorrelated (bottom left), and autoregressive with negative autocorrelation (bottom right).

and the simple IS estimator:

$$\widehat{\text{ATE}}^{m,b} = \frac{1}{n}\sum_{i=1}^{n}\left[\frac{1}{K_1(m-b)}\sum_{k:A_{km+1}=1}\sum_{s=b}^{m}R_{i,km+s} - \frac{1}{K_0(m-b)}\sum_{k:A_{km+1}=0}\sum_{s=b}^{m}R_{i,km+s}\right],$$

$$\widehat{\text{ATE}}^{m,\text{IS}} = \frac{4}{nT}\sum_{i=1}^{n}\sum_{t=1}^{T}\left(A_{it}-\frac{1}{2}\right)R_{it},$$

where $K_a = |\{k \in \{0,\ldots,K-1\} : A_{km+1} = a\}|$ for $a \in \{0,1\}$.

We adopt the same experimental setup as in the linear/nonlinear DGP setting (see Subsection 5.1), using a burn-in length $b = 1$, and report the log(MSE) values of various ATE estimators under different designs and choices of $m$ in Figure 5. The results clearly demonstrate the superior performance of our ATE estimators over those based on the alternative designs.

***Real Data-based Simulation (Continued).*** The first dataset covers the period from Dec. 5th, 2018, to Jan. 13th, 2019, with thirty minutes defined as one time unit, resulting in $T = 48$. The second dataset spans from May 17th, 2019, to June 25th, 2019, with one-hour time units, leading to $T = 24$. A summary of the bootstrap-assisted procedure is provided in Algorithm 5. Specifically, for each dataset, we first fit the data based on the linear models in (1). We apply ridge regression to estimate the regression coefficients with the regularization parameter determined by minimizing the generalized cross-validation criterion (Wahba, 1975). This yields the estimators $\{\widehat{\alpha}_t\}_t$, $\{\widehat{\beta}_t\}_t$, $\{\widehat{\phi}_t\}_t$ and $\{\widehat{\Phi}_t\}_t$. However, $\{\gamma_t\}_t$ and $\{\Gamma_t\}_t$ remain unidentifiable, since $A_t = 0$ almost surely. We then calculate the residuals in the reward and state regression models based on these estimators as follows:

$$\widehat{e}_{i,t} = R_{i,t} - \widehat{\alpha}_t - S_{i,t}^{\top}\widehat{\beta}_t, \quad \widehat{E}_{i,t} = S_{i,t+1} - \widehat{\phi}_t - \widehat{\Phi}_t S_{i,t}. \tag{12}$$

To generate simulation data with varying sizes of treatment effect, we introduce the treatment effect ratio parameter $\lambda$ and manually set the treatment effect parameters $\widehat{\gamma}_t = \delta_1 \times (\sum_i R_{i,t}/(100 \times N))$ and $\widehat{\Gamma}_t = \delta_2 \times (\sum_i S_{i,t}/(100 \times N))$. The treatment effect ratio essentially corresponds to the ratio of the ATE and the baseline policy's average return. A discussion on the choice of the treatment effect ratio can be found in subsection 5.2.

Finally, to create a dataset spanning $n$ days, actions are generated according to the chosen design as described in Section 3. We then sample i.i.d. standard Gaussian noises $\{\xi_i\}_{i=1}^{n}$. For the $i$-th day, we uniformly sample an integer $I \in \{1,\ldots,N\}$,

---

**Algorithm 5** Bootstrap-based Simulation

---

1: **Input:** Real data $\{(S_{it}, R_{it}) : 1 \leq i \leq n; 1 \leq t \leq T\}$, the adjustment parameters for the ratios $(\delta_1, \delta_2)$, $m$-switchback design, the bootstraped sample size $n$, random seed, the bootstrapped size $B$.

2: **Output:** The RSMEs, Biases, and SDs of different ATE estimators.

3: **Step 1:** Calculate the least square estimates $\{\widehat{\alpha}_t\}$, $\{\widehat{\beta}_t\}$, $\{\widehat{\phi}_t\}$, $\{\widehat{\Phi}_t\}$ in the model (1), treatment effect parameters $\{\widehat{\gamma}_t\}$, $\{\widehat{\Gamma}_t\}$, and the residuals of the reward model and state regression model by Equation (12).

4: **for** $b = 1$ **to** $B$ **do**

5:     Sample the number of days $n$ from $\{1, \ldots, N\}$ with replacement and generate i.i.d. normal random variables $\xi_i^b \sim N(0, 1)$.

6:     Generate pseudo rewards $\left\{\widehat{R}_{i,t}^b\right\}_{i,t}$ and states $\left\{\widehat{S}_{i,t}^b\right\}_{i,t}$ using:

$$\widehat{R}_{i,t}^b = [1, (\widehat{S}_{i,t}^b)^\top, A_{i,t}] \begin{pmatrix} \widehat{\alpha}_t \\ \widehat{\beta}_t \\ \widehat{\gamma}_t \end{pmatrix} + \xi_i^b \widehat{e}_{i,t}, \quad \widehat{S}_{i,t+1}^b = [\widehat{\phi}_t, \widehat{\Phi}_t, \widehat{\Gamma}_t] \begin{pmatrix} 1 \\ \widehat{S}_{i,t}^b \\ A_{i,t} \end{pmatrix} + \xi_i^b \widehat{E}_{i,t}.$$

7:     Calculate the set of estimators $\left\{\text{ATE}_{\text{SB}}^{(m),b}\right\}_{b,m}$ by OLS, LSTD, and DRL.

8: **end for**

---

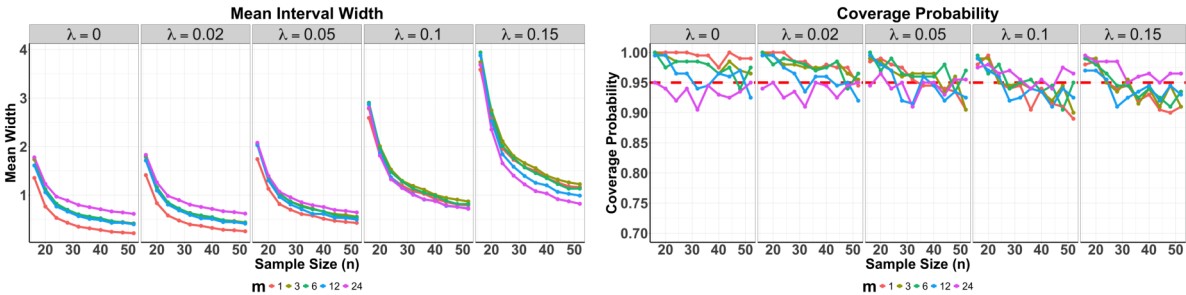

*Figure S3.* Confidence intervals in real-data-based simulation.

set the initial state to $S_{I,1}$, and generate rewards and states according to (1) with the estimated $\{\widehat{\alpha}_t\}_t$, $\{\widehat{\beta}_t\}_t$, $\{\widehat{\phi}_t\}_t$, $\{\widehat{\Phi}_t\}_t$, the specified $\{\widehat{\gamma}_t\}_t$ and $\{\widehat{\Gamma}_t\}_t$, and the error residuals given by $\{\xi_i \widehat{e}_{i,t} : 1 \leq t \leq T\}$ and $\{\xi_i \widehat{E}_{i,t} : 1 \leq t \leq T\}$, respectively. This ensures that the error covariance structure of the simulated data closely resembles that of the real datasets. Based on the simulated data, we similarly apply OLS, LSTD and DRL to estimate the ATE and compare their MSEs under various switchback designs.

***Confidence intervals.*** Since we adopt RL-based estimators for A/B testing, existing methods developed in the reinforcement learning literature can be directly applied for CI construction (see e.g., Shi, 2025, Section 5). In real-data-based simulation, we use nonparametric bootstrap to construct CIs for OLS-based ATE estimators, and report both the coverage probability (CP) and the average CI width of these CIs in Figure S3. It can be seen that most CPs are over 92%, close to the nominal level. Meanwhile,

- For small values of $\lambda$, more frequent policy switch reduces the average CI width.
- When $\lambda$ is increased to 10%, AD produces the narrowest CI on average.

These results verify our claim that a reduction in MSE directly translates to a shorter CI.

***Other results.*** We further visualize the biases and standard deviations of the ATE estimators in synthetic experiments in Figure S4. It can be seen that under the nonlinear DGP, the OLS-based ATE estimators exhibit larger absolute bias than the LSTD- and DRL-based estimators, primarily due to the misspecification of the linear model.

Similarly, Figure S5 displays the biases and standard deviations of the ATE estimators across two real datasets. The standard

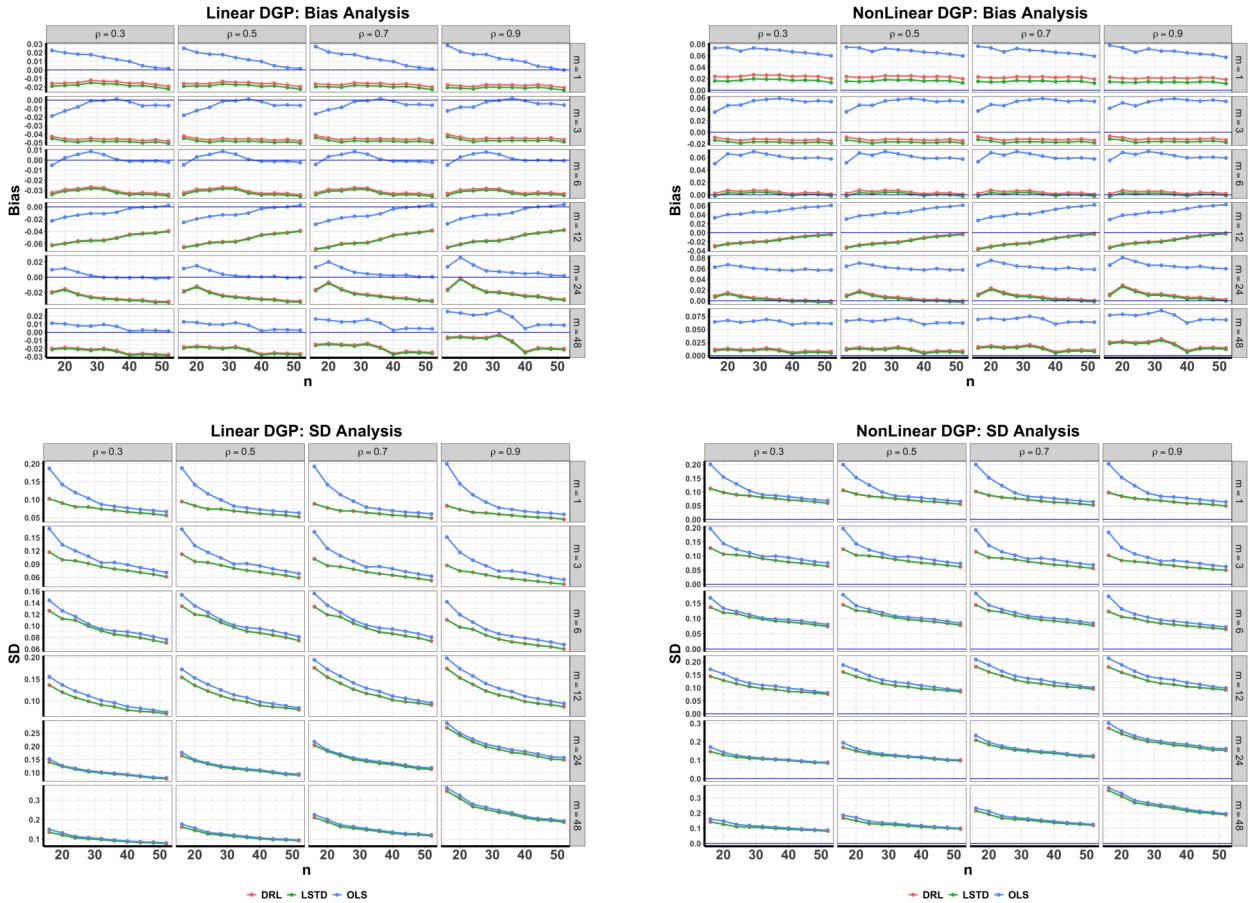

*Figure S4.* Comparison of bias (top row) and standard deviation (bottom row) of different ATE estimators under varying combinations of $(n, m, \rho)$, for linear (left column) and nonlinear (right column) data-generating processes.

deviations of all estimators increase, as expected, as the effect size $\lambda$ or $m$ increases. The biases of the OLS and DRL estimators remain relatively stable with respect to $m$ in the setting of weak carryover effects. However, the LSTD estimator experiences a large bias with a larger effect size. Since these biases are caused by model misspecification, they remain roughly constant across different designs.

## D. Assumptions, Proofs of Theories and Corollaries

In this section, we first present Assumptions 2 – 8 that are needed to establish Theorem 1. Next, we discuss in detail the order of magnitude of the reminder term in Theorem 1. Finally, we provide the proofs of our theorem and corollaries. Throughout this section, we assume without loss of generality that the state space $\mathcal{S}$ is discrete – a typical assumption in RL (see e.g., Sutton & Barto, 2018).

### D.1. Assumptions

**Assumption 2** (Bounded ATE). *The absolute value of the OLS-based ATE estimator is bounded by* $R_{\max}$.

**Assumption 3** (Non-singular covariance matrix). *For any* $1 \leq t \leq T$, *the covariance matrix* $Cov(S_t)$ *is non-singular, whose minimum eigenvalue is larger than* $\max(\epsilon, \bar{c}\delta)$ *for some fixed constant* $\epsilon > 0$ *and some sufficiently large constant* $\bar{c} > 0$.

**Assumption 4** (Bounded regression coefficients). *For any* $1 \leq t \leq T$, $\|\Phi_t\|_2 \leq \rho_\Phi$ *for some constant* $0 < \rho_\Phi < 1$.

**Assumption 5** (Bounded states). *The state dimension* $d$ *is fixed, and the states are contained within a compact ball, i.e., there exists a constant* $C > 0$ *such that for all* $t$, *the states satisfy* $\|S_t\|_2 \leq C$.

**Assumption 6** (Bounded transition functions). *The transition functions* $\{p_t\}$ *are uniformly bounded away from zero and*

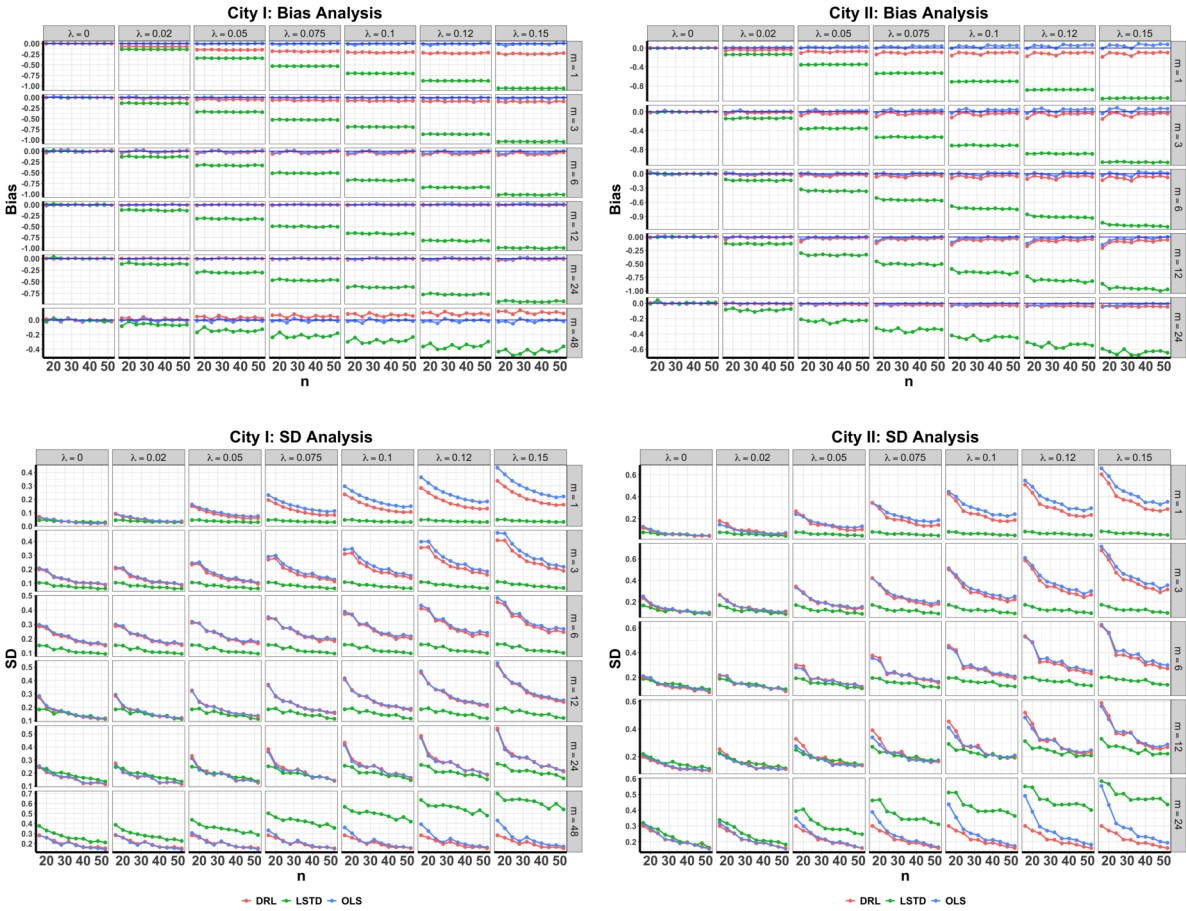

*Figure S5.* Comparison of bias (top row) and standard deviation (bottom row) of different ATE estimators under varying combinations of $(n, m, \lambda)$, based on real datasets from City I (left column) and City II (right column)

*infinity.*

**Assumption 7** (Bounded temporal difference errors). *The absolute value of the temporal difference error $R_t + V_{t+1}(S_{t+1}) - V_t(S_t)$ is of the order $O(R_{\max})$ where the big-O term is uniform in $t$.*

**Assumption 8** (Sieve basis functions). *(i) For any $t \leq T$, there exists a constant $c \geq 1$ such that $c^{-1} \leq \lambda_{\min}\left[ \sum_{s \in \mathcal{S}} \varphi_t(s)\varphi_t^\top(s) \right] \leq \lambda_{\max}\left[ \sum_{s \in \mathcal{S}} \varphi_t(s)\varphi_t^\top(s) \right] \leq c$, where $\lambda_{\min}[\cdot]$ and $\lambda_{\max}[\cdot]$ denote the minimum and maximum eigenvalues of a matrix, respectively; (ii) $\sup_{s \in \mathcal{S}} \|\varphi_t(s)\|_2 < \infty$.*

**Assumption 9** (Convergence rates). *$err_\omega^2 = \max_{t,a,m} \mathbb{E}|\widehat{\omega}_{t,-k}^{a,m}(s) - \omega_t^{a,m}(s)|^2$ and $err_v^2 = R_{\max}^{-2} \max_{t,a,m} \mathbb{E}|\widehat{V}_{t,-k}^{a,m}(s) - V_t^a(s)|^2$ are of the order $o(n^{-1/4})$.*

First, Assumptions 2 and 7 are mild. Given the boundedness of the rewards, it is reasonable to assume both the ATE and the temporal difference error are bounded as well.

Second, Assumptions 3 and 4 are also mild. In particular, Assumption 4 is the classical no unit root assumption for the state process. Both assumptions are crucial to ensure the consistency of the OLS estimators defined in Section 4.1.

Third, in many real-world applications, states are naturally bounded, making Assumption 5 reasonable.

Fourth, Assumption 6 is a standard condition often used to ensure the asymptotic distribution of the LSTD estimator (Shi et al., 2022; 2023b). This assumption is intrinsically linked to the overlap condition, which is critical for maintaining the boundedness of the density ratio $\omega_t$. The overlap condition is commonly imposed in OPE (see e.g., Kallus & Uehara, 2020; 2022; Liao et al., 2022).

Fifth, Assumption 8 is quite reasonable and is automatically satisfied when tensor product B-splines or wavelet basis functions are used for $\varphi_t$ (Chen & Christensen, 2015).

Lastly, we denote the estimated value function by $\widehat{V}_{t,-k}^{a,m}$, where the superscript $m$ highlight its dependence on the $m$-switchback design – different values of $m$ yield different value estimators. The supremum in Assumption 9 is taken with respect to $a, t$ and $m$, but not over $k$, since the trajectories are i.i.d., making the expectation invariant across different $k$. Assumptions of this type are commonly imposed in the literature for valid statistical inference of the ATE (see e.g., Chernozhukov et al., 2018).

### D.2. Theorem 1

In the following, we provide a detailed statement of Theorem 1. Its proof will be presented in the next subsection.

**Theorem 1**. Under the given conditions, the difference in the MSE of the ATE estimator between the alternating-day design and an $m$-switchback design (where each switch duration equals $m$) is lower bounded by

$$\frac{16}{nT^2} \sum_{\substack{k_2-k_1=1,3,5,\dots \\ 0 \le k_1 < k_2 < T/m}} \sum_{l_1,l_2=1}^{m} \sigma_e(l_1+k_1m, l_2+k_2m) - \frac{c\delta R_{\max}^2}{n} - \text{reminder term},$$

for some constant $c > 0$ and some reminder term whose order if estimator-dependent:

- For the **OLS estimator**, under Assumptions 1– 5, its reminder term $= O(n^{-3/2}R_{\max}^2 \log(nT))$.

- For the **DRL estimator**, under Assumptions 1, 6, 7 and 9 its reminder term is of the order

$$O\left[ \max\left( \frac{R_{\max}^2 \text{err}_\omega}{n}, \frac{R_{\max}^2 \text{err}_v}{n}, \frac{R_{\max}^2 \text{err}_\omega \text{err}_v}{\sqrt{n}} \right) \right].$$

- For the **LSTD estimator**, under Assumptions 1, 6 – 9, its reminder term is of the order

$$O\left[ \max\left( \frac{R_{\max}^2 \sqrt{\log(nT)} \text{err}_\omega}{n}, \frac{R_{\max}^2 \sqrt{\log(nT)} \text{err}_v}{n}, \frac{R_{\max}^2 \text{err}_\omega \text{err}_v}{\sqrt{n}} \right) \right].$$

### D.3. Proof of Theorem 1-OLS and Corollaries 1–3

In this subsection, we first consider the OLS estimator and prove all the corollaries. The proofs for the LSTD and DRL estimators are given in the next subsection. Our proof heavily relies on the strength of carryover effects $\delta$, which implies that $\|\Gamma_t\| = O(\delta)$ under the assumption of linear model (1). A key ingredient of the proof of Theorem 1 is the following lemma, which demonstrates that, as $\delta \to 0$, each state becomes asymptotically uncorrelated with the current action. It largely simplifies the calculation of the asymptotic variance of the OLS estimator.

**Lemma 1.** *Suppose both Assumption 4 and the linear model assumption in* (1) *are satisfied. Then for each $t$, we have* $\|Cov(S_t, A_t)\|_2 = O(\delta)$.

*Proof.* According to (9), we obtain that

$$\text{Cov}(S_t, A_t) = \left( \Pi_{k=1}^{t-1} \Phi_k \right) \text{Cov}(S_1, A_t) + \sum_{k=1}^{t-1} \left( \Pi_{l=k+1}^{t-1} \Phi_l \right) \Gamma_k \text{Cov}(A_k, A_t) + \sum_{k=1}^{t-1} \left( \Pi_{l=k+1}^{t-1} \Phi_l \right) \text{Cov}(E_k, A_t),$$

where we recall that $E_t = S_{t+1} - \mathbb{E}(S_{t+1}|A_t, S_t)$.

For each design, $A_t$ is uniquely determined by $A_1$. More specifically, it can take one of two values: either $A_1$ or $1 - A_1$. Since $A_1$ is uncorrelated with $S_1$ and $\{E_k\}_k$, the same holds true for $A_t$. It follows that

$$\text{Cov}(S_t, A_t) = \sum_{k=1}^{t-1} \left( \Pi_{l=k+1}^{t-1} \Phi_l \right) \Gamma_k \text{Cov}(A_k, A_t).$$

Since each $\|\Gamma_t\|$ is $O(\delta)$, and Assumption 4 implies that $\max_t \|\Phi_t\| \le \rho_\Phi < 1$, we have

$$\|\sum_{k=1}^{t-1}\Big(\Pi_{l=k+1}^{t-1}\Phi_l\Big)\Gamma_k\text{Cov}(A_k,A_t))\|_2 \le \sum_{k=1}^{t-1}\|\Pi_{l=k+1}^{t-1}\Phi_l\|\cdot\|\Gamma_k\|_2 \le \sum_{k=1}^{t-1}\Pi_{l=k+1}^{t-1}\|\Phi_l\|\cdot O(\delta) \le \sum_{k=1}^{t-1}\rho_\Phi^{t-k-1}\cdot O(\delta),$$

we have $\|\text{Cov}(S_t,A_t)\|_2 = O(\frac{\delta(1-\rho_\Phi^{t-1})}{1-\rho_\Phi}) = O(\delta)$, for any $t=1,\cdots,T$. This completes the proof of Lemma 1. $\qquad\square$

The next lemma obtains a linear representation for the OLS-based ATE estimator.

**Lemma 2.** *Suppose Assumptions 1, and 3 – 5 and the linear model assumption in* (1) *are satisfied. For sufficiently large* $n$ *and any large constant* $\kappa > 0$, *we have with probability at least* $1 - O((nT)^{-\kappa})$ *that the difference between the OLS-based ATE estimator and the ATE itself can be represented by*

$$\frac{4}{nT}\sum_{i=1}^{n}\sum_{t=1}^{T}(A_{i,t}-\frac{1}{2})e_{i,t} + \frac{4}{nT}\sum_{i=1}^{n}\sum_{t=2}^{T}\beta_t^\top\Big[\sum_{k=1}^{t-1}(\Phi_{t-1}\cdots\Phi_{k+1})(A_{i,k}-\frac{1}{2})E_{i,k}\Big]$$

$$O\Big(\frac{\delta}{T}\sum_{t=1}^{T}\Big\|\frac{1}{n}\sum_{i=1}^{n}\mu_{i,t}\Big\|_2\Big) + O\Big(\frac{R_{\max}\log(nT)}{n}\Big), \tag{13}$$

*where* $\mu_{i,t}$ *denote the vector of residuals*

$$\mu_{i,t} = (e_{i,t}, A_{i,t}e_{i,t}, S_{i,t}^\top e_{i,t}, R_{\max}E_{i,t}^\top, R_{\max}A_{i,t}E_{i,t}^\top, R_{\max}vec(S_{i,t}E_{i,t}^\top))^\top,$$

*and* $vec(\bullet)$ *denotes the operator that vectorize a matrix into a row vector.*

Here, the first two terms in (13) are closely related to the autocorrelation term in (5). Indeed, (5) equals the difference in their variance between AD and an $m$-switchback design. Meanwhile, the third term is closely related to the carryover effect term in (5) and the last term is a high-order reminder term.

*Proof.* For any $m$-switchback design, since the initial action is uniformly generated from $\{0,1\}$, we have $\mathbb{E}A_t = 1/2$, $t = 1,\ldots,T$. According to Lemma 1,

$$\mathbb{E}\begin{pmatrix}1\\S_t\\A_t\end{pmatrix}(1,S_t^\top,A_t) = \begin{pmatrix}1 & \mathbb{E}S_t^\top & \mathbb{E}A_t\\\mathbb{E}S_t & \mathbb{E}S_tS_t^\top & \mathbb{E}A_tS_t\\\mathbb{E}A_t & \mathbb{E}A_tS_t^\top & \mathbb{E}A_t\end{pmatrix} = \underbrace{\begin{pmatrix}1 & \mathbb{E}S_t^\top & 0.5\\\mathbb{E}S_t & \mathbb{E}S_tS_t^\top & 0.5\mathbb{E}S_t+O(\delta)\\0.5 & 0.5\mathbb{E}S_t^\top+O(\delta) & 0.5\end{pmatrix}}_{\Sigma_t}. \tag{14}$$

We first show that $\Sigma_t$ is invertible under Assumptions 3 and 5. Define

$$\Sigma_t^* = \begin{pmatrix}1 & \mathbb{E}S_t^\top & 0.5\\\mathbb{E}S_t & \mathbb{E}S_tS_t^\top & 0.5\mathbb{E}S_t\\0.5 & 0.5\mathbb{E}S_t^\top & 0.5\end{pmatrix}.$$

With some calculations, we can represent $\Sigma_t^*$ by

$$\underbrace{\begin{pmatrix}1 & & \\ \mathbb{E}S_t & 1 & \\ 0.5 & & 1\end{pmatrix}}_{L}\begin{pmatrix}1 & & \\ & \text{Cov}(S_t) & \\ & & 0.25\end{pmatrix}\underbrace{\begin{pmatrix}1 & \mathbb{E}S_t^\top & 0.5\\ & 1 & \\ & & 1\end{pmatrix}}_{L^\top},$$

where $L$ is a lower-triangular matrix and is hence invertible. Under Assumption 3, $\Sigma_t^*$ is invertible and its inverse satisfies

$$\|(\Sigma_t^*)^{-1}\|_2 = \sup_{a:\|a\|_2=1}a^\top\underbrace{\begin{pmatrix}1 & -\mathbb{E}S_t^\top & -0.5\\ & 1 & \\ & & 1\end{pmatrix}\begin{pmatrix}1 & & \\ & \text{Cov}^{-1}(S_t) & \\ & & 4\end{pmatrix}\begin{pmatrix}1 & & \\ -\mathbb{E}S_t & 1 & \\ -0.5 & & 1\end{pmatrix}}_{\Sigma_t^{*-1}}a, \tag{15}$$

which is of the order $O(\lambda_{\min}^{-1}[\mathrm{Cov}(S_t)])$ under Assumption 3.

Consequently, the maximum eigenvalue of $(\Sigma_t^*)^{-1}$ is of the order $O(\lambda_{\min}^{-1}[\mathrm{Cov}(S_t)])$. Equivalently, the minimum eigenvalue of $\Sigma_t^*$ is lower bounded by $c\lambda_{\min}[\mathrm{Cov}(S_t)]$ for some constant $c > 0$. It follows that

$$\lambda_{\min}(\Sigma_t) = \inf_{a:\|a\|_2=1} a^\top \Sigma_t a \geq \inf_{a:\|a\|_2=1} a^\top \Sigma_t^* a - \|\Sigma_t - \Sigma_t^*\|_2 = c\lambda_{\min}[\mathrm{Cov}(S_t)] - O(\delta), \tag{16}$$

and the rightmost term satisfies

$$c\lambda_{\min}[\mathrm{Cov}(S_t)] - O(\delta) = \frac{c}{2}\lambda_{\min}[\mathrm{Cov}(S_t)] + \frac{c}{2}\lambda_{\min}[\mathrm{Cov}(S_t)] - O(\delta) \geq \frac{c\epsilon}{2} + \frac{c\bar{c}\delta}{2} - O(\delta) \geq \frac{c\epsilon}{2}, \tag{17}$$

under Assumption 3, which requires that $\lambda_{\min}[\mathrm{Cov}(S_t)] \geq \max(\epsilon, \bar{c}\delta)$ for some sufficiently large constant $\bar{c} > 0$. This proves the invertibility of $\Sigma_t$.

We next analyze the ATE estimator. Recall from (2) that

$$\mathrm{ATE} = \frac{1}{T}\sum_{t=1}^{T} \gamma_t + \frac{1}{T}\sum_{t=2}^{T} \beta_t^\top \Big[\sum_{k=1}^{t-1}(\Phi_{t-1}\Phi_{t-2}\ldots\Phi_{k+1})\Gamma_k\Big]$$

The OLS-based estimator is constructed by plugging the estimated $\{\gamma_t, \beta_t, \Phi_t, \Gamma_t\}_t$ into this expression. Applying Taylor's expansion to the OLS-based estimator around the true parameter values yields that

$$\mathrm{ATE}_{\mathrm{SB}}^{(m)} - \mathrm{ATE} = \underbrace{\frac{1}{T}\sum_{t=1}^{T}(\widehat{\gamma}_t - \gamma_t)}_{I_1} + \underbrace{\frac{1}{T}\sum_{t=2}^{T} \beta_t^\top \Big[\sum_{k=1}^{t-1}(\Phi_{t-1}\ldots\Phi_{k+1})(\widehat{\Gamma}_k - \Gamma_k)\Big]}_{I_2}$$
$$+ \underbrace{\frac{1}{T}\sum_{t=2}^{T}(\widehat{\beta}_t - \beta_t)^\top \Big[\sum_{k=1}^{t-1}(\Phi_{t-1}\ldots\Phi_{k+1})\Gamma_k\Big]}_{I_3} + \underbrace{\sum_{t=2}^{T-1}\frac{d\mathrm{IE}}{d\Phi_t}(\widehat{\Phi}_t - \Phi_t)}_{I_4} + \text{second-order term,} \tag{18}$$

where IE is a shorthand for the indirect/delayed effect $T^{-1}\sum_{t=2}^{T}\beta_t^\top[\sum_{k=1}^{t-1}(\Phi_{t-1}\ldots\Phi_{k+1})\Gamma_k]$.

Notice that the first four terms on the RHS of (18) (e.g., $I_1$-$I_4$) are the first-order terms, which we now analyze.

**Analysis of $I_1$.** By definition, we have

$$\begin{pmatrix} \widehat{\phi}_t - \phi_t \\ \widehat{\beta}_t - \beta_t \\ \widehat{\gamma}_t - \gamma_t \end{pmatrix} = \widehat{\Sigma}_t^{-1}\frac{1}{n}\sum_{i=1}^{n}\begin{pmatrix} e_{i,t} \\ S_{i,t}e_{i,t} \\ A_{i,t}e_{i,t} \end{pmatrix}. \tag{19}$$

Together with (14), one can show under Assumptions 1 and 3 that

$$\begin{pmatrix} \widehat{\phi}_t - \phi_t \\ \widehat{\beta}_t - \beta_t \\ \widehat{\gamma}_t - \gamma_t \end{pmatrix} = \underbrace{\Sigma_t^{*-1}\frac{1}{n}\sum_{i=1}^{n}\begin{pmatrix} e_{i,t} \\ S_{i,t}e_{i,t} \\ A_{i,t}e_{i,t} \end{pmatrix}}_{\text{leading term}} + \underbrace{[\widehat{\Sigma}_t^{-1} - \Sigma_t^{*-1}]\frac{1}{n}\sum_{i=1}^{n}\begin{pmatrix} e_{i,t} \\ S_{i,t}e_{i,t} \\ A_{i,t}e_{i,t} \end{pmatrix}}_{\text{reminder term}}.$$

Consider the leading term first. Recall that (15) obtains a closed-form expression for $\Sigma_t^{*-1}$. With some calculations, it can be shown that the last row of $\Sigma_t^{*-1}$ equals $(-2, 0, 4)$. This leads to

$$\text{the last entry of the leading term} = \frac{4}{n}\sum_{i=1}^{n}(A_{i,t} - \frac{1}{2})e_{i,t}. \tag{20}$$

Next, consider the reminder term. Note that $(\widehat{\Sigma}_t^{-1} - \Sigma_t^{*-1}) = \widehat{\Sigma}_t^{-1}(\Sigma_t^* - \widehat{\Sigma}_t)\Sigma_t^{*-1}$. The matrix in the middle can be decomposed into the sum of $\Sigma_t^* - \Sigma_t$ and $\Sigma_t - \widehat{\Sigma}_t$. The spectral norm of the first difference, $\Sigma_t^* - \Sigma_t$, is of the order $O(\delta)$.

Below, we aim to apply the matrix Bernstein's inequality (Tropp, 2012) to upper bound the spectral norm of the second difference.

Denote $Z_{it} = (1, S_{it}^\top, A_{it})^\top \in \mathbb{R}^{d+2}$, and $X_{it} = Z_{it} Z_{it}^\top \in \mathbb{R}^{(d+2) \times (d+2)}$, $X_{it}^* = X_{it} - \mathbb{E} X_{it} \in \mathbb{R}^{(d+2) \times (d+2)}$. Clearly, $\Sigma_t = \mathbb{E} X_{it}$, $\widehat{\Sigma}_t = n^{-1} \sum_{i=1}^n X_{it}$. Under the Assumptions 3 and 5, we have $\max_t (\|\widehat{\Sigma}_t\|_2, \|\Sigma_t\|_2) \leq c_\Sigma$ and $\max_{i,t} \|X_{it}^*\|_2 \leq c_\Sigma$, for some finite constant $0 < c_\Sigma < \infty$. Similarly, let $\nu^2 = \max_t \||\sum_{i=1}^n \mathbb{E}(X_{it}^*)^2\|$. We have $\nu^2 = O(n)$. Applying the matrix Bernstein's inequality allows us to obtain the following high probability tail bound for $\max_t \|n^{-1} \sum_{i=1}^n X_{it}^*\|_2$,

$$\mathbb{P}\Big( \max_t \|\frac{1}{n} \sum_{i=1}^n X_{it}^*\|_2 \leq \tau \Big) \geq 1 - 2T(d+2) \exp\Big( -\frac{n^2 \tau^2 / 2}{\nu^2 + \frac{n \tau c_\Sigma}{3}} \Big), \tag{21}$$

for every $\tau > 0$. As the state dimension $d$ is fixed, by setting $\tau$ to be proportional to $n^{-1/2} \kappa \sqrt{\log(nT)}$, the above inequality holds with high probability, at least $1 - O((nT)^{-\kappa})$, for any sufficient large $\kappa > 0$.

To summarize, on the event where (21) holds with $\tau$ proportional to $n^{-1/2} \kappa \sqrt{\log(nT)}$, we have $\|\widehat{\Sigma}_t - \Sigma_t^*\|_2 = O(\delta + n^{-1/2} \kappa \sqrt{\log(nT)})$. Given a sufficiently large $n$, using similar arguments to the proofs of (16) and (17), we can show that

$$\lambda_{\min}(\widehat{\Sigma}_t^{-1}) \geq \frac{c\epsilon}{4}. \tag{22}$$

Since

$$\widehat{\Sigma}_t^{-1} - \Sigma_t^{*-1} = \Sigma_t^{*-1}[\Sigma_t^* - \Sigma_t]\widehat{\Sigma}_t^{-1} + \Sigma_t^{*-1}[\Sigma_t - \widehat{\Sigma}_t]\widehat{\Sigma}_t^{-1},$$

the reminder term can be similarly decomposed into

$$\Sigma_t^{*-1}[\Sigma_t^* - \Sigma_t]\widehat{\Sigma}_t^{-1}\frac{1}{n} \sum_{i=1}^n \begin{pmatrix} e_{i,t} \\ S_{i,t} e_{i,t} \\ A_{i,t} e_{i,t} \end{pmatrix} + \Sigma_t^{*-1}[\Sigma_t - \widehat{\Sigma}_t]\widehat{\Sigma}_t^{-1}\frac{1}{n} \sum_{i=1}^n \begin{pmatrix} e_{i,t} \\ S_{i,t} e_{i,t} \\ A_{i,t} e_{i,t} \end{pmatrix}.$$

According to (19), (22), the boundedness of $\|(\Sigma_t^*)^{-1}\|_2$ (see (15)) and that $\|\Sigma_t^* - \Sigma_t\|_2 = O(\delta)$, the first term is of the order $O(\delta \|n^{-1} \sum_{i=1}^n Z_{i,t} e_{i,t}\|_2)$.

Consider the second term. By definition,

$$(\alpha_t, \beta_t^\top, \gamma_t)^\top = \Sigma_t^{-1} \mathbb{E}(1, S_t, A_t)^\top r(S_t, A_t).$$

Combining (16) with (17) yields the boundedness of $\|\Sigma_t^{-1}\|_2$. This together with the bounded states and bounded rewards assumptions leads to

$$\sup_t \|(\alpha_t, \beta_t^\top, \gamma_t)\|_2 = O(R_{\max}). \tag{23}$$

This together with the bounded rewards and states assumptions yield that $\max_t |e_t| = O(R_{\max})$. Using similar arguments to those for establishing (21), it can be shown that $\|n^{-1} \sum_{i=1}^n Z_{it} e_{it}\|_2$ is of the order $n^{-1/2} \kappa R_{\max} \sqrt{\log(nT)}$ with probability at least $1 - O((nT)^{-\kappa})$. This together with (21), (22) and the boundedness of $\|(\Sigma_t^*)^{-1}\|_2$ yields that the second term is of the order $O(n^{-1} R_{\max} \log(nT))$, with probability at least $1 - O((nT)^{-\kappa})$.

As such, we obtain

$$\text{the reminder term} = O\Big( \|\frac{\delta}{n} \sum_{i=1}^n Z_{i,t} e_{i,t}\|_2 \Big) + O\Big( \frac{R_{\max} \log(nT)}{n} \Big),$$

which together with (20) yields that

$$I_1 = \frac{4}{nT} \sum_{i=1}^n \sum_{t=1}^T (A_{i,t} - \frac{1}{2}) e_{i,t} + O\Big( \frac{\delta}{T} \sum_{t=1}^T \Big\| \frac{1}{n} \sum_{i=1}^n Z_{i,t} e_{i,t} \Big\|_2 \Big) + O\Big( \frac{R_{\max} \log(nT)}{n} \Big). \tag{24}$$

**Analysis of** $I_2$. The analysis of $I_2$ is very similar to that of $I_1$. Specifically, using similar arguments to the proof of (24), it can be shown that

$$\widehat{\Gamma}_t - \Gamma_t = \frac{4}{n}\sum_{i=1}^{n}(A_{i,t} - \frac{1}{2})E_{i,t} + O\Big(\frac{\delta}{T}\sum_{t=1}^{T}\Big\|\frac{1}{n}\sum_{i=1}^{n}\mathrm{vec}(Z_{i,t}E_{i,t}^{\top})\Big\|_2\Big) + O\Big(\frac{\log(nT)}{n}\Big). \tag{25}$$

By (23), $\sup_t \|\beta_t\|_2 = O(R_{\max})$. Since $\max_t \|\Phi_t\|_2 \le \rho_\Phi$, using similar arguments to the proof of Lemma 1, we obtain that $\sup_t \|\beta_t^{\top}\sum_{k=1}^{t-1}(\Phi_{t-1}\ldots\Phi_{k+1})\|_2 = O(R_{\max})$. This together with (25) yields that

$$I_2 = \frac{4}{nT}\sum_{i=1}^{n}\sum_{t=2}^{T}\beta_t^{\top}\Big[\sum_{k=1}^{t-1}(\Phi_{t-1}\cdots\Phi_{k+1})(A_{i,k} - \frac{1}{2})E_{i,k}\Big]$$
$$+O\Big(\frac{\delta R_{\max}}{T}\sum_{t=1}^{T}\Big\|\frac{1}{n}\sum_{i=1}^{n}\mathrm{vec}(Z_{i,t}E_{i,t}^{\top})\Big\|_2\Big) + O\Big(\frac{R_{\max}\log(nT)}{n}\Big). \tag{26}$$

**Analysis of** $I_3$. Since $\|\Gamma_t\|_2 = O(\delta)$ and $\max_t \|\Phi_t\|_2 \le \rho_\Phi$, using similar arguments to the proof of Lemma 1, it can be shown that $\sup_t \|\sum_{k=1}^{t-1}(\Phi_{t-1}\ldots\Phi_{k+1})\Gamma_k\|_2 = O(\delta)$. It follows that

$$I_3 = O\Big(\frac{\delta}{T}\sum_{t=2}^{T}\|\widehat{\beta}_t - \beta_t\|_2\Big). \tag{27}$$

According to (22), $\|\widehat{\beta}_t - \beta_t\|_2$ is of the order of magnitude $O(\|n^{-1}\sum_{i=1}^{n}Z_{i,t}e_{i,t}\|_2)$. This together with (27) leads to

$$I_3 = O\Big(\frac{\delta}{T}\sum_{t=1}^{T}\|\frac{1}{n}\sum_{i=1}^{n}Z_{i,t}e_{i,t}\|_2\Big). \tag{28}$$

**Analysis of** $I_4$. Using similar arguments to the proof of the proof of Lemma 1, one can show that $\max_t \|\frac{d\mathrm{IE}}{d\Phi_t}\|_2 = O(T^{-1}\delta R_{\max})$. Meanwhile, similar to the analysis of $I_3$, we can show that $\|\widehat{\Phi}_t - \Phi_t\|_2$ is of the same order of magnitude to $\|n^{-1}\sum_{i=1}^{n}Z_{i,t}E_{i,t}\|_2$. As such, we obtain that

$$I_4 = O\Big(\frac{\delta R_{\max}}{T}\sum_{t=1}^{T}\|\frac{1}{n}\sum_{i=1}^{n}\mathrm{vec}(Z_{i,t}E_{i,t}^{\top})\|_2\Big). \tag{29}$$

**Second-order terms**. Using similar arguments to the proof of (21), it can be shown that these second-order terms are of the order $O(n^{-1}\log(nT)R_{\max})$, with probability at least $1 - O((nT)^{-\kappa})$. Together with (18), (24), (26), (28) and (29), we conclude the proof of Lemma 2. $\qquad\square$

**Proof of Theorem 1-OLS**

*Proof.* To ease notation, let $c_t$ denote the $d$-dimensional vector $[\sum_{k=t}^{T-1}\beta_{k+1}^{\top}(\Phi_k\cdots\Phi_{t+1})]^{\top}$. According to the Lemma 2, with a sufficiently large $\kappa > 0$, the asymptotic MSE of the ATE estimator is given by

$$\mathrm{MSE}(\mathrm{ATE}_{\mathrm{SB}}^{(m)}) = \mathbb{E}(\mathrm{ATE}_{\mathrm{SB}}^{(m)} - \mathrm{ATE})^2 \mathbb{P}(\text{All the high probability bounds in Lemma 2 hold})$$
$$+ \mathbb{E}(\mathrm{ATE}_{\mathrm{SB}}^{(m)} - \mathrm{ATE})^2 \times \mathbb{P}(\text{One of the high probability bounds in Lemma 2 does not hold}) \tag{30}$$
$$= \mathbb{E}(\mathrm{ATE}_{\mathrm{SB}}^{(m)} - \mathrm{ATE})^2 \times \Big[1 - O((nT)^{-\kappa})\Big] + \mathbb{E}(\mathrm{ATE}_{\mathrm{SB}}^{(m)} - \mathrm{ATE})^2 \times \Big[O((nT)^{-\kappa})\Big].$$

When the high probability bounds do not holds, our ATE estimator can be bounded by $|\mathrm{ATE}_{\mathrm{SB}}^{(m)}| \le R_{\max}$ under Assumption 2. Since $\kappa$ can be made arbitrarily large, the second term in the last line can be made arbitrarily small. Consequently, in the rest of the proof, we focus on the case where all high probability bounds in Lemma 2 hold. All the expectations below are calculated by explicitly assuming that these bounds hold.

Let $I$ denote the third term in (13) that is of the order $O(T^{-1}\delta \sum_{t=1}^{T} \|n^{-1} \sum_{i=1}^{n} \mu_{i,t}\|_2)$ and $I^*$ denote the second-order term in (13) that is of the order $O(n^{-1} R_{\max} \log(nT))$. Since the residual process $\{e_t\}_t$ is independent of all state-action pairs, it is also independent of $\{E_t\}_t$. Consequently, we obtain that

$$
\text{MSE}(\text{ATE}_{\text{SB}}^{(m)}) = \frac{16}{nT^2} \text{Var}\Big[\sum_{t=1}^{T} (A_t - \frac{1}{2})e_t\Big] + \frac{16}{nT^2} \text{Var}\Big[\sum_{t=1}^{T-1} (A_t - \frac{1}{2})c_t^\top E_t\Big]
$$
$$
+ \mathbb{E}(I + I^*)^2 + 2\mathbb{E}(I + I^*)\Big[\frac{4}{nT} \sum_{i=1}^{n} \sum_{t=1}^{T} (A_{i,t} - \frac{1}{2})e_{i,t}\Big] + 2\mathbb{E}(I + I^*)\Big[\frac{4}{nT} \sum_{i=1}^{n} \sum_{t=1}^{T-1} (A_{i,t} - \frac{1}{2})c_t^\top E_{i,t}\Big]. \tag{31}
$$

Let us first focus on the second line of (31). According to Cauchy-Schwarz inequality, the first term on the second line can be upper bounded by $2\mathbb{E}I^2 + 2\mathbb{E}(I^*)^2$. Using Cauchy-Schwarz inequality again, we have

$$
2\mathbb{E}I^2 = O\Big(\mathbb{E}\Big|\frac{\delta}{T} \sum_{t=1}^{T} \Big\|\frac{1}{n} \sum_{i=1}^{n} \mu_{i,t}\Big\|_2\Big|^2\Big) = O\Big(\frac{\delta^2}{T} \sum_{t=1}^{T} \mathbb{E}\Big\|\frac{1}{n} \sum_{i=1}^{n} \mu_{i,t}\Big\|_2^2\Big).
$$

We have shown that $\max_t |e_t| = O(R_{\max})$ in the analysis of $I_1$. Using similar arguments, it can be shown that $\max_t \|E_t\|_2 = O(1)$ under Assumption 5. With some calculations, we can obtain that $\mathbb{E}I^2 = O(n^{-1}\delta^2 R_{\max}^2)$. Meanwhile, according to Lemma 2, we have that $\mathbb{E}(I^*)^2 = O(n^{-2} R_{\max}^2 \log^2(nT))$. Consequently, we have

$$
\mathbb{E}(I + I^*)^2 = O\Big(\frac{\delta^2 R_{\max}^2}{n}\Big) + O\Big(\frac{R_{\max}^2 \log^2(nT)}{n^2}\Big). \tag{32}
$$

Similarly, using Cauchy-Schwarz inequality, the second and third terms in the second line of (31) can be upper bounded by

$$
\sqrt{\mathbb{E}(I + I^*)^2 \mathbb{E}\Big|\frac{4}{nT} \sum_{i=1}^{n} \sum_{t=1}^{T} (A_{i,t} - \frac{1}{2})e_{i,t}\Big|^2} \text{ and } \sqrt{\mathbb{E}(I + I^*)^2 \mathbb{E}\Big|\frac{4}{nT} \sum_{i=1}^{n} \sum_{t=1}^{T-1} (A_{i,t} - \frac{1}{2})c_t^\top E_{i,t}\Big|^2},
$$

respectively. Using (32), one can similarly show that the above two expressions are of the order

$$
O\Big(\frac{\delta R_{\max}^2}{n}\Big) + O\Big(\frac{R_{\max}^2 \log(nT)}{n^{3/2}}\Big). \tag{33}
$$

Since the state space is discrete, $\delta$ – which equals the difference in two probability mass functions – is bounded. Given a sufficiently large $n$, $\mathbb{E}(I + I^*)^2$ is upper bounded by (33) as well. Consequently, the second line of (31) is also of the order specified in (33).

We next consider the second term on the RHS of the first line of (31). The sequence $\{E_t\}$ forms a martingale difference sequence with respect to the filtration $\langle \sigma(\mathcal{F}_t) : t \geq 1 \rangle$ where $\mathcal{F}_t = \{S_j, A_j\}_{j \leq t}$, and is uncorrelated with the sequence $\{A_t\}_t$ under the switchback design. Additionally, under both the alternating-day design and the switchback design, each $A_t$ follows a Bernoulli(0.5) random variable. Its marginal variance is given by 0.25. As such, the second term in the first line of (31) equals

$$
\sum_{t=1}^{T-1} \frac{4c_t^\top \text{Cov}(E_t)c_t}{nT^2}, \tag{34}
$$

and is design-independent.

Finally, we analyze the first term on the RHS of the first line of (31). Consider a given $m$-switchback design. For any integers $1 \leq t_1 \leq t_2 \leq T$, we represent them as $t_1 = l_1 + k_1 m$ and $t_2 = l_2 + k_2 m$ such that $1 \leq l_1, l_2 \leq m$ and $0 \leq k_1, k_2 < T/m$. By definition, $\text{Cov}(A_{t_1}, A_{t_2})$ equals 0.25 if $k_2 - k_1$ is even and $-0.25$ otherwise. Then the first term on the RHS of (31) can thus be represented as

$$
\frac{4}{nT^2} \sum_{l_1=1}^{m} \sum_{l_2=1}^{m} \sum_{k_1=0}^{T/m-1} \sum_{k_2=0}^{T/m-1} (-1)^{|k_1-k_2|}\sigma_e(l_1 + k_1 m, l_2 + k_2 m).
$$

When $m = T$, it equals

$$\frac{4}{nT^2} \sum_{l_1=1}^{m} \sum_{l_2=1}^{m} \sum_{k_1=0}^{T/m-1} \sum_{k_2=0}^{T/m-1} \sigma_e(l_1 + k_1 m, l_2 + k_2 m).$$

Together with (33) and (34), we obtain that

$$\text{MSE}(\text{ATE}_{\text{SB}}^{(m)}) = \frac{4}{nT^2} \sum_{l_1=1}^{m} \sum_{l_2=1}^{m} \sum_{k_1=0}^{T/m-1} \sum_{k_2=0}^{T/m-1} (-1)^{|k_1-k_2|} \sigma_e(l_1 + k_1 m, l_2 + k_2 m)$$

$$+ \frac{4}{nT^2} \sum_{t=1}^{T-1} c_t^\top \text{Cov}(E_t) c_t + O\left(\frac{\delta R_{\max}^2}{n}\right) + O\left(\frac{R_{\max}^2 \log(nT)}{n^{3/2}}\right).$$

Notice that for AD, we have $\Sigma_t = \Sigma_t^*$ and the third term on the RHS of (18) equals zero. This leads to the following lower bound for the difference in the MSE:

$$\text{MSE}(\text{ATE}_{\text{AD}}) - \text{MSE}(\text{ATE}_{\text{SB}}^{(m)}) \geq \frac{16}{nT^2} \sum_{\substack{k_2-k_1=1,3,5,\dots \\ 0 \leq k_1 < k_2 < T/m}} \sum_{l_1=1}^{m} \sum_{l_2=1}^{m} \sigma_e(l_1 + k_1 m, l_2 + k_2 m) - \frac{c \delta R_{\max}^2}{n} - O\left(\frac{R_{\max}^2 \log(nT)}{n^{3/2}}\right),$$

for some constant $c > 0$. The proof is hence completed. $\qquad\square$

### Proof of Corollary 1.

*Proof.* Recall that under the autoregressive covariance structure, $\sigma_e(t_1, t_2) = \sigma^2 \rho^{t_2-t_1}$ for some $0 < \rho < 1$ and any $t_1 \leq t_2$. It follows that

$$\frac{16}{T^2} \sum_{\substack{k_2-k_1=1,3,5,\dots \\ 0 \leq k_1 < k_2 < T/m}} \sum_{l_1=1}^{m} \sum_{l_2=1}^{m} \sigma_e(l_1 + k_1 m, l_2 + k_2 m)$$

$$= \frac{16}{nT^2} \sum_{\substack{k_2-k_1=1,3,5,\dots \\ 0 \leq k_1 < k_2 < T/m}} \sum_{l_1=1}^{m} \sum_{l_2=1}^{m} \sigma^2 \rho^{(k_2-k_1)m+(l_2-l_1)}$$

$$= \frac{16}{nT^2} \sum_{\substack{k_2-k_1=1,3,5,\dots \\ 0 \leq k_1 < k_2 < T/m}} \sigma^2 \rho^{(k_2-k_1)m} \sum_{l_1=1}^{m} \sum_{l_2=1}^{m} \rho^{l_2-l_1}.$$

With some calculations, it is immediate to see that

$$\sum_{l_1=1}^{m} \sum_{l_2=1}^{m} \rho^{l_2-l_1} = \sum_{l_1=1}^{m} \rho^{-l_1} \sum_{l_2=1}^{m} \rho^{l_2} = \frac{\rho^{-1}(1-\rho^{-m})}{1-\rho^{-1}} \frac{\rho(1-\rho^m)}{1-\rho} = \frac{\rho^{1-m}(1-\rho^m)^2}{(1-\rho)^2}. \tag{35}$$

Similarly, when $K = \frac{T}{m}$ is odd,

$$\sum_{\substack{k_2-k_1=1,3,5,\dots \\ 0 \leq k_1 < k_2 < K}} \rho^{(k_2-k_1)m} = \sum_{k_2 \in \{1,\dots,K-2\}} \rho^{k_2 m} + \sum_{k_2 \in \{2,\dots,K-1\}} \rho^{(k_2-1)m} + \dots + \rho^m + \rho^m$$

$$= 2\frac{\rho^m - \rho^{Km}}{1-\rho^{2m}} + 2\frac{\rho^m - \rho^{(K-2)m}}{1-\rho^{2m}} + \dots + 2\frac{\rho^m - \rho^{3m}}{1-\rho^{2m}} + 2\frac{\rho^m - \rho^m}{1-\rho^{2m}}$$

$$= \frac{\rho^m(K+1)}{1-\rho^{2m}} - \frac{2\rho^m(1-\rho^{(K+1)m})}{(1-\rho^{2m})^2} = \frac{\rho^m[T/m - 1 - (T/m+1)\rho^{2m} + 2\rho^{T+m}]}{(1-\rho^{2m})^2}.$$

On the other hand, when $K$ is even, we have

$$\sum_{\substack{k_2-k_1=1,3,5,\dots \\ 0\leq k_1<k_2<K}} \rho^{(k_2-k_1)m} = \sum_{k_2\in\{1,\dots,K-1\}} \rho^{k_2 m} + \sum_{k_2\in\{2,\dots,K-2\}} \rho^{(k_2-1)m} + \dots + \rho^m + \rho^m$$

$$= \frac{\rho^m - \rho^{(K+1)m}}{1-\rho^{2m}} + 2\frac{\rho^m - \rho^{(K-1)m}}{1-\rho^{2m}} + \dots + 2\frac{\rho^m - \rho^{3m}}{1-\rho^{2m}} + 2\frac{\rho^m - \rho^m}{1-\rho^{2m}}$$

$$= \frac{\rho^m(K+1) - \rho^{(K+1)m}}{1-\rho^{2m}} - \frac{2\rho^m(1-\rho^{Km})}{(1-\rho^{2m})^2} = \frac{\rho^m[T/m-1-(T/m+1)\rho^{2m}+\rho^T+\rho^{T+2m}]}{(1-\rho^{2m})^2}.$$

To summarize, we obtain that

$$\sum_{\substack{k_2-k_1=1,3,5,\dots \\ 0\leq k_1<k_2<K}} \rho^{(k_2-k_1)m} = \begin{cases} \dfrac{\rho^m[T/m-1-(T/m+1)\rho^{2m}+2\rho^{T+m}]}{(1-\rho^{2m})^2}, & K \text{ is odd} \\[4mm] \dfrac{\rho^m[T/m-1-(T/m+1)\rho^{2m}+\rho^T+\rho^{T+2m}]}{(1-\rho^{2m})^2}, & K \text{ is even}. \end{cases}$$

In either case, as $T \to \infty$, $T^{-1}\sum_{\substack{k_2-k_1=1,3,5,\dots \\ 0\leq k_1<k_2<K}} \rho^{(k_2-k_1)m}$ becomes asymptotically equivalently to $\rho^m/[m(1-\rho^{2m})]$. Combining this together with (35) yields the desired result. The proof is hence completed. □

*Proof of Corollary 2.*

*Proof.* Under the moving average covariance structure, we have for any $t_1 \leq t_2$ that

$$\text{Cov}(e_{t_1}, e_{t_2}) = \frac{1}{K}\mathbb{E}(\varepsilon_{t_1+1}+\varepsilon_{t_1+2}+\dots\varepsilon_{t_1+K})(\varepsilon_{t_2+1}+\varepsilon_{t_2+2}+\dots\varepsilon_{t_2+K}) = \sigma^2\frac{[K-t_2+t_1]_+}{K},$$

where $[z]_+ = \max(z,0)$ for any $z \in \mathbb{R}$. Accordingly, we have

$$\frac{16}{T^2}\sum_{\substack{k_2-k_1=1,3,5,\dots \\ 0\leq k_1<k_2<T/m}} \sum_{l_1=1}^m \sum_{l_2=1}^m \sigma_e(l_1+k_1 m, l_2+k_2 m)$$

$$= \frac{16\sigma^2}{T^2}\sum_{\substack{k_2-k_1=1,3,5,\dots \\ 0\leq k_1<k_2<T/m}} \sum_{l_1=1}^m \sum_{l_2=1}^m \frac{1}{K}[K-(k_2-k_1)m-(l_2-l_1)]_+. \tag{36}$$

Since $m \geq K$, the expression $K-(k_2-k_1)m-(l_2-l_1)$ remains positive only when $k_2-k_1=1$ and that $l_1-l_2 > m-K$. It follows that the RHS of (36) equals

$$\frac{16\sigma^2}{T^2}\sum_{\substack{k_2-k_1=1 \\ 0\leq k_1<k_2<T/m}} \sum_{l_1=1}^m \sum_{l_2=1}^m \frac{[K-m+l_1-l_2]_+}{K} = \frac{16\sigma^2(T/m-1)}{T^2}\sum_{l_1=1}^m \sum_{l_2=1}^m \frac{[K-m+l_1-l_2]_+}{K}$$

$$= \frac{16\sigma^2(T/m-1)}{T^2}\left(\frac{1+2+\dots+K-1}{K}+\frac{1+2+\dots+K-2}{K}+\dots+\frac{1}{K}\right)$$

$$= \frac{8\sigma^2(T/m-1)(K^2-1)}{3T^2}.$$

The proof is hence completed. □

*Proof of Corollary 3*

*Proof.* Recall that under the exchangeable covariance structure, $\sigma_e(t_1, t_2) = \sigma^2[\rho\mathbb{I}(t_1 \neq t_2) + \mathbb{I}(t_1 = t_2)]$ for any $t_2 \geq t_1$. It follows that

$$\frac{16}{T^2} \sum_{\substack{k_2-k_1=1,3,5,\ldots \\ 0 \leq k_1 < k_2 < T/m}} \sum_{l_1=1}^{m} \sum_{l_2=1}^{m} \sigma_e(l_1 + (k_1-1)m, l_2 + (k_2-1)m)$$

$$= \frac{16}{nT^2} \sum_{\substack{k_2-k_1=1,3,5,\ldots \\ 0 \leq k_1 < k_2 < T/m}} \sum_{l_1=1}^{m} \sum_{l_2=1}^{m} \sigma^2 \rho = \frac{16m^2}{nT^2} \sum_{\substack{k_2-k_1=1,3,5,\ldots \\ 0 \leq k_1 < k_2 < T/m}} \sigma^2 \rho.$$

When $m < T$ and $T/m$ is even, the number of elements in the set $\{(k_1, k_2) : k_2 - k_1 = 1, 3, 5, \ldots; 0 \leq k_1 < k_2 < T/m\}$ is given by $T/(2m) + 2[T/(2m) - 1] + \ldots + 2 = T^2/(4m^2)$. Similarly, when $m < T$ and $T/m$ is odd, it can be shown that the aforementioned set contains $T^2/(4m^2) - 1/4$ elements. It follows that the reduction in MSE equals $4n^{-1}\sigma^2\rho$ when $T/m$ is even and $4n^{-1}\sigma^2\rho(1 - m^2/T^2)$ otherwise. The proof is hence completed. $\square$

### D.4. Proofs of Theorem 1-DRL and LSTD

We first establish the proof for the DRL estimator, followed by the proof for LSTD. With a slight abuse of notation, we use $\omega_t^{a,m}(s) := \frac{p_t^a(s)}{p_t^m(s|a)}$ to represent the IS weight of the conditional probability mass function (pmf) of the state given the action. In particular, the numerator denotes the pmf of $S_t$ under the target policy $a$ and the denominator denotes the conditional pmf of $S_t$ under the $m$-switchback design given $A_t = a$. Additionally, recall that $\omega_t^{a,m}(s, a') := \frac{p_t^a(s,a')}{p_t^m(s,a')}$ denotes the IS weight of the pmf for the state-action pair.

***Proof of Theorem 1-DRL***

*Proof.* Recall that the cross-fitted version of the DRL estimator is given by

$$\text{ATE}_{\text{DRL}}^{(m)} = \frac{1}{nT} \sum_{k=1}^{K} \sum_{i \in \mathcal{D}_k} \psi(\{S_{i,t}, A_{i,t}, R_{i,t}\}_t; \{\widehat{V}_{t,-k}^{a,m}\}_{t,a}, \{\widehat{\omega}_{t,-k}^{a,m}\}_{t,a}).$$

Similar to the proof of Theorem 6 in Kallus & Uehara (2022), we can show that the ATE estimator $\text{ATE}_{\text{DRL}}^{(m)}$ is asymptotically equivalent to its "oracle" version $\text{ATE}_{\text{DRL}}^{(m)*}$ which works as well as if the nuisance functions were known in advance. In particular, we have

$$\text{ATE}_{\text{DRL}}^{(m)} = \underbrace{\frac{1}{nT} \sum_{i=1}^{n} \psi(\{S_{i,t}, A_{i,t}, R_{i,t}\}_t; \{V_t^a\}_{t,a}, \{\omega_t^{a,m}\}_{t,a})}_{\text{ATE}_{\text{DRL}}^{(m)*}} + \text{reminder term.}$$

Notice that the oracle estimator is unbiased. Hence, we only need to compute the asymptotic variances of the oracle estimators under different switchback designs to compare their MSEs.

The rest of the proof is divided into two parts. In Part I, we calculate the asymptotic variance of the "oracle" version of DRL estimator. In Part II, we focus on the calculation of the reminder term.

***Part I:*** Consider the sequence of temporal difference (TD) errors $\{\varepsilon_t^a\}$ where each $\varepsilon_t^a = R_t + V_{t+1}^a(S_{t+1}) - V_t^a(S_t)$. We decompose each TD error into the sum of $e_t$ and $\epsilon_t^a = r_t(A_t, S_t) + V_{t+1}^a(S_{t+1}) - V_t^a(S_t)$, which yields two residual sequences. The first sequence, $\{e_t\}_t$, is correlated over time. Under the Markov assumption and the conditional mean independence assumption (CMIA), the second sequence can be shown to form a martingale difference sequence with respect to the filtration $\langle \sigma(\mathcal{F}_t) : t \geq 1 \rangle$ defined in the proof of Theorem 1-OLS; see e.g., the proof of Theorem 1 in Shi et al. (2022).

Additionally, the two error processes are mutually independent. This yields

$$
\begin{aligned}
n\mathrm{Var}(\mathrm{ATE}_{\mathrm{DRL}}^{(m)*}) &= \frac{1}{T^2}\mathrm{Var}\Big[V_1^1(S_1) - V_1^0(S_1) + \sum_{t=1}^{T}\omega_t^{1,m}(A_t, S_t)\varepsilon_t^1 - \sum_{t=1}^{T}\omega_t^{0,m}(A_t, S_t)\varepsilon_t^0\Big] \\
&= \frac{1}{T^2}\mathrm{Var}\Big[V_1^1(S_1) - V_1^0(S_1)\Big] + \frac{1}{T^2}\sum_{t=1}^{T}\mathrm{Var}\Big[\sum_{a=0}^{1}(-1)^{a+1}\omega_t^{a,m}(A_t, S_t)\epsilon_t^a\Big] \\
&\quad + \frac{1}{T^2}\mathrm{Var}\Big[\sum_{a=0}^{1}\sum_{t=1}^{T}(-1)^{a+1}\omega_t^{a,m}(A_t, S_t)e_t\Big].
\end{aligned}
\tag{37}
$$

Notice that the first term on the second line is design-independent.

Since

$$
\omega_t^{a,m}(A_t, S_t) = \frac{p_t^a(A_t, S_t)}{p_t^m(A_t, S_t)} = \frac{p_t^a(S_t)\mathbb{I}(A_t = a)}{p_t^m(A_t, S_t)} = \frac{p_t^a(S_t)\mathbb{I}(A_t = a)}{p_t^m(S_t|A_t)p_t^m(A_t)} = \frac{p_t^a(S_t)\mathbb{I}(A_t = a)}{p_t^m(S_t|a)p_t^m(a)},
$$

where $p_t^m(a)$ denotes the probability that $A_t = a$ under the $m$-switchback design, which is equal to $1/2$ by construction. With some calculations, the second term on the second line can be shown to equal

$$
\begin{aligned}
\frac{1}{T^2}\sum_{a=0}^{1}\sum_{t=1}^{T}\mathrm{Var}\Big[\omega_t^{a,m}(A_t, S_t)\epsilon_t^a\Big] &= \frac{4}{T^2}\sum_{a=0}^{1}\sum_{t=1}^{T}\mathbb{E}\frac{[p_t^a(S_t)]^2\mathbb{I}(A_t = a)\sigma_{\epsilon,t}^2(a, S_t)}{[p_t^m(S_t|a)]^2} \\
&= \frac{2}{T^2}\sum_{a=0}^{1}\sum_{t=1}^{T}\sum_{s\in\mathcal{S}}\frac{[p_t^a(s)]^2\sigma_{\epsilon,t}^2(a, s)}{p_t^m(s|a)},
\end{aligned}
\tag{38}
$$

where $\sigma_{\epsilon,t}^2(a, s) = \mathrm{Var}(\epsilon_t^a|A_t = a|S_t = s)$.

We note that

- Similar to the proof of model-based methods, according to the definition of $\delta$, it can be shown that $p_t^a(s) = p_t^m(s|a) + O(\delta)$ for any $t \geq 2$ and $p_1^a(s) = p_1^m(s|a)$. Meanwhile, by Assumption 6, the transition function is bounded away from zero. This implies that $p_t^m$ is bounded away from zero as well, for any $t \geq 2$, and hence $p_t^a(s)/p_t^m(s|a) = 1 + O(\delta)$ for any $t \geq 1$.

- Similarly, one can also show that $\sum_{s\in\mathcal{S}}|\sigma_{\epsilon,t}^2(1, s) - \sigma_{\epsilon,t}^2(0, s)| = O(\delta R_{\max}^2)$, and $\max(\sigma_{\epsilon,t}^2(0, s), \sigma_{\epsilon,t}^2(1, s)) = O(R_{\max}^2)$ under Assumption 7.

As such, the second line of (38) is equivalent to

$$
\begin{aligned}
\frac{2}{T^2}\sum_{a=0}^{1}\sum_{t=1}^{T}\sum_{s\in\mathcal{S}}\frac{[p_t^a(s)]^2\sigma_{\epsilon,t}^2(a, s)}{p_t^m(s|a)} &= \frac{2}{T^2}\sum_{a=0}^{1}\sum_{t=1}^{T}\sum_{s\in\mathcal{S}}p_t^a(s)\sigma_{\epsilon,t}^2(a, s) + O(\delta R_{\max}^2) \\
&= \frac{2}{T^2}\sum_{a=0}^{1}\sum_{t=1}^{T}\sum_{s\in\mathcal{S}}p_t^a(s)\sigma_{\epsilon,t}^2(0, s) + O(\delta R_{\max}^2).
\end{aligned}
$$

Consequently, the above expression is asymptotically design-independent, i.e.,

$$
\frac{1}{T^2}\sum_{a=0}^{1}\sum_{t=1}^{T}\mathrm{Var}\Big[\omega_t^{a,m}(A_t, S_t)\epsilon_t^a\Big] = \frac{2}{T^2}\sum_{t=1}^{T}[\mathbb{E}^0\sigma_{\epsilon,t}^2(0, S_t) + \mathbb{E}^1\sigma_{\epsilon,t}^2(0, S_t)] + O(\delta R_{\max}^2).
\tag{39}
$$

Based on the above discussion, it suffices to compare the third line of (37) to compare different switchback designs. Similar to (39), we can show that the third line is asymptotically equivalent to

$$
\frac{4}{T^2}\sum_{a=0}^{1}\mathrm{Var}\Big[\sum_{t=1}^{T}\mathbb{I}(A_t = a)e_t\Big] + O(\delta R_{\max}^2).
$$

Using similar arguments in the proof of Theorem 1-OLS, we obtain that the lower bound of difference in the variance between the alternating-day design and the $m$-switchback design is given by

$$\frac{16}{nT^2} \sum_{\substack{k_2-k_1=1,3,5,\dots \\ 0 \le k_1 < k_2 < T/m}} \sum_{l_1=1}^{m} \sum_{l_2=1}^{m} \sigma_e(l_1 + k_1 m, l_2 + k_2 m) - \frac{c\delta R_{\max}^2}{n} - \text{reminder term},$$

for some constant $c > 0$.

**Part II:** It suffices to upper bound the absolute value of the reminder term, i.e., the difference between $\text{ATE}_{\text{DRL}}^{(m)}$ and $\text{ATE}_{\text{DRL}}^{(m)*}$. With some calculations, the reminder term can be shown to equal the sum of the following three terms:

$$J_1 := \frac{1}{nT} \sum_{k=1}^{K} \sum_{i \in \mathcal{D}_k} [\psi(\{S_{i,t}, A_{i,t}, R_{i,t}\}; \{V_t^a\}_{t,a}; \{\widehat{\omega}_{t,-k}^{a,m}\}_{t,a}) - \psi(\{S_{i,t}, A_{i,t}, R_{i,t}\}; \{V_t^a\}_{t,a}; \{\omega_t^{a,m}\}_{t,a})].$$

$$J_2 := \frac{1}{nT} \sum_{k=1}^{K} \sum_{i \in \mathcal{D}_k} [\psi(\{S_{i,t}, A_{i,t}, R_{i,t}\}; \{\widehat{V}_{t,-k}^{a,m}\}_{t,a}; \{\omega_t^{a,m}\}_{t,a}) - \psi(\{S_{i,t}, A_{i,t}, R_{i,t}\}; \{V_t^a\}_{t,a}; \{\omega_t^{a,m}\}_{t,a})].$$

$$J_3 := \frac{2}{nT} \sum_{k=1}^{K} \sum_{i \in \mathcal{D}_k} \sum_{t=1}^{T} \sum_{a=0}^{1} (-1)^{a+1} \mathbb{I}(A_{i,t} = a) [\widehat{\omega}_{t,-k}^{a,m}(S_{i,t}) - \omega_t^{a,m}(S_{i,t})]$$
$$\times [\widehat{V}_{t+1,-k}^{a,m}(S_{i,t+1}) - \widehat{V}_{t,-k}^{a,m}(S_{i,t}) - V_{t+1}^a(S_{i,t+1}) + V_t^a(S_{i,t})].$$

We analyze each of these terms below.

**Analysis of $J_1$.** With some calculations, we can represent $J_1$ by

$$J_1 = \frac{2}{nT} \sum_{k=1}^{K} \sum_{i \in \mathcal{D}_k} \sum_{t=1}^{T} \sum_{a=0}^{1} (-1)^{a+1} \mathbb{I}(A_{i,t} = a) \left[\widehat{\omega}_{t,-k}^{a,m}(S_{i,t}) - \omega_t^{a,m}(S_{i,t})\right] \left[R_{i,t} + V_{t+1}^a(S_{i,t+1}) - V_t^a(S_{i,t})\right]$$

$$= \frac{2}{nT} \sum_{k=1}^{K} \sum_{i \in \mathcal{D}_k} \sum_{t=1}^{T} \sum_{a=0}^{1} (-1)^{a+1} \mathbb{I}(A_{i,t} = a) \left[\widehat{\omega}_{t,-k}^{a,m}(S_{i,t}) - \omega_t^{a,m}(S_{i,t})\right] \varepsilon_{i,t}^a.$$

It follows from the Cauchy-Schwarz inequality that

$$\mathbb{E}J_1^2 = \mathbb{E} \left\{ \frac{2}{nT} \sum_{k=1}^{K} \sum_{i \in \mathcal{D}_k} \sum_{t=1}^{T} \sum_{a=0}^{1} (-1)^{a+1} \mathbb{I}(A_{i,t} = a) \left[\widehat{\omega}_{t,-k}^{a,m}(S_{i,t}) - \omega_t^{a,m}(S_{i,t})\right] \varepsilon_{i,t}^a \right\}^2$$

$$\le \frac{4K}{n^2 T^2} \sum_{k=1}^{K} \mathbb{E} \left\{ \sum_{i \in \mathcal{D}_k} \sum_{t=1}^{T} \sum_{a=0}^{1} \left[\widehat{\omega}_{t,-k}^{a,m}(S_{i,t}) - \omega_t^{a,m}(S_{i,t})\right] \varepsilon_{i,t}^a \right\}^2$$

$$= \frac{4K}{n^2 T^2} \sum_{k=1}^{K} |\mathcal{D}_k| \text{Var} \left\{ \sum_{t=1}^{T} \sum_{a=0}^{1} \left[\widehat{\omega}_{t,-k}^{a,m}(S_t) - \omega_t^{a,m}(S_t)\right] \varepsilon_t^a \right\}$$

$$+ \frac{4K}{n^2 T^2} \sum_{k=1}^{K} \left\{ \mathbb{E} \left[ \sum_{i \in \mathcal{D}_k} \sum_{t=1}^{T} \sum_{a=0}^{1} \left[\widehat{\omega}_{t,-k}^{a,m}(S_{i,t}) - \omega_t^{a,m}(S_{i,t})\right] \varepsilon_{i,t}^a \right] \right\}^2.$$

Observe that the conditional mean of the temporal difference error is zero, given any state-action pair. Using Cauchy-Schwarz inequality again, we have

$$\mathbb{E}J_1^2 \le \frac{4K}{n^2 T^2} \sum_{k=1}^{K} |\mathcal{D}_k| \text{Var} \left\{ \sum_{t=1}^{T} \sum_{a=0}^{1} \left[\widehat{\omega}_{t,-k}^{a,m}(S_t) - \omega_t^{a,m}(S_t)\right] \varepsilon_t^a \right\}$$

$$\le \frac{4K}{n^2 T} \sum_{k=1}^{K} |\mathcal{D}_k| \sum_{t=1}^{T} \mathbb{E} \left\{ \sum_{a=0}^{1} \left[\widehat{\omega}_{t,-k}^{a,m}(S_t) - \omega_t^{a,m}(S_t)\right] \varepsilon_t^a \right\}^2$$

$$\le \frac{8K}{n^2 T} \sum_{k=1}^{K} |\mathcal{D}_k| \sum_{t=1}^{T} \sum_{a=0}^{1} \mathbb{E} \left\{ \left[\widehat{\omega}_{t,-k}^{a,m}(S_t) - \omega_t^{a,m}(S_t)\right] \varepsilon_t^a \right\}^2.$$

Under Assumption 7, one can show that $\sup_{t,a} |\varepsilon_t^a| = O(R_{\max})$. It follows that $\mathbb{E}(J_1^2) = O(n^{-1} R_{\max}^2 \text{err}_\omega^2)$.

**Analysis of $J_2$.** Similar to the analysis of $J_1$, we can rewrite $J_2$ as

$$\frac{1}{nT} \sum_{k=1}^{K} \sum_{i \in \mathcal{D}_k} \left[ \widehat{V}_{1,-k}^{1,m}(S_{i,1}) - V_1^1(S_{i,1}) + V_1^0(S_{i,1}) - \widehat{V}_{1,-k}^{0,m}(S_{i,1}) \right]$$

$$+ \frac{2}{nT} \sum_{k=1}^{K} \sum_{i \in \mathcal{D}_k} \sum_{t=1}^{T} \sum_{a=0}^{1} (-1)^{a+1} \mathbb{I}(A_{i,t} = a) \omega_t^{a,m}(S_{i,t}) \left[ \widehat{V}_{t+1,-k}^{a,m}(S_{i,t+1}) - V_{t+1}^a(S_{i,t+1}) + V_t^a(S_{i,t}) - \widehat{V}_{t,-k}^{a,m}(S_{i,t}) \right].$$

Using Cauchy-Schwarz inequality, its second moment can be similarly bounded by

$$\frac{K}{n^2 T^2} \sum_{k=1}^{K} \text{Var} \left\{ \sum_{i \in \mathcal{D}_k} \left[ \left[ \widehat{V}_{1,-k}^{1,m}(S_{i,1}) - V_1^1(S_{i,1}) + V_1^0(S_{i,1}) - \widehat{V}_{1,-k}^{0,m}(S_{i,1}) \right] \right. \right.$$

$$\left. + \sum_{t=1}^{T} \sum_{a=0}^{1} 2(-1)^{a+1} \mathbb{I}(A_{i,t} = a) \omega_t^{a,m}(S_{i,t}) \left[ \widehat{V}_{t+1,-k}^{a,m}(S_{i,t+1}) - V_{t+1}^a(S_{i,t+1}) + V_t^a(S_{i,t}) - \widehat{V}_{t,-k}^{a,m}(S_{i,t}) \right] \right] \right\}$$

$$+ \frac{K}{n^2 T^2} \sum_{k=1}^{K} \left\{ \mathbb{E} \left[ \sum_{i \in \mathcal{D}_k} \left[ \left[ \widehat{V}_{1,-k}^{1,m}(S_{i,1}) - V_1^1(S_{i,1}) + V_1^0(S_{i,1}) - \widehat{V}_{1,-k}^{0,m}(S_{i,1}) \right] \right. \right. \right.$$

$$\left. \left. \left. + \sum_{t=1}^{T} \sum_{a=0}^{1} 2(-1)^{a+1} \mathbb{I}(A_{i,t} = a) \omega_t^{a,m}(S_{i,t}) \left[ \widehat{V}_{t+1,-k}^{a,m}(S_{i,t+1}) - V_{t+1}^a(S_{i,t+1}) + V_t^a(S_{i,t}) - \widehat{V}_{t,-k}^{a,m}(S_{i,t}) \right] \right] \right] \right\}^2. \tag{40}$$

By the double robustness property, under correct specification of $w_t^{a,m}$, the squared bias term in (40) is equal to zero. Thus, it suffices to upper bound the variance term. With some calculations, we have

$$\frac{K}{n^2 T^2} \sum_{k=1}^{K} |\mathcal{D}_k| \text{Var} \left\{ \left[ \widehat{V}_{1,-k}^{1,m}(S_1) - V_1^1(S_1) + V_1^0(S_1) - \widehat{V}_{1,-k}^{0,m}(S_1) \right] \right.$$

$$\left. + \sum_{t=1}^{T} \sum_{a=0}^{1} 2(-1)^{a+1} \mathbb{I}(A_t = a) \omega_t^{a,m}(S_t) \left[ \widehat{V}_{t+1,-k}^{a,m}(S_{t+1}) - V_{t+1}^a(S_{t+1}) + V_t^a(S_t) - \widehat{V}_{t,-k}^{a,m}(S_t) \right] \right\}$$

$$\leq \frac{2K}{n^2 T^2} \sum_{k=1}^{K} |\mathcal{D}_k| \mathbb{E} \left\{ \left[ \widehat{V}_{1,-k}^{1,m}(S_1) - V_1^1(S_1) + V_1^0(S_1) - \widehat{V}_{1,-k}^{0,m}(S_1) \right] \right\}^2$$

$$+ \frac{16K}{n^2 T^2} \sum_{k=1}^{K} |\mathcal{D}_k| \sum_{a=0}^{1} \mathbb{E} \left\{ \sum_{t=1}^{T} \omega_t^{a,m}(S_t) \left[ \widehat{V}_{t+1,-k}^{a,m}(S_{t+1}) - V_{t+1}^a(S_{t+1}) + V_t^a(S_t) - \widehat{V}_{t,-k}^{a,m}(S_t) \right] \right\}^2$$

$$\leq \frac{4K}{n^2 T^2} \sum_{k=1}^{K} |\mathcal{D}_k| \sum_{a=0}^{1} \mathbb{E} \left[ \widehat{V}_{a,-k}^{a,m}(S_1) - V_1^a(S_1) \right]^2 + \frac{32K}{n^2 T} \sum_{k=1}^{K} |\mathcal{D}_k| \sum_{a=0}^{1} \sum_{t=1}^{T} C_\omega^2 \mathbb{E} \left[ \widehat{V}_{t,-k}^{a,m}(S_t) - V_t^a(S_t) \right]^2,$$

where the two inequalities follow from the Cauchy-Schwarz inequality and $C_\omega$ denotes the upper bound of $w_t^{a,m}$, which is finite under Assumption 6.

As such, by the definition of $\text{err}_v$, we have $\mathbb{E}(J_2^2) = O(n^{-1} R_{\max}^2 \text{err}_v^2)$.

**Analysis of $J_3$.** Similarly, using Cauchy-Schwarz inequality, $\mathbb{E}(J_3^2)$ can be upper bounded by

$$\frac{4K}{n^2 T^2} \sum_{k=1}^{K} \mathbb{E} \left\{ \sum_{i \in \mathcal{D}_k} \sum_{t=1}^{T} \sum_{a=0}^{1} [\widehat{\omega}_{t,-k}^{a,m}(S_{i,t}) - \omega_t^{a,m}(S_{i,t})][\widehat{V}_{t+1,-k}^{a,m}(S_{i,t+1}) - \widehat{V}_{t,-k}^{a,m}(S_{i,t}) - V_{t+1}^a(S_{i,t+1}) + V_t^a(S_{i,t})] \right\}^2$$

$$= \frac{4K}{n^2 T^2} \sum_{k=1}^{K} |\mathcal{D}_k| \text{Var} \left\{ \sum_{t=1}^{T} \sum_{a=0}^{1} [\widehat{\omega}_{t,-k}^{a,m}(S_t) - \omega_t^{a,m}(S_t)][\widehat{V}_{t+1,-k}^{a,m}(S_{t+1}) - \widehat{V}_{t,-k}^{a,m}(S_t) - V_{t+1}^a(S_{t+1}) + V_t^a(S_t)] \right\} \tag{41}$$

$$+ \frac{4K}{n^2 T^2} \sum_{k=1}^{K} \left\{ |\mathcal{D}_k|^2 \mathbb{E} \left[ \sum_{t=1}^{T} \sum_{a=0}^{1} [\widehat{\omega}_{t,-k}^{a,m}(S_t) - \omega_t^{a,m}(S_t)][\widehat{V}_{t+1,-k}^{a,m}(S_{t+1}) - \widehat{V}_{t,-k}^{a,m}(S_t) - V_{t+1}^a(S_{t+1}) + V_t^a(S_t)] \right] \right\}^2.$$

Here, the squared bias term on the third line is the dominating factor. Applying Cauchy-Schwarz inequality again, it can be upper bounded by $O(R_{\max}^2 \text{err}_\omega^2 \text{err}_v^2)$. Meanwhile, using similar arguments to the analysis of $J_2$, we can show that the second line can be upper bounded by $O(n^{-1}R_{\max}^2\text{err}_v^2)$. Thus, $\mathbb{E}[J_3^2] = O(R_{\max}^2\text{err}_\omega^2\text{err}_v^2 + n^{-1}R_{\max}^2\text{err}_v^2)$.

To summarize, we have show that $\mathbb{E}J_1^2 + \mathbb{E}J_2^2 + \mathbb{E}J_3^2$ is of the order

$$O\Big[\max\Big(\frac{R_{\max}^2\text{err}_\omega^2}{n}, \frac{R_{\max}^2\text{err}_v^2}{n}, R_{\max}^2\text{err}_\omega^2\text{err}_v^2\Big)\Big].$$

Using Cauchy-Schwarz inequality, we obtain that $\mathbb{E}(J_1 + J_2 + J_3)^2$ is of the same order of magnitude. Notice that

$$\text{MSE}(\text{ATE}_{\text{DRL}}^{(m)}) = \text{MSE}(\text{ATE}_{\text{DRL}}^{(m)*}) + \mathbb{E}(J_1 + J_2 + J_3)^2 + 2\mathbb{E}(\text{ATE}_{\text{DRL}}^{(m)*}(J_1 + J_2 + J_3)),$$

where the absolute value of the last term can be upper bounded by

$$\sqrt{\mathbb{E}(J_1 + J_2 + J_3)^2\mathbb{E}(\text{ATE}_{\text{DRL}}^{(m)*})^2} = O\Big[\max\Big(\frac{R_{\max}^2\text{err}_\omega}{n}, \frac{R_{\max}^2\text{err}_v}{n}, \frac{R_{\max}^2\text{err}_\omega\text{err}_v}{\sqrt{n}}\Big)\Big]$$

As the error terms $\text{err}_v$ and $\text{err}_w$ are $o(n^{-1/4})$ and hence bounded, combining these results together yields the desired assertion. $\qquad\square$

### Proof of Theorem 1-LSTD.

*Proof.* The proof of Theorem 1-LSTD relies on demonstrating that the LSTD estimator can be expressed as a DRL estimator (whose detailed form is given in Equation (45) below), allowing us to apply the proof techniques from Theorem 1-DRL to establish Theorem 1-LSTD. We break the rest of the proof into two parts. In Part I, we establish the equivalence between the two estimators. In Part II, our aim is to calculate the order of the difference between the RL estimator $\text{ATE}_{\text{DRL}}^{(m)}$ and its "oracle" version $\text{ATE}_{\text{DRL}}^{(m)*}$, and then analyze its order of magnitude.

**Part I**. Recall that the value function estimator obtained via LSTD is given by $\varphi_t^\top(s)\widehat{\theta}_{t,a,m}$. The DRL estimator, whose equivalence to LSTD we aim to establish, uses this estimated value along with a linearly parameterized $\widehat{\omega}_t^{a,m}(s)$. Specifically, for each $t \geq 1$, we set $\widehat{\omega}_t^{a,m}(s) = \varphi_t^\top(s)\widehat{\alpha}_{t,a,m}$ where the estimators $\{\widehat{\alpha}_{t,a,m}\}_t$ are computed in a forward manner. By definition

$$\omega_1^{a,m}(s) = \frac{p_1(s)}{p_1^m(s|a)} = \frac{p_1(s)\mathbb{P}(A_1 = a)}{b_1(a|s)p_1(s)} = \frac{1}{2b_1(a|s)}, \tag{42}$$

where $b_1$ denotes the propensity score whose oracle value is a constant function equal to 0.5 by design. To estimate $\alpha_{1,a,m}$, notice that according to (42), $\mathbb{E}[\omega_1^{a,m}(S_1)\mathbb{I}(A_1 = a)h(S_1) - h(S_1)] = 0$ for any function $h$. This motivates us to compute $\widehat{\alpha}_{1,a,m}$ by solving the following estimating equation,

$$\frac{1}{n}\sum_{i=1}^n\Big[\varphi_1^\top(S_{i,1})\widehat{\alpha}_{1,a,m}\mathbb{I}(A_{i,1} = a)\varphi_1(S_{i,1}) - \varphi_1(S_{i,1})\Big] = 0,$$

which yields

$$\widehat{\alpha}_{1,a,m} = \underbrace{\Big[\frac{1}{n}\sum_{i=1}^n\varphi_1(S_{i,1})\varphi_1^\top(S_{i,1})\mathbb{I}(A_{i,1} = a)\Big]^{-1}}_{\widehat{\Sigma}_1^a}\Big[\frac{1}{n}\sum_{i=1}^n\varphi_1(S_{i,1})\Big]. \tag{43}$$

Next, consider a given $t \geq 2$. Using the Bayes rule, it is straightforward to show that

$$\omega_t^{a,m}(s) = \frac{p_t^a(s)}{p_t^m(s|a)} = \frac{p_t^a(s)\mathbb{P}(A_t = a)}{p_t^m(a,s)} = \frac{p_t^a(s)}{2p_t^m(a,s)},$$

where $p_t^m(a,s)$ denote the join distribution function of $(A_t, S_t)$. Consequently, using the Bayes rule again, we obtain that

$$\mathbb{E}[\omega_{t-1}^{a,m}(S_{t-1})|A_{t-1} = a, S_t] = \frac{p_t^a(S_t)}{2p_t^m(a, S_t)},$$

for any $a$ and $S_t$. This motivates us to estimate $\alpha_{t,a,m}$ by solving the following estimating equation,

$$\frac{1}{n} \sum_{i=1}^{n} \left[ \varphi_{t-1}^{\top}(S_{i,t-1}) \widehat{\alpha}_{t-1,a,m} \mathbb{I}(A_{i,t-1} = a) \varphi_t(S_{i,t}) \right] = \frac{1}{n} \sum_{i=1}^{n} \left[ \varphi_t^{\top}(S_{i,t}) \widehat{\alpha}_{t,a,m} \mathbb{I}(A_{i,t} = a) \varphi_t(S_{i,t}) \right],$$

which yields that

$$\widehat{\alpha}_{t,a,m} = \underbrace{\left[ \frac{1}{n} \sum_{i=1}^{n} \varphi_t(S_{i,t}) \varphi_t^{\top}(S_{i,t}) \mathbb{I}(A_{i,t} = a) \right]^{-1}}_{\widehat{\Sigma}_t^a} \underbrace{\left[ \frac{1}{n} \sum_{i=1}^{n} \varphi_t(S_{i,t}) \varphi_{t-1}^{\top}(S_{i,t-1}) \mathbb{I}(A_{i,t-1} = a) \right]}_{\widehat{\Sigma}_{t,t-1}^a} \widehat{\alpha}_{t-1,a,m}, \tag{44}$$

and hence

$$\widehat{\alpha}_{t,a,m} = (\widehat{\Sigma}_t^a)^{-1} \widehat{\Sigma}_{t,t-1}^a (\widehat{\Sigma}_{t-1}^a)^{-1} \widehat{\Sigma}_{t-1,t-2}^a \cdots (\widehat{\Sigma}_1^a)^{-1} \left[ \frac{1}{n} \sum_{i=1}^{n} \varphi_1(S_{i,1}) \right].$$

This leads to the following DRL estimator

$$\text{ATE}_{\text{DRL}}^{(m)} = \frac{1}{nT} \sum_{i=1}^{n} \psi(\{S_{i,t}, A_{i,t}, R_{i,t}\}; \{\widehat{V}_t^{a,m}\}_{t,a}; \{\widehat{\omega}_t^{a,m}\}_{t,a}). \tag{45}$$

We aim to show that this DRL estimator equals the LSTD estimator based on $\{\widehat{V}_t^{a,m}\}_{t,a}$. According to the definition of the estimating function $\psi$, the DRL estimator can be naturally decomposed into two parts where the first part coincides the LSTD estimator and the second part corresponds to the augmentation term,

$$\frac{2}{nT} \sum_{i=1}^{n} \sum_{t=1}^{T} \sum_{a=0}^{1} (-1)^{a+1} \mathbb{I}(A_{i,t} = a) \widehat{\omega}_t^{a,m}(S_{i,t}) [R_{i,t} + \widehat{V}_{t+1}^{a,m}(S_{i,t+1}) - \widehat{V}_t^{a,m}(S_{i,t})].$$

As such, it suffices to show the augmentation term equals zero, or

$$\frac{1}{n} \sum_{i=1}^{n} \mathbb{I}(A_{i,t} = a) \widehat{\omega}_t^{a,m}(S_{i,t}) [R_{i,t} + \widehat{V}_{t+1}^{a,m}(S_{i,t+1}) - \widehat{V}_t^{a,m}(S_{i,t})] = 0, \tag{46}$$

for each $a$ and $t$. By (44), the left-hand-side (LHS) of (46) equals

$$(\widehat{\alpha}_{t-1,a,m})^{\top} (\widehat{\Sigma}_{t,t-1}^a)^{\top} \left\{ \frac{1}{n} \sum_{i=1}^{n} (\widehat{\Sigma}_t^a)^{-1} \mathbb{I}(A_{i,t} = a) \varphi_t(S_{i,t}) [R_{i,t} + \widehat{V}_{t+1}^{a,m}(S_{i,t+1}) - \widehat{V}_t^{a,m}(S_{i,t})] \right\}.$$

According to the LSTD algorithm, it is immediate to see that the statistic inside the curly brackets equals zero. This formally verifies (46). The proof of Part I is hence completed.

***Part II***. In Part II, we aim to show (45) is asymptotically equivalent to its oracle estimator

$$\text{ATE}_{\text{DRL}}^{(m)*} = \frac{1}{nT} \sum_{i=1}^{n} \psi(\{S_{i,t}, A_{i,t}, R_{i,t}\}; \{V_t^a\}_{t,a}; \{\omega_t^{a,m}\}_{t,a}).$$

The proof is very similar to that of Theorem 1-DRL; for brevity, we provide only a sketch. The main difference lies in the absence of cross-splitting for this DRL estimator.

Similarly, the difference between $\text{ATE}_{\text{DRL}}^{(m)}$ and $\text{ATE}_{\text{DRL}}^{(m)*}$ can be expressed as the sum of the following three terms:

$$J_4 := \frac{1}{nT} \sum_{i=1}^{n} [\psi(\{S_{i,t}, A_{i,t}, R_{i,t}\}; \{V_t^a\}_{t,a}; \{\widehat{\omega}_t^{a,m}\}_{t,a}) - \psi(\{S_{i,t}, A_{i,t}, R_{i,t}\}; \{V_t^a\}_{t,a}; \{\omega_t^{a,m}\}_{t,a})].$$

$$J_5 := \frac{1}{nT} \sum_{i=1}^{n} [\psi(\{S_{i,t}, A_{i,t}, R_{i,t}\}; \{\widehat{V}_t^{a,m}\}_{t,a}; \{\omega_t^{a,m}\}_{t,a}) - \psi(\{S_{i,t}, A_{i,t}, R_{i,t}\}; \{V_t^a\}_{t,a}; \{\omega_t^{a,m}\}_{t,a})].$$

$$\begin{aligned} J_6 := & \frac{2}{nT} \sum_{i=1}^{n} \sum_{t=1}^{T} \sum_{a=0}^{1} (-1)^{a+1} \mathbb{I}(A_{i,t} = a) [\widehat{\omega}_t^{a,m}(S_{i,t}) - \omega_t^{a,m}(S_{i,t})] \\ & \times [\widehat{V}_{t+1}^{a,m}(S_{i,t+1}) - \widehat{V}_t^{a,m}(S_{i,t}) - V_{t+1}^a(S_{i,t+1}) + V_t^a(S_{i,t})]. \end{aligned}$$

**Analysis of** $J_4$. We further decompose $J_4$ into the sum of

$$\frac{1}{nT}\sum_{i=1}^{n}[\psi(\{S_{i,t}, A_{i,t}, R_{i,t}\}; \{V_t^a\}_{t,a}; \{\omega_t^{a,m*}\}_{t,a}) - \psi(\{S_{i,t}, A_{i,t}, R_{i,t}\}; \{V_t^a\}_{t,a}; \{\omega_t^{a,m}\}_{t,a})], \tag{47}$$

and

$$\frac{1}{nT}\sum_{i=1}^{n}[\psi(\{S_{i,t}, A_{i,t}, R_{i,t}\}; \{V_t^a\}_{t,a}; \{\widehat{\omega}_t^{a,m}\}_{t,a}) - \psi(\{S_{i,t}, A_{i,t}, R_{i,t}\}; \{V_t^a\}_{t,a}; \{\omega_t^{a,m*}\}_{t,a})], \tag{48}$$

where $\omega_t^{a,m*}(s) = \varphi_t^\top(s)\alpha_{t,a,m}^*$, and $\alpha_{t,a,m}^* := \min_{\alpha_{t,a,m}\in\mathbb{R}^L}\mathbb{E}(\omega_t^{a,m}(S_t) - \varphi_t^\top(S_t)\alpha_{t,a,m})^2$. Then (47) is equal to

$$J_{4a} := \frac{2}{nT}\sum_{i=1}^{n}\sum_{t=1}^{T}\sum_{a=0}^{1}(-1)^{a+1}\mathbb{I}(A_{i,t} = a)\left[\omega_t^{a,m*}(S_{i,t}) - \omega_t^{a,m}(S_{i,t})\right]\varepsilon_{i,t}^a,$$

whereas (48) is equal to

$$J_{4b} := \frac{2}{nT}\sum_{i=1}^{n}\sum_{t=1}^{T}\sum_{a=0}^{1}(-1)^{a+1}\mathbb{I}(A_{i,t} = a)\left[\widehat{\omega}_t^{a,m}(S_{i,t}) - \omega_t^{a,m*}(S_{i,t})\right]\varepsilon_{i,t}^a.$$

Note that $\mathbb{E}(J_{4a}) = 0$ because the conditional mean of the temporal difference error is equal to zero, given any state-action pair. Thus, we obtain

$$\mathbb{E}(J_{4a})^2 \leq \frac{4}{nT^2}\text{Var}\left\{\sum_{t=1}^{T}\sum_{a=0}^{1}\left[\omega_t^{a,m*}(S_t) - \omega_t^{a,m}(S_t)\right]\varepsilon_t^a\right\} \leq O\left(\frac{8R_{\max}^2}{nT}\right)\sum_{a=0}^{1}\sum_{t=1}^{T}\mathbb{E}\left[\omega_t^{a,m*}(S_t) - \omega_t^{a,m}(S_t)\right]^2,$$

where the second inequality follows from the Cauchy-Schwarz inequality and that $\max_{t,a}|\varepsilon_t^a| = O(R_{\max})$. This leads to $\mathbb{E}(J_{4a}^2) = O(n^{-1}R_{\max}^2\text{app\_err}_\omega^2)$, where we denote app\_err$_\omega^2$ by $\max_{a,m,t}\mathbb{E}|\omega_t^{a,m}(S_t) - \varphi_t^\top(S_t)\alpha_{t,a,m}^*|^2$.

It remains to bound $\mathbb{E}(J_{4b}^2)$. By definition, we can rewrite $J_{4b}$ as follows:

$$\frac{2}{T}\sum_{t=1}^{T}\sum_{a=0}^{1}(-1)^{a+1}(\widehat{\alpha}_{t,a,m} - \alpha_{t,a,m}^*)^\top\left\{\frac{1}{n}\sum_{i=1}^{n}\varphi_t(S_{i,t})\mathbb{I}(A_{i,t} = a)\varepsilon_{i,t}^a\right\}.$$

Observe that the conditional mean of the temporal difference error is zero, given any state-action pair. Consequently, the random vector enclosed within the curly brackets have a mean of zero. Additionally, under Assumption 7, each summand within the curly brackets is bounded by $O(n^{-1}R_{\max})$. Meanwhile, under Assumption 8-(i), we have $\max_t\|\mathbb{E}[\varphi_t(S_t)\varphi_t^\top(S_t)]\|_2 = O(1)$. Consequently, the variance of each summand within the curly brackets is bounded by $O(n^{-2}R_{\max}^2)$. It follows from Bernstein's inequality (see e.g., Van & Wellner, 1996) that with probability at least $1 - O(n^{-\kappa}T^{-\kappa})$ for some sufficiently large $\kappa > 0$, all elements in the curly brackets are upper bounded by $O(n^{-1/2}R_{\max}\sqrt{\log(nT)})$. As such, we have

$$\mathbb{E}(J_{4b}^2) = O\left(\frac{R_{\max}^2\log(nT)\text{est\_err}_\omega^2}{n}\right),$$

where we denote est\_err$_\omega^2 = \max_{a,t,m}\mathbb{E}\|\widehat{\alpha}_{t,a,m} - \alpha_{t,a,m}^*\|_2^2$.

Finally, note that the definition of $\alpha_{t,a,m}^*$ yields that the approximation error $\omega_t^{a,m}(S_t) - \omega_t^{a,m*}(S_t)$ is uncorrelated with the estimation error $\omega_t^{a,m*}(S_t) - \widehat{\omega}_t^{a,m}(S_t)$. Additionally, the non-singularity assumption in Assumption 8-(ii) yields that the second moment of $\omega_t^{a,m*}(S_t) - \widehat{\omega}_t^{a,m}(S_t)$ is of the same order to that of $\widehat{\alpha}_{t,a,m} - \alpha_{t,a,m}^*$. Consequently, we have

$$\mathbb{E}(J_4^2) \leq 2\mathbb{E}J_{4a}^2 + 2\mathbb{E}J_{4b}^2 = O\left(\frac{R_{\max}^2\log(nT)[\text{app\_err}_\omega^2 + \text{est\_err}_\omega^2]}{n}\right) = O\left(\frac{R_{\max}^2\log(nT)\text{err}_\omega^2}{n}\right).$$

**Analysis of** $J_5$. Similar to the analysis of $J_4$, it can be shown that

$$\mathbb{E}(J_5^2) = O\left(\frac{R_{\max}^2\log(nT)\text{err}_v^2}{n}\right).$$

**Analysis of** $J_6$. Define $J_6^*$ as

$$\frac{2}{T} \sum_{t=1}^{T} \sum_{a=0}^{1} (-1)^{a+1} \mathbb{E}\Big\{ \mathbb{I}(A_t = a)[\widehat{\omega}_t^{a,m}(S_t) - \omega_t^{a,m}(S_t)][\widehat{V}_{t+1}^{a,m}(S_{t+1}) - \widehat{V}_t^{a,m}(S_t) - V_{t+1}^a(S_{t+1}) + V_t^a(S_t)] \Big| \widehat{\omega}_t^{a,m}, \widehat{V}_t^{a,m}, \widehat{V}_{t+1}^{a,m} \Big\}.$$

Notice that $J_6^*$ can be viewed as the expected value of $J_6$ when the estimated IS ratio and value function are fixed.

Using similar arguments in the analysis of $J_3$, it can be shown that $\mathbb{E}(J_6^*)^2 = O(R_{\max}^2 \mathrm{err}_w^2 \mathrm{err}_v^2)$.

It remains to consider $J_6 - J_6^*$, which we denote by $\bar{J}_6(\{\widehat{\omega}_t^{a,m} - \omega_t^{a,m}\}_{t,a,m}, \{\widehat{V}_t^{a,m} - V_t^a\}_{t,a,m})$, to highlight its dependence upon the estimated IS ratio and value function as well as their ground truths. This term can be decomposed into $\bar{J}_{6a} + \bar{J}_{6b} + \bar{J}_{6c} + \bar{J}_{6d}$ where

$$
\begin{aligned}
\bar{J}_{6a} &= \bar{J}_6(\{\omega_t^{a,m*} - \omega_t^{a,m}\}_{t,a,m}, \{V_t^{a,m*} - V_t^a\}_{t,a,m}), \\
\bar{J}_{6b} &= \bar{J}_6(\{\omega_t^{a,m*} - \omega_t^{a,m}\}_{t,a,m}, \{\widehat{V}_t^{a,m} - V_t^{a,m*}\}_{t,a,m}), \\
\bar{J}_{6c} &= \bar{J}_6(\{\widehat{\omega}_t^{a,m} - \omega_t^{a,m*}\}_{t,a,m}, \{V_t^{a,m*} - V_t^a\}_{t,a,m}), \\
\bar{J}_{6d} &= \bar{J}_6(\{\widehat{\omega}_t^{a,m} - \omega_t^{a,m*}\}_{t,a,m}, \{\widehat{V}_t^{a,m} - V_t^{a,m*}\}_{t,a,m}).
\end{aligned}
$$

We note that:

- $\bar{J}_{6a}$ can be represented by an average of i.i.d. mean zero random variable. Using similar arguments to those for the second line of (41), we can show that $\mathbb{E}(\bar{J}_{6a}^2)$ is of the order $O(n^{-1} R_{\max}^2 \mathrm{app\_err}_v^2)$.

- Similar to the analysis of $J_{4b}$, we can represent $\bar{J}_{6b}$ as an average $T^{-1} \sum_{t=1}^{T} \bar{J}_{6b}^{(t)}$ where each $\bar{J}_{6b}^{(t)}$ can be represented by the inner product of an average of i.i.d. mean zero random vector and $(\widehat{\theta}_{t,a,m} - \theta_{t,a,m}^*)$. The bound $\mathbb{E}(\bar{J}_{6b}^2) = O(n^{-1} R_{\max}^2 \log(nT) \mathrm{est\_err}_v^2)$ follows similarly to that of $\mathbb{E}(J_{4b}^2)$.

- $\bar{J}_{6c}$ can be analyzed very similarly to $\bar{J}_{6b}$, leading to $\mathbb{E}\bar{J}_{6c}^2 = O(n^{-1} R_{\max}^2 \log(nT) \mathrm{app\_err}_v^2)$.

- Finally, we can express $\bar{J}_{6d}$ as an average $T^{-1} \sum_{t=1}^{T} \bar{J}_{6d}^{(t)}$ where each $\bar{J}_{6d}^{(t)}$ is given by

$$(\widehat{\theta}_{t,a,m} - \theta_{t,a,m}^*)^\top \Big( \frac{1}{n} \sum_{i=1}^{n} M_i \Big)(\widehat{\alpha}_{t,a,m} - \alpha_{t,a,m}^*),$$

where $M_i$s are i.i.d. mean zero matrices. Based on this identity, one can show that $\mathbb{E}(\bar{J}_{6d}^2)$ is of the order $O(n^{-1} R_{\max}^2 \log(nT) \mathrm{est\_err}_v^2)$, using similar arguments to those for the analysis of $\bar{J}_{6b}$.

Combining these results, one can apply arguments similar to those for the analysis of $J_4$ to show that $\mathbb{E}(J_6 - J_6^*)^2 = O(n^{-1} R_{\max}^2 \log(nT) \mathrm{err}_v^2)$. Together with $\mathbb{E}(J_6^*)^2 = O(R_{\max}^2 \mathrm{err}_w^2 \mathrm{err}_v^2)$ and the Cauchy-Schwarz inequality, we obtain that

$$\mathbb{E}(J_6)^2 = O\Big( \frac{R_{\max}^2 \log(nT) \mathrm{err}_v^2}{n} \Big) + O(R_{\max}^2 \mathrm{err}_v^2 \mathrm{err}_\omega^2).$$

The rest of the proof follows similarly to that of DRL. $\qquad\square$

