# OpenReview forum: "Unraveling the Interplay between Carryover Effects and Reward Autocorrelations in Switchback Experiments"
_ICML.cc/2025/Conference — ICML 2025 poster_

### Official Review · Reviewer_EcPH · 2025-03-12

**Overall Recommendation:** 4

**Summary:**

This work investigates the effectiveness of the switchback experimental design in A/B testing. The paper studies this design with multiple estimators and proves results that are estimator-agnostic. These results link the efficiency of the switchback design to the carryover effect and the correlation structure between rewards and help construct a guideline to when to use switchback experimentation instead of the alternating day design. The benefits are also confirmed by empirical experimentation.

## After rebuttal: I think that this work meets ICML standards and is worth sharing with the community. I raised my score to reflect this.

**Claims And Evidence:**

The main claim is supported by convincing evidence.

**Essential References Not Discussed:**

N/A.

**Experimental Designs Or Analyses:**

The experimental design looks valid.

**Methods And Evaluation Criteria:**

The evaluation criteria makes sense.

**Other Comments Or Suggestions:**

See questions below please.

**Other Strengths And Weaknesses:**

**Strengths:**
- The paper treats the important A/B testing problem and provides insight into the relatively new switchback experimental design and guidelines on when to use it.
- The paper is well-positioned compared to related work.
- The results derived are quite general and do not depend on a heavy assumption.
- The claims made are supported with interesting theoretical and empirical results.

**Weaknesses:**
- The paper can be hard to follow if the reader is not familiar with the literature.
- The paper focuses on the switchback and alternating day designs but does a poor job motivating them compared to vanilla A/B testing.

**Questions For Authors:**

I have a few questions to the authors:

- When there is no carryover effect, how does the standard A/B testing estimator (split the population, run the two policies independently on A/B populations) compare to alternative day/switchback? does the A/B testing estimator match the alternating day design (given the independence assumption)?

- Is the MSE really what we care about in these scenarios? Can we construct a valid confidence interval to use for decision making with the switchback estimator?

- The model proposed generalises the MDP framework but its limitations were not discussed. How can we sure that the insight derived from this model are transferable to real world scenarios?

I am leaning for an accept for this paper, I just want my questions to be clarified and I will reconsider my score.

**Relation To Broader Scientific Literature:**

This work improves our understanding of the switchback design as an alternative to A/B testing/Alternating day design. It uses the general MDP formulation to shed light on the value of the switchback design and provide practical guidelines to which strategy to adopt depending on the problem in hand.

**Theoretical Claims:**

I did not check fully the correctness of the proofs.

---

> ### Author Rebuttal · Authors · 2025-03-28
>
> > **Comparison against standard A/B**
>
> Excellent comment! First, we would like to clarify that different estimators apply to different use cases. Taking ridesharing as an example, there are three different types of experiments: (i) temporal randomization (over time), (ii) spatial randomization (across geographic areas), and (iii) user-level randomization (across drivers/passengers). Our primary focus is (i), which applies to the evaluation of order dispatch policies that must be implemented city-wide, making (ii) and (iii) unsuitable. Spatial randomization (ii) is typically used for testing localized subsidy policies in different regions, while user-level randomization (iii) applies when
> assigning personalized subsidies to individual users. Second, when constraining to temporal randomization, the population corresponds to the entire time horizon, with each time interval representing an individual unit. When no carryover effects exist and
> residuals are uncorrelated (satisfying i.i.d. assumptions), similar to Theorem 1, we can show that standard uniform randomization over time is equivalent to both AD and SB. This is because temporal ordering becomes irrelevant under the uncorrelatedness assumption. Similarly, when randomizing is conducted at the daily level rather than per time unit, standard procedures are equivalent to AD designs.
>
> We will be very happy to include these discussions in the paper shall it be accepted, to better connect AD, SB
> to vanilla A/B testing.
>
> > **MSE & confidence intervals (CIs)**
>
> This is another excellent comment. We agree that CIs and p-values are equally important, as A/B testing is essentially a statistical inference problem. Our designs are tailored to minimize the MSE of the resulting ATE estimators. A closer look at the proof of Theorem 1 suggests that the three RL-based estimators are asymptotically normal, and their MSEs are dominated by their asymptotic variances. As such, optimal designs minimizing the variance of the resulting ATE estimator also minimize the length of confidence intervals and maximize the power of the resulting hypothesis test.
>
> As we use RL-based estimators for A/B testing, existing methods developed in the RL literature are directly applicable for CI construction. These methods can be categorized into the following four types:
>
> 1. **Concentration-inequality-based methods** that construct CI based on concentration inequalities (Thomas et al., 2015, AAAI; Thomas & Brunskill, 2016, ICML; Feng et al., 2020, ICML)
>
> 2. **Normal-approximation-based methods** that utilize the asymptotic normality of the ATE estimator to construct Wald-type CIs (Luckett et al., 2020, JASA; Liao et al., 2021 JASA; Shi et al., 2021, ICML; Kallus and Uehara et al., 2022, OR, Liao et al., 2022, AoS; Shi et al., 2022, JRSSB; Wang et al., 2023, AoS)
>
> 3. **Bootstrap methods** that employ resampling for CI construction (Hanna et al., 2017, AAAI; Hao et al., 2021, ICML)
>
> 4. **Empirical likelihood methods** (Dai et al., 2020, NeurIPS).
>
> During the rebuttal, we conducted additional simulation studies, employed a non-parametric bootstrapping method to construct CIs and reported the coverage probability (CP) & the average CI width in these [plots](https://www.dropbox.com/scl/fo/knx6re4t6gzlh6911bzwe/AAvJOdKE1zy14xHbwqZtKlA?rlkey=e5pfd7grawxqt336a96sn9o35&st=guzcuroc&dl=0). It can be seen that most CPs are over 92%, close to the nominal level. Meanwhile:
>
> * For small values of $\lambda$, more frequent policy switch reduces the average CI width;
> * When $\lambda$ is increased to 10%, AD produces the narrowest CI on average.
>
> These results verify our claim that a reduction in MSE directly translates to a shorter CI.
>
> > **Limitations**.
>
> Our framework relies on the Markov assumption. We have discussed the validity of this assumption in our motivating ridesharing example and outlined several potential approaches to handle settings when this assumption is violated. Refer to our response #2 to Reviewer yQjJ.
>
> > **Transfer findings to real world scenarios**.
>
> In the discussion section, we proposed a three-step workflow for applying our theoretical framework to
> real-world experimental design:
>
> (i) To make sure our findings are transferable, the first step is to properly determine the time interval to satisfy the Markov assumption. When historical dataset is available, this assumption can be verified based on existing tests (Chen & Hong, 2012, Econ Theory; Shi et al., 2020, ICML; Zhou, et al., 2023,
> JRSSB).
>
> (ii) The second step is to examine the size of the carryover effects, which in practice can be estimated using company's simulator that evaluate new product's impact prior to deployment.
>
> (iii) The last step is to determine the correlation structure, which can again be achieved using the historical data.
>
> > **Presentations**.
>
> We will polish the paper to make it more accessible to general audience.

---

> > ### Comment · Reviewer_EcPH · 2025-04-04
> >
> > Thank you for the detailed rebuttal. I have a question about the decision making part:
> >
> > All of the methods you cited require the estimator to be unbiased, or else your estimator will concentrate around a biased quantity. It is good to have smaller confidence intervals but it's not a good sign if your estimator does not concentrate towards the true ATE value.
> >
> > ATE estimation via DRL can present biases when the value nor the importance weights are correctly specified. How can we be sure that they are correctly specified? In the other hand, do we know how this bias dissipates? Is the estimator consistent for example? Do we have theoretical results proving that one can omit this bias?

---

> > > ### Author Response · Authors · 2025-04-06
> > >
> > > Excellent comment. We first clarify that in theory, even when both the estimated value function and importance weights are misspecified, the DRL estimator can remain __asymptotically unbiased__ -- the bias is of order $o_p(n^{-1/2})$ and therefore much smaller than its root MSE -- provided that both estimated nuisance functions (e.g., value and importance weights) converge to their ground truth at a rate of $o_p(n^{-1/4})$. This finding is also implied by our proof of Theorem 1 in Appendix C.4, which we elaborate on below. In practice, these rate conditions can be met by using flexible machine learning models, e.g., neural networks, to estimate the nuisance functions. Our empirical study, reported in the [link](https://www.dropbox.com/scl/fo/parfie6b7vaft6rxnaomc/AH_wdfm5zAxK_oyGAlYXAps?rlkey=f3lew1k3hf74psbhkkx8drv24&st=j5hwjr1t&dl=0), also confirms that the __biases of both DRL and LSTD estimators are consistently much smaller than their root MSEs__, under the two data generating processes in our simulation studies. Even when the $o_p(n^{-1/4})$ condition is violated, one can adopt __DRL variants__ that either debias this estimator or account for model misspecification to __produce valid confidence intervals__, as we describe later.
> > >
> > > We next discuss the convergence rate condition, along with the intuition behind why DRL's bias remains negligible. The $o_p(n^{-1/4})$ condition is __mild__, as $o_p(n^{-1/4})$ is much slower than the standard parametric rate of $O_p(n^{-1/2})$ and practically feasible to achieve given the approximation capabilities of modern machine learning models. Similar rate conditions have been widely imposed in the causal inference and RL literature (see, e.g., Chernozhukov et al., 2018 ; Kallus and Uehara, 2020; Farrell et al., 2021).
> > >
> > > The reason DRL remains asymptotically unbiased under this condition lies in its Neyman orthogonality property (see Equation (1.8) in Chernozhukov et al., 2018), which ensures that DRL's bias due to estimation errors in the nuisance functions only appears at the __second order__. More precisely, the bias is proportional to the product of the estimation errors of the value function and the importance weights. This is formalized in Part II of our proof of Theorem 1-DRL in Appendix C.4, where the term $J_3$ in Equation (26) is shown to be upper bounded by the final term in Equation (30), which is indeed a __product__ of the two errors. Consequently, to ensure the bias remains $o_p(n^{-1/2})$, it is sufficient for both errors to decay at a rate of $o_p(n^{-1/4})$.
> > >
> > > Finally, we note that the $o_p(n^{-1/4})$ convergence rate condition can be further relaxed to $O_p(n^{-\kappa})$ for any $\kappa > 0$. This is a much weaker condition, as $\kappa$ can be arbitrarily small. In such cases, one can adopt the proposal by Shi et al. (2021) to deeply debias the DRL estimator, ensuring that its bias remains much smaller than its root MSE. Moreover, even in cases where the nuisance function estimators do not converge at all (i.e., $\kappa = 0$), one may still obtain valid confidence intervals by adopting the proposals by Jiang and Huang (2020) and Zhou et al. (2023). These methods enlarge the resulting confidence interval by explicitly accounting for model misspecification error to maintain valid coverage.
> > >
> > > **References**
> > >
> > > [1] Chernozhukov, V., Chetverikov, D., Demirer, M., Duflo, E., Hansen, C., Newey, W., & Robins, J. (2018). Double/debiased machine learning for treatment and structural parameters.
> > >
> > > [2] Kallus, N., & Uehara, M. (2020). Double reinforcement learning for efficient off-policy evaluation in markov decision processes. Journal of Machine Learning Research, 21(167), 1-63.
> > >
> > > [3] Farrell, M. H., Liang, T., & Misra, S. (2021). Deep neural networks for estimation and inference. Econometrica, 89(1), 181-213.
> > >
> > > [4] Shi, C., Wan, R., Chernozhukov, V., & Song, R. (2021, July). Deeply-debiased off-policy interval estimation. In International conference on machine learning (pp. 9580-9591). PMLR.
> > >
> > > [5] Jiang, N., & Huang, J. (2020). Minimax value interval for off-policy evaluation and policy optimization. Advances in Neural Information Processing Systems, 33, 2747-2758.
> > >
> > > [6] Zhou, W., Li, Y., Zhu, R., & Qu, A. (2023). Distributional shift-aware off-policy interval estimation: A unified error quantification framework. arXiv preprint arXiv:2309.13278.

---

### Official Review · Reviewer_yQjJ · 2025-03-23

**Overall Recommendation:** 3

**Summary:**

This work studies how carryover effects and autocorrelations influence switchback experiments in A/B testing. The authors conduct theoretical and experimental analyses and evaluate three main estimators using both synthetic and real-world datasets.

**Claims And Evidence:**

Yes.

**Essential References Not Discussed:**

No, the literature review is extensive.

**Experimental Designs Or Analyses:**

Yes, I checked the design of both experiments and the details in Appendices A and B.

**Methods And Evaluation Criteria:**

Yes.

**Other Comments Or Suggestions:**

1. This work could introduce more real-world scenarios with A/B testing data to verify the results.

2. Placing Fig.6 under Fig.5 may look better.

**Other Strengths And Weaknesses:**

**Strengths:**

> First, this work is well-structured and easy to follow. To my knowledge, this work is the first formal exploration of switchback experiments in A/B testing with common switchback strategies, demonstrating that the SB strategy can outperform the AD (alternating-day) design under specific conditions. The toy example effectively clarifies the motivation behind the research and subsequent experimental designs. Notably, the authors validated their findings using real-world data, observing trends consistent with theoretical predictions.


> Additionally, the paper provides actionable insights (the take-home message) that are valuable for both researchers and practitioners in the field.



**Weaknesses:**

> First, the definitions and quantitative measurements of carryover effects and reward autocorrelations are under-explored. Consequently, terms like "positively correlated," "negatively correlated," and "large carryover effect" remain somewhat subjective (though the metrics introduced in Section 4.2 could serve as a reference).



> Another limitation is the assumption of the Markovian property, which may not hold in real-world scenarios. Given the complexity of capturing complete market features and participants’ policies, the system is often partially observable, aligning more closely with a POMDP. Incorporating data from more realistic scenarios could refine the model’s applicability.

**Questions For Authors:**

1. The carryover effect seems to be the one-step effect when the new policy takes over the baseline policy. However, the new policy may have long-term impacts. For example, in ride-hailing scenarios, a coupon delivered to the customer may be consumed in the most expensive order, which may exceed the duration of a single policy in the SB experiment. Why do carryover effects only cover the single-step case?

2. If *δ* in Sec. 4.2 is close to 0, the new policy and baseline policy cause very similar transitions, and in other words, the transition is irrelevant to the action. However, we do not know the influence of a non-zero $\delta$. Could you provide some illustrations or measurements on the transitions under the new policy and the baseline when $\delta$ is fixed to zero and when $\delta$ takes on different non-zero values?

3. When $\lambda$ in Sec. 5.2 is set to 15\%, the results of AD are reversed when compared to Fig.3 and Fig.4. What are the results when $\lambda$ is set in the range (5\%,15\%)?

**Relation To Broader Scientific Literature:**

This work contributes to the policy deployment in the real world, thus potentially aiding the development of offline RL.

**Theoretical Claims:**

No, I only read the theorem and three corollaries in the main text.

---

> ### Author Rebuttal · Authors · 2025-03-28
>
> > **Subjective definitions**
>
> We did not define these terms to make the paper easy to follow. To address your comment, here are their precise mathematical definitions:
>
> * ''Positively (negatively) correlated'' refers to the covariance $\sigma_e(t_1, t_2)$ being positive (resp. negative);
> *  A ''large carryover effect" occurs when  $\delta 	\gg (R\_{\max} T)^{-2} \sum_{\substack{k_2-k_1=1,3,5, \ldots\\ 0\le k_1<k_2< T/m}} \sum_{l_1,l_2=1}^{m} \sigma_{e}(l_1+k_1m, l_2+k_2 m)$ so that Equation (5) will be dominated by the second term.
>
> > **Markov assumption (MA)**
>
> We thank the reviewer for highlighting the MA, which is critical for our RL-based estimators. In collaboration with our ride-sharing industry partner, we've observed that intervals of 30 minutes or 1 hour typically satisfy MA, showing strong lag-1 correlations with rapidly decaying higher-order correlations. This justifies the use of RL in our application.
>
> When applied to more general applications, we recommend to properly select the interval length to meet MA as an initial step in the design of experiments (see Fig. 6). If that's challenging, we further propose three approaches below, tailored to different degrees of violation of MA. Our current results directly extend to the first two cases, while the third case requires further investigation:
>
> 1.  **Mild violation**: Future observations depend on the current observation-action pair and a few past observations. This mild violation can be easily addressed by redefining the state to include recent past observations. With this modified state, MA is satisfied. Our RL-based estimators and theoretical results remain valid.
>
> 2. **Moderate violation**: Future observations depend on a few past observation-action pairs. Here, the RL-based estimators remain applicable if the state includes these historical state-action pairs. However, our theoretical results on optimal designs must be adjusted. Preliminary analyses show that, under weak carryover effects and positively correlated residuals, the optimal switching interval extends to 1+k (where k is the number of included past actions) rather than switching at every time step. This is because each observed reward is affected by a k+1 consecutive actions, not just the most recent action. More frequent switching under these conditions causes considerable distributional shift, inflating the variance of the ATE estimator.
>
> 3. **Severe violation**:  Data follows a POMDP. Although the existing literature provides doubly robust estimators and AD-like optimal designs (Li et al., 2023, NeurIPS) to handle such non-Markov MDPs, these estimators suffer from the "curse of horizon" (Liu et al., 2018, NeurIPS). Recent advances propose more efficient POMDP-based estimators (Liang & Recht, 2025, Biometrika) and designs (Sun et al., 2024, arXiv:2408.05342); however, these proposals are limited to linear models. Extending these methodologies to accommodate more general estimation procedures (e.g., Uehara et al., 2023, NeurIPS) represents an important direction for future research.
>
> We also remark that in the first two cases, existing tests are available for testing the Markov assumption and for order selection (Chen & Hong, 2012, Econ Theory; Shi et al., 2020, ICML; Zhou, et al., 2023, JRSSB).
>
> > **More real-world scenarios**
>
> Refer to our response #2 to Reviewer xXCu regarding other data sources.
>
> > **One-step effect**
>
> We appreciate this insightful comment and would like to clarify that our analysis does not assume a one-step carryover effect. Under the MDP framework, each action potentially influences all subsequent rewards throughout the trajectory, and our RL-based estimators explicitly account for these long-term effects. As illustrated in Figure 2, an action (e.g., $A_{t-1}$) affects not only the immediate next reward ($R_t$) but also subsequent rewards. Nevertheless, the MDP structure is first-order Markovian, implying that these delayed effects propagate entirely through the state $S_{t+1}$.
>
> > **Measuring $\delta$**
>
> $\delta$ can be measured using the size of delayed effects (e.g., $\lambda$), defined as the difference between the ATE and the direct effect. This is because under the MDP formulation, delayed effects are caused by differences in state transitions between different actions. Thus, these effects serve as natural measures for $\delta$. Practically, $\lambda$ can be estimated by computing the difference between our RL-based estimator for ATE and existing direct effect estimators (Dudik, 2014, Statistical Science).
>
> > **Results when $\lambda=$5% ~15\%**
>
> We conducted additional simulations using real datasets from the two cities and reported the [results](https://www.dropbox.com/scl/fi/h6pkz5y2pqybfodu5okn6/plot_all_rmse.png?rlkey=pq87oyi40xtlsbyxwqxyi8bp1&st=8zw4zrtv&dl=0) when $\lambda= $ (7.5%, 10%, 12.5%). Notably, as $\lambda$ increases, AD appears as the optimal design due to the increase in the carryover effect. This again aligns with our theory.

---

> > ### Comment · Reviewer_yQjJ · 2025-04-08
> >
> > Thanks for the clarification.

---

### Official Review · Reviewer_xXCu · 2025-03-24

**Overall Recommendation:** 3

**Summary:**

The paper explores various switchback experimental designs within MDPs, analyzing how carryover effects and reward autocorrelations affect estimator performance in A/B tests. The authors suggest a practical workflow for choosing designs based on the interplay of carryover effects and autocorrelation of rewards, demonstrating empirically (via simulations and real-world datasets) the conditions under which alternating-day (AD) designs or high-frequency switchbacks (SB) are more efficient.

**Claims And Evidence:**

Yes.

**Essential References Not Discussed:**

No.

**Experimental Designs Or Analyses:**

In real-data experiments (Section 5.2), the authors rely on a single ridesharing company's data, which inherently contains significant structural autocorrelations and daily cycles. Could the authors provide sensitivity analyses or robustness checks to other data sources?

**Methods And Evaluation Criteria:**

Yes.

**Other Comments Or Suggestions:**

No.

**Other Strengths And Weaknesses:**

No.

**Questions For Authors:**

Could authors provide a clearer comparison or benchmarking of their suggested RL-based estimators against more conventional estimators from econometrics that already handle autocorrelation and carryover effects effectively?

**Relation To Broader Scientific Literature:**

The paper makes a meaningful contribution by connecting RL-based policy evaluation with experimental design under temporal dependencies. It extends prior work by systematically analyzing how different estimators behave under varying carryover and autocorrelation conditions, which hasn’t been thoroughly explored before.

**Theoretical Claims:**

The correctness appears acceptable overall.

However, I found the three corollaries a little oversimplified. I was expecting to see some more realistic assumptions of covariance structures, like mixing conditions.

---

> ### Author Rebuttal · Authors · 2025-03-28
>
> **1. Realistic Covarince Structures** :
> In Corollaries 1–3, we focused on autoregressive, moving average, and exchangeable covariance structures primarily to derive closed-form expressions for the first term (AC(m)) in Equation (5). While these structures might seem simple, they are widely used in practice (Williams, 1952, Biometrika; Berenblut et al., 1974, Biometrika;  Zeger et al., 1988, Biometrika).
>
> During the rebuttal, we have relaxed these assumptions and obtained the following corollaries that allow more realistic covariance structures:
>
> **Corollary 4 (Truncated covariances)**. Suppose $\sigma_e(i,j)=0$ whenever $|i-j|> K$ but remains positive when $|i-j|\le K$. Then AC(m) decreases monotonically in $m$ for any $m\ge K$.
>
> **Corollary 5 (Decaying covariances)**. Suppose $\sigma_e$ is nonnegative and satisfies $\sigma_e(i,j)=O(|i-j|^{-c})$ for any $c>2$. Then for any $m_1\le m_2$, we have $AC(m_1) \ge AC(m_2) + o(T^{-1}m_1^{-1})$.
>
> The truncated covariance assumption in Corollary 4 is substantially weaker than the moving average assumption in Corollary 2. First, it does not require the covariance function to be stationary. Second, residuals are not required to be sums of white noise processes.
>
> Similarly, the decaying covariance assumption in Corollary 5 is considerably weaker than the autoregressive assumption in Corollary 1. Like Corollary 4, it does not require stationarity. Additionally, it accommodates polynomial decay of covariance functions with respect to the time lag, thus allowing for stronger temporal dependencies compared to the exponential decay assumed in Corollary 1. As a trade-off, Corollary 5 provides a weaker guarantee: monotonicity holds asymptotically, only when $m_1$ and $m_2$ are sufficiently large.
>
> > **2. Sensitivity Analysis and Other Data Sources**:
>
> Regarding Sensitivity Analysis:
>
> (i) In synthetic environments, we conducted a sensitivity analysis to assess the robustness of our results under four additional covariance structures: **moving average, exchangeable, uncorrelated, and autoregressive with a negative coefficient**. The findings, summarized in Figures S1 & S2 (Appendix B), consistently align with our theories.
>
> (ii) In the real-data analysis, we analyzed two datasets collected from different cities. We conducted a sensitivity analysis on the effect size, from **zero to small, moderate, and large**. Once again, these results support our theoretical conclusions.
>
> Regarding Other Data Sources:
>
> Publicly available experimental datasets suitable for our analysis are scarce. However, during the rebuttal, we found a publicly available [simulator](https://github.com/callmespring/MDPOD), which simulates a ridesharing market environment described in Xu et al. (2018, KDD). In this simulator, drivers and customers interact within a  9×9 spatial grid over 20 time steps per day. We used this simulator to conduct additional simulation studies, comparing the MDP-based order dispatch policy (Xu et al., 2018, KDD)  against a distance-based dispatch method that minimizes the total distance between drivers and passengers. We varied the number of days $n=28,35,42,49,56$ and set $m=1,5,10,20.$ For each scenario, we tested 100 orders with the number of drivers generated from a uniform distribution $U(40,45).$
>
> We report the RMSEs of ATE estimators computed under different designs, aggregated over 200 simulations, in this [link](https://www.dropbox.com/scl/fo/dp1j8bqnrkerckiz4mgvd/APksU1ySy9EI6DRjuIu67fE?rlkey=2s6hr0kk9ye0pau9qveapfnen&st=uz6alqna&dl=0), see the first plot. It can be seen that less frequent switching generally achieves smaller RMSE (m=10, 20). Further analysis shows that ATE is relatively small (~2%), suggesting weak carryover effects. Additionally, we observed negatively correlated reward residuals (see the second plot in the link). Under these conditions, our theory suggests that AD performs the best (see bullet point 3 on page 7), which aligns with the empirical results.
>
> > **3. RL-based vs Conventional Estimators**:
>
> We ran extra simulations to compare the RL-based estimator with three non-RL estimators:
>
> * Bojinov et al.'s (2023, Management Science) sequential importance sampling (IS) estimator, which handles carryover effects through time-dependent IS ratios;
>
> * Hu et al.'s (2022, arXiv:2209.00197) difference-in-mean estimator, which employs burn-in to discard rewards during policy transitions to handle carryover effects;
>
> * Xiong et al.'s (2024, arXiv:2406.06768) simple IS estimator that doesn't handle carryover effects.
>
> As shown in our [results](https://www.dropbox.com/scl/fi/51e0y436ekpbzmprdycxo/compare_LogMSE_all.png?rlkey=02cjwfxl4y3m659fg3j3zc21c&st=g66v0ps9&dl=0), RL-based estimators consistently outperform methods (i) and (ii) in all cases. Moreover, DRL and LSTD perform much better than (iii) in most cases.
>
> We deeply appreciate your thoughtful suggestions and will incorporate these analyses into our paper shall it be accepted.

---

### Decision · Program_Chairs · 2025-05-01

**Decision:**

Accept (poster)

**Comment:**

This work investigates the effectiveness of switchback experimental designs in A/B testing—specifically, the idea that the component being tested (e.g., a website) can be repeatedly switched between treatment and control over time to mitigate issues like non-independence. The authors explored various switchback designs and analyzed how carryover effects and reward autocorrelations influence the performance of different estimators in A/B tests. They also proposed concrete guidelines for selecting appropriate designs based on the interplay between carryover and autocorrelation, and empirically characterized (via simulations and real-world datasets) the conditions under which other designs may be more efficient.

All reviewers agree that this paper makes a meaningful contribution by connecting policy evaluation with experimental design under temporal dependencies. One reviewer pointed out that this paper nicely extends prior work by systematically analyzing the behavior of various estimators under different conditions. Another reviewer highlighted the paper's relevance to offline RL. All reviewers agree that the literature review is thorough and that this paper is well-structured and easy to follow. While a few concerns were raised about the datasets used to evaluate the proposed ideas, reviewers overall praised the experimental setup designed by the authors to validate their theoretical findings. Finally, two reviewers explicitly noted that all main claims made in this work are supported by convincing evidence and that the theoretical results are general and do not depend on overly strong assumptions.

Reviewers did raise a few concerns. One reviewer argued that although the paper’s theoretical contributions appear sound, three corollaries rely on potentially unrealistic assumptions. The authors responded by clarifying that these assumptions are widely used in practice (and provided relevant citations). They also relaxed some of these assumptions and derived new corollaries consistent with more general and realistic covariance structures. This reviewer also requested sensitivity and robustness analyses; the authors clarified that they had already assessed robustness (under four additional covariance structures) and that the corresponding results, shown in Appendix B, are consistent with their main theoretical claims. Additionally, the authors also ran new simulations comparing their RL-based estimator with three non-RL baselines, further confirming that RL-based estimators consistently outperform these alternatives.

A second reviewer noted that some terms in the paper—such as “positively correlated”—were never defined and “*(...) remain somewhat subjective*”. The authors provided precise mathematical definitions for all such terms, explaining that they were originally omitted due to space constraints. This reviewer was also concerned about the method's reliance on the Markov assumption, which they argued may not hold in practice. The authors acknowledged that this is a key assumption, noted that it is not uncommon, and explained how they empirically verified that it holds in the real-world datasets used to validate their claims. The reviewer responded that the authors' rebuttal addressed some of their concerns but did not specify which ones remained unresolved. They ultimately kept their score, primarily due to the concern about the Markov assumption, while also acknowledging that this is “*a good paper [that is] above the bar*”.

Finally, a third reviewer raised the question of whether Mean Squared Error (MSE) should be the key objective of interest in this setting, suggesting that valid confidence intervals might also be appropriate. The authors formally explained why their designs are tailored to minimize MSE and presented additional simulation results demonstrating that reductions in MSE also translate to narrower confidence intervals.

Although reviewers raised relevant concerns, the consensus is that this paper offers valuable theoretical and practical insights and that all key claims were adequately supported. Based on the reviewers’ feedback and the authors’ thorough rebuttal, I do not see any critical flaws, missing analyses or comparisons, or methodological issues. Importantly, the constructive discussion between authors and reviewers, along with the reviewers’ acknowledgment of the paper’s contributions, gives me confidence that the ICML community will benefit from the insights presented in this work.